# PERMUTATION-INVARIANT SPECTRAL LEARNING VIA DYSON DIFFUSION

## ABSTRACT

Diffusion models are central to generative modeling and have been adapted to graphs by diffusing adjacency matrix representations. The challenge of having up to $n!$ such representations for graphs with $n$ nodes is only partially mitigated by using permutation-equivariant learning architectures. Despite their computational efficiency, existing graph diffusion models struggle to distinguish certain graph families, unless graph data are augmented with ad hoc features. This shortcoming stems from enforcing the inductive bias within the learning architecture. In this work, we leverage random matrix theory to analytically extract the spectral properties of the diffusion process, allowing us to push the inductive bias from the architecture into the dynamics. Building on this, we introduce the Dyson Diffusion Model, which employs Dyson's Brownian Motion to capture the spectral dynamics of an Ornstein–Uhlenbeck process on the adjacency matrix while retaining all non-spectral information. We demonstrate that the Dyson Diffusion Model learns graph spectra accurately and outperforms existing graph diffusion models.

## 1 INTRODUCTION

Diffusion models are a key class of generative models based on noising data with Stochastic Differential Equations (SDEs) and learning their time reversal (Sohl-Dickstein et al., 2015; Song et al., 2021; Ho et al., 2020). They provide state-of-the-art results in many domains such as audio (Zhang et al., 2023) and vision (Croitoru et al., 2023). Generalizing diffusion models from Euclidean space to graphs offers promising applications in numerous areas, such as biology (Watson et al., 2023) or combinatorial optimization (Sun & Yang, 2023). However, while diffusing adjacency matrix representations is straightforward and popular (Niu et al., 2020; Jo et al., 2022; Vignac et al., 2022), this approach faces a major obstacle: Each graph with $n$ vertices has up to $n!$ representations as adjacency matrices. Therefore, if one aims to use a diffusion model on the space of matrices, one must learn $n!$ representations *per graph*. This is not feasible. Previous works tackled this problem by shifting the inductive bias of permutation invariance to the learning algorithm: If the neural network was permutation equivariant, training on one of the (up to $n!$ many) matrix representations would suffice. For example, Niu et al. (2020) and Jo et al. (2022) used message-passing graph neural networks (GNNs) while ConGress (Vignac et al., 2022) applied graph transformers. However, these learning architectures have a "blind spot" detailed below.

**Theoretical Limitations.** The "blind spot" arises from the limited ability of these models to solve Graph Isomorphism (GI): determining if two graphs are structurally identical regardless of node labeling. While permutation equivariance ensures that the model produces consistent outputs for different representations of the same graph, it does not guarantee that the model can differentiate between two structurally different (non-isomorphic) graphs. The failure is a result of how these architectures aggregate information: permutation-symmetric operations – message passing in GNNs and self-attention in graph transformers – can collapse distinct graphs with similar neighborhoods to the same representation, treating them as identical. More formally, GI is a challenging problem in algorithm theory, and it remains unknown whether GI $\in$ P (Babai, 2016). Since a polynomial time (learning) algorithm perfectly distinguishing all graphs would prove GI $\in$ P, contemporary (polynomial-time) graph learning algorithms must compromise on expressivity.

**Extracting Permutation-Invariant Information from Graph Diffusion.** When diffusing an entire adjacency matrix, state-of-the-art work pushes the entire inductive bias into the learning algorithm

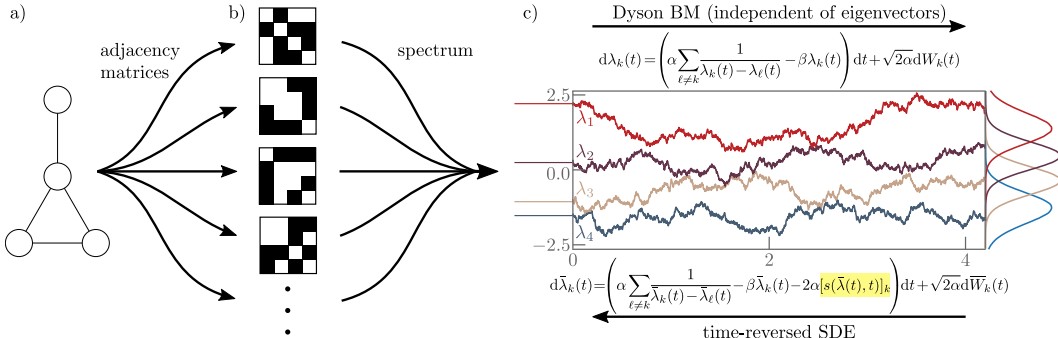

Figure 1: Dyson Diffusion model and its application to graph spectra: A graph on $n$ vertices (a) has up to $n!$ representations as adjacency matrices (b). For an OU-driven diffusion on *any* adjacency matrix, the permutation-invariant spectrum (c) evolves according to the same SDE (Dyson-BM). An exemplary path of the $n$, non-intersecting, eigenvalues is shown. The marginals of the invariant density for the $\lambda_k$ are depicted on the far right. The DyDM diffusion model learns the score $s(\lambda, t)$ (highlighted in yellow) to generate spectra via the time-reversed SDE (5).

(Niu et al., 2020; Jo et al., 2022) with possible data augmentation (Huang et al., 2022; Vignac et al., 2022; Xu et al., 2024). However, as we show below, this is neither necessary nor desirable (see the previous discussion on blind spots). Using techniques from random matrix theory, we show that an Ornstein–Uhlenbeck (OU) diffusion on the graph can be *dissected* into diffusion of the (permutation invariant) spectrum and diffusion of the (permutation-dependent) eigenvectors. Our method therefore allows us to learn the spectrum while preserving all remaining information. Moreover, since the spectrum is inherently permutation-invariant, we can parameterize the score using a much broader range of learning architectures, expanding the scope to architectures able to distinguish between graphs that are equivalent in the Weisfeiler-Leman (WL) sense (Morris et al., 2019; Xu et al., 2018).[1]

**Information in the Spectrum.** The spectrum of a graph encodes key structural features including connectivity, expansion, and subgraph patterns. Moreover, non-isomorphic WL-equivalent graphs typically have distinct spectra (Huang & Yau, 2024). An idea, therefore, is to augment the graph data based on spectral information (Vignac et al., 2022; Xu et al., 2024). Our work is based on an entirely different method, exploiting analytical insight from random matrix theory to dissect the spectral from the remaining information, allowing to push the inductive bias from the architecture to the dynamics.

**Dyson's Brownian Motion.** For an OU process on the space of symmetric matrices, the eigenvalues follow a well-characterized stochastic differential equation (SDE), the so-called Dyson Brownian Motion (DBM), which is inherently permutation invariant. Thus – in contrast to Niu et al. (2020); Jo et al. (2022); Vignac et al. (2022) – the score of DBM can be parameterized with any (not necessarily permutation invariant) neural network. Moreover, contrary to Luo et al. (2024), the remaining information is not lost (see Theorem 3.2).

**Contributions.** The main contributions of this work are as follows.

- We introduce the novel Dyson Diffusion Model (DyDM) in Section 3, which extracts the spectral dynamics from an OU-driven diffusion. DyDM allows to learn the spectra of graphs without the need of GNNs or graph transformers while preserving the remaining information of the graph and allowing us to compute an eigenvector SDE.

- We demonstrate in Section 4 that DyDM is more effective than existing GNN-based and graph-transformer-based methods for learning graph spectra.

- We illustrate the struggle of GNN-based graph diffusion models in Figure 2 and Section 4.1.

---

[1]We give a theoretical discussion in form of the WL-equivalence class in Section 2 and demonstrate the challenge of those architectures empirically in Figure 2.

Beyond graphs, our framework applies generally to symmetric matrices where spectral information is often key. For instance, in statistics, it can encode the importance of principal components (James et al., 2023), while in dynamical systems,[2] it reflects the stability and timescales of linear operators.

**Notation.** We work on the set of symmetric real matrices $\mathrm{Sym}(\mathbb{R}^{n \times n}) := \{A \in \mathbb{R}^{n \times n} : A^T = A\}$. The positive integers up to $n$ are denoted by $[n] := \{1, \ldots n\}$. We consider undirected graphs $G = (V, E, w)$ where $V$ is a finite set with edges $E \subseteq \{S \subseteq V : 1 \leq |S| \leq 2\}$ allowing for self-loops and weights $w : E \to \mathbb{R}$. The family of such graphs of size $n$ is $\mathcal{G}^n := \{G = (V, E, w) : |V| = n\}$. For a graph $G \in \mathcal{G}^n$ with $\mathbb{R}^V = \{f : V \to \mathbb{R}\}$ being the space of functions from $V$ to $\mathbb{R}$, we let $A \equiv A(G) : \mathbb{R}^V \to \mathbb{R}^V$ be the adjacency *operator*, defined for $f \in \mathbb{R}^V$ and $v \in V$ as the weighted sum of $f$ applied to the neighbors of $v$ as

$$Af(v) := \sum_{\{u,v\} \in E} w(\{u,v\}) f(u).$$

As the graphs are undirected, the adjacency operator is self-adjoint with respect to the standard inner product on $\mathbb{R}^V$. The operator $A$ therefore has $n$ real eigenvalues $\lambda_1 \geq \lambda_2 \geq \ldots \geq \lambda_n$. We denote the ordered spectrum of the adjacency operator by

$$\lambda(G) := \lambda(A(G)) := \{(\lambda_1, \lambda_2, \ldots, \lambda_n) : \lambda_1 \geq \ldots \geq \lambda_n\}.$$

Unless defined explicitly otherwise, we use $n \in \mathbb{N}$ for the (vertex) size of the graph, and $N \in \mathbb{N}$ for the number of samples. We denote by $s \sim \mathcal{N}(0, I_d)$ that the $d$-dimensional random vector $s$ has multivariate normal distribution with 0 mean and unit covariance $I_d$.

## 2 LIMITATIONS OF EXISTING DIFFUSION MODELS

We consider the following matrix-valued OU SDE starting from some data $M(0) \in \mathrm{Sym}(\mathbb{R}^{n \times n})$ given for $1 \leq i \leq j \leq n$ by

$$\mathrm{d}M_{ji}(t) = \mathrm{d}M_{ij}(t) = -\beta M_{ij}(t)\mathrm{d}t + D_{ij}\mathrm{d}B_{ij}(t), \tag{1}$$

with diffusion coefficient $D_{ij} := \sqrt{(1 + \delta_{ij})\alpha}$ for any constants $\alpha, \beta \in \mathbb{R}^+$, where $B_{ij}(t) = B_{ji}(t)$ are independent Brownian motions and $\delta_{ij} = 1$ if and only if $i = j$, and 0 otherwise. We consider eq. (1) on the space of symmetric matrices to represent undirected graphs $G \in \mathcal{G}^n$.

Equation (1) is an entry-wise OU process[3] that preserves the symmetry of the matrix. In standard diffusion models we run eq. (1) until some time $T > 0$ from data samples. To fix notation, denote by $p_t$ the distribution of $M(t)$ induced by (1) for $t \geq 0$.

The time-reversal of the diffusion (1) over the time interval $[0, T]$ is a diffusion initialized by sampling $M(T) \sim p_T$ and satisfying for $1 \leq i \leq j \leq n$

$$\mathrm{d}M_{ij}(t) = -\left\{\beta M_{ij}(t) + D_{ij}^2 \left[s(M(t), t)\right]_{ij}\right\} dt + D_{ij}\mathrm{d}\bar{B}_{ij}(t), \tag{2}$$

for independent Brownian motions $\bar{B}_{ij}(t)$ (Anderson, 1982; Song et al., 2021). In Equation (2), $s(M, t)$ represents the score matrix at time $t \in (0, T]$, i.e., $s(M, t) := \boldsymbol{\nabla}_M \log p_t(M)$. This score is intractable but, as the OU process has tractable Gaussian transition densities, we can obtain an estimate $s_\theta(M, t)$ of it by minimizing the denoising score matching loss (Song et al., 2021)

$$L(\theta) = \mathbb{E}_{t \sim \mathrm{Unif}[0,T], M(0) \sim p_0, M(t) \sim p_{t|0}(\cdot|M(0))} \left[\left\|s_\theta(M(t), t) - \boldsymbol{\nabla}_{M(t)} \log p_{t|0}(M(t)|M(0))\right\|_2^2\right]. \tag{3}$$

Approximate samples of $p_0$ can then be obtained by simulating an approximation of Equation (2) obtained by sampling $M(T)$ from the Gaussian invariant distribution of (1), that is, $M_{ij}^{\mathrm{inv}} \sim \mathcal{N}(0, \alpha(1 + \delta_{ij})/2\beta)$, and using $s_\theta(M, t)$ in place of $s(M, t)$.

**Challenges posed by graphs.** When working with graphs, to obtain an adjacency matrix for a given weighted graph $G = (V, E, w) \in \mathcal{G}^n$, we need to fix an ordering $(v_1, \ldots, v_n)$ of all vertices. In

---

[2]For instance, Markov jump dynamics in detailed balance systems where the generator is symmetric in the steady state basis, see Pavliotis (2014).

[3]i.e. a diffusion with linear restoring drift, $-\beta M_{ij}(t)$ in eq. (1), pushing the dynamics back to its mean.

fact, given the ordering $(v_1, \ldots, v_n)$, the associated matrix $M = (m_{ij})_{1 \leq i,j \leq n} \in \mathrm{Sym}(\mathbb{R}^{n \times n})$ has entries $m_{ij} = w(\{v_i, v_j\})$ if $\{v_i, v_j\} \in E$ and 0 otherwise. The challenge is that the adjacency operator $A(G)$ for a given graph $G \in \mathcal{G}^n$ admits up to $n!$ distinct adjacency matrices. For instance, in Figure 1 we see four different matrix representations of the same graph.

**Why should we enforce this inductive bias?** One could think that the reason for using the inductive bias stems from wanting that graph generative models assign uniform probability to each of the (up to) $n!$ many representations. But this can be easily achieved by applying a random independent permutation to the output of the generative model.

Instead, the challenge stems from the following problem: In diffusion models, we learn a function (i.e. the score) on a set of graphs, say[4] $\Omega$, rather than the distribution directly. Learning on the adjacency representations would correspond to learning on $\Omega \times S_n$. We demonstrate on a toy example in Corollary M.1 that *not* leveraging the inductive bias, i.e. learning on a space of size $\Omega \times S_n$, leads to an explosion of the average mean squared error: Learning a function on a fixed number of $k$ objects (say unweighted graphs on $n$ nodes) from $N$ samples leads to a mean squared error of order $\Theta(n!/N)$. In contrast, making use of the inductive bias leads to an average mean squared error of $\Theta(1/N)$. Noting that already on $n = 10$ nodes, we have $n! > 3 \cdot 10^6$, we see the clear argument *for* using the inductive bias.

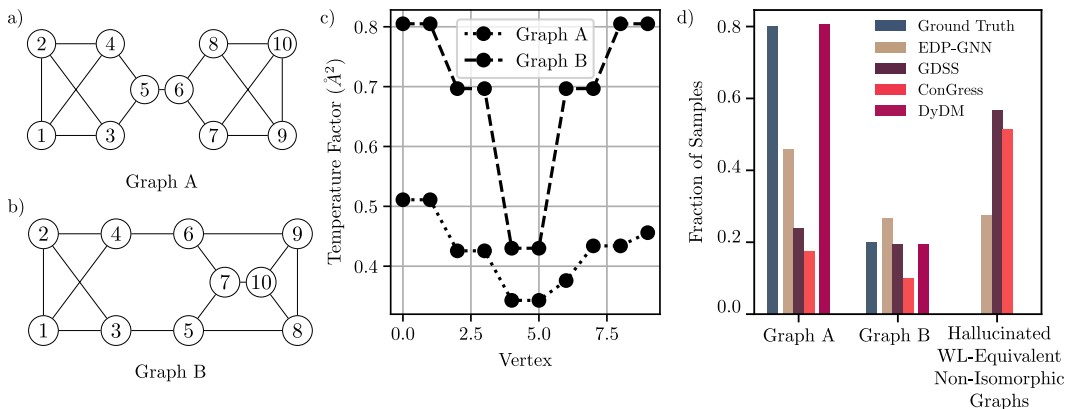

Figure 2: Struggle of GNN-based and graph-transformer-based models with two WL-equivalent graphs: Graphs A and Graphs B are WL-equivalent, but non-isomorphic. Also physically, they have very different properties, such as different temperature factors (c) and a different cut size. Upon training on a 80% Graph A and 20% Graph B dataset, state-of-the-art GNN-based (EDP-GNN,GDSS) and graph-transformer-based (ConGress) models learn the WL-equivalence class quickly but fail to generate the underlying distribution among the two graphs, with some even predominantly hallucinating WL-equivalent but non-isomorphic graphs (d).

**Cost of pushing inductive bias entirely into architecture.** One solution would be to impose the inductive bias in the learning architecture. This is what state-of-the-art graph diffusion models do (Niu et al., 2020; Jo et al., 2022; Vignac et al., 2022).

However, using these architectures comes at a cost. As argued in the introduction, since $\mathrm{GI} \in \mathrm{P}$ remains unknown, we expect some limitations. In the case of GNNs this compromise has been precisely characterized: GNNs cannot distinguish between the large families of so-called Weisfeiler-Leman (WL) equivalent graphs (Morris et al., 2019; Xu et al., 2018). For example, all $k$-regular graphs on $n$ vertices for any fixed $k \in \mathbb{N}$ are WL-equivalent (see Lemma 2.1 below and its proof in Appendix N) and therefore indistinguishable for these architectures, which may lead to hallucination.

**Lemma 2.1** (WL-equivalence of $k$-regular graphs). *For every fixed $n, k \in \mathbb{N}$, all $k$-regular graphs $G \in \mathcal{G}^n$ are WL equivalent. Moreover, every graph $G \in \mathcal{G}^n$ that is WL equivalent to a $k$-regular graph is $k$-regular.*

This is a vast class, since e.g. on $n = 20$ vertices, there are $510'489$ many non-isomorphic, connected 3-regular and thereby WL-equivalent graphs (Meringer, 1999). In particular, we demonstrate on a

---

[4]Importantly, $\Omega$ refers to *the set of graphs* and *not* the set of (permutation-sensitive) adjacency matrices.

simple example in Figure 2 that both GNN- and graph-transformer-based methods fail to learn a distribution on two particular WL-equivalent graphs: during sampling, they either fail to learn the distribution on both WL-equivalent graphs or hallucinate other, WL-equivalent but non-isomorphic, graphs (Fig. 2d). This can also be seen during learning: The EDP-GNN model learns the WL-equivalence class quickly (after 500 epochs) but then keeps hallucinating non-isomorphic but WL-equivalent graphs, and in particular struggles to learn the right distribution between graphs $A$ and $B$ for the remaining 4500 epochs (see Appendix R for details). Importantly, those graphs are very different. For instance, if the graphs represented Gaussian Network Models for macromolecules (Tirion, 1996), physical observables such as the temperature factors in X-ray scattering (Haliloglu et al., 1997) would be clearly distinct, see Fig. 2c. Therefore, a diffusion model based on GNNs will suffer from this expressivity blind spot. More generally, any diffusion model relying on a graph-specific learning algorithm will have limited expressivity.

# 3 DYSON DIFFUSION MODEL

## 3.1 DYSON BROWNIAN MOTION

Dyson (1962) showed that the spectrum of eq. (1) behaves as $n$ positively charged particles in a one-dimensional Coulomb gas. These particles exhibit Brownian motion, but with a pairwise repulsion force proportional to their inverse distance so that their paths do not cross (see Fig. 1c). More precisely, Dyson proved that the spectrum of the entry-wise Ornstein-Uhlenbeck process from eq. (1) follows the SDE given in Theorem 3.1. Without loss of generality, we restrict the domain to the *Weyl Chamber* $C_n := \left\{ \lambda \in \mathbb{R}^n : \lambda_1 > \ldots > \lambda_n \right\}$.

**Theorem 3.1** (Eigenvalue SDE, Dyson (1962))**.** *Denote by* $\lambda(t) = (\lambda_1(t), \ldots, \lambda_n(t))$ *the ordered spectrum of the matrix-valued Ornstein-Uhlenbeck process* $M(t)$ *of eq.* (1). *Then assuming that the initial matrix* $M(0)$ *has simple spectrum,* $\lambda(t)$ *satisfies for all* $1 \leq k \leq n$ *the stochastic differential equation*

$$\mathrm{d}\lambda_k(t) = \left( \alpha \sum_{\ell \neq k} \frac{1}{\lambda_k(t) - \lambda_\ell(t)} - \beta \lambda_k(t) \right) \mathrm{d}t + \sqrt{2\alpha}\mathrm{d}W_k(t), \qquad \text{(Dyson-BM)}$$

*for* $W_1, \ldots, W_n$ *independent standard Brownian motions. Moreover the unique stationary distribution of* (Dyson-BM) *has density*

$$p_{\mathrm{inv}}(\lambda) = \frac{1}{Z} \exp(-U(\lambda)) \qquad for \qquad U(\lambda) = \frac{\beta}{2\alpha} \sum_k \lambda_k^2 - \sum_{k < \ell} \ln |\lambda_k - \lambda_\ell|, \qquad (4)$$

*for* $\lambda \in C_n$ *and* $Z$ *a normalizing constant so that* $p_{\mathrm{inv}}$ *corresponds to a probability measure.*

For completeness, a full proof of Theorem 3.1 is given in Appendix B, where we generalize a well-known proof to arbitrary coefficients $\alpha, \beta$. We note that the assumption of Theorem 3.1 that $M_0$ has simple spectrum is minor. Indeed, generic random graphs or matrices have simple spectra, and in the case of eigenvalues with higher multiplicity, we can perturb the spectrum to be simple.
From Dyson-BM we see that the eigenvalues perform a Brownian motion in a confining potential with a repulsion force: once a pair of eigenvalues $\lambda_k, \lambda_l$ comes too close, they become repelled with a force $\alpha/(\lambda_k - \lambda_l)$ inversely proportional to their separation. A remarkable property of Theorem 3.1 is that the evolution of the spectrum is decoupled from all other information about the matrix: the spectral SDE (Theorem 3.1) is independent of the eigenvectors. This is the key analytical insight that motivates our approach.
Furthermore, conditioned on the eigenvalues, the remaining information captured in form of the eigenvectors can be deduced as we show in Theorem 3.2. This generalizes a statement of Allez et al. (2014), and we give a proof in Appendix C.

**Theorem 3.2** (Eigenvector SDE)**.** *Denote by* $(v_1(t), \ldots, v_n(t))$ *the orthonormal eigenvectors associated to the eigenvalues of Theorem 3.1. Assuming that the initial matrix* $M(0)$ *has simple spectrum,* $v_k(t)$ *satisfies for* $k \in [n]$ *the stochastic differential equation*

$$\mathrm{d}v_k(t) = -\frac{\alpha}{2} \sum_{\ell \neq k} \frac{1}{(\lambda_k(t) - \lambda_\ell(t))^2} v_k(t) \mathrm{d}t + \sqrt{\alpha} \sum_{\ell \neq k} \frac{1}{\lambda_k(t) - \lambda_\ell(t)} v_\ell(t) \mathrm{d}w_{\ell k}(t)$$

$$\text{(Eigenvector-SDE)}$$

*for $\{w_{ij:i\neq j}\}$ standard Brownian motions independent of the eigenvalue trajectories, with $w_{ji} = w_{ij}$.*

## 3.2 FROM THE DYSON SDE TO A DIFFUSION MODEL

Despite its advantages, constructing a diffusion model based on Dyson-BM poses several challenges, e.g. dealing with a singular drift, non-Gaussian conditional density, etc. (see Appendix L for details). As described below, with DyDM we overcome these obstacles and design an efficient diffusion model for the spectrum, which can distinguish between spectra of graphs that GNNs are blind to (Fig. 2) and which does *not* require ad hoc data augmentation.

The time-reversal of the Dyson-BM in the sense of Anderson (1982) reads

$$
\mathrm{d}\bar{\lambda}_k(t) = \left[ \alpha \sum_{\ell \neq k} \frac{1}{\bar{\lambda}_k(t) - \bar{\lambda}_\ell(t)} - \beta\bar{\lambda}_k(t) - 2\alpha[s(\bar{\lambda}(t), t)]_k \right] \mathrm{d}t + \sqrt{2\alpha}\mathrm{d}\overline{W}_k(t), \tag{5}
$$

where we aim to learn the score $s(\lambda, t) \coloneqq \boldsymbol{\nabla}_\lambda \log p_t(\lambda)$. Because the coefficients in Dyson-BM are non-Lipschitz, the applicability of Anderson (1982) is not immediate. Accordingly, in Appendix D we sketch a proof of existence and uniqueness of a strong solution and verify that Anderson's time reversal applies.

**Making the loss tractable.** Learning the loss function $s(\lambda, t)$ as in eq. (3) is not feasible for the Dyson-BM, since a closed form of the conditional distribution $p_{t|0}$ is not known in contrast to the OU process. To overcome this, we follow a derivation in the style of (De Bortoli et al., 2022) to obtain the loss function – up to constants in $\theta$ – for any $h \in \mathbb{R}^+$

$$
\tilde{L}(\theta) = \mathbb{E}_{t\sim\mathrm{Unif}[0,T],\lambda_t\sim p_t,\lambda_{t+h}\sim p_{t+h|t}(\cdot|\lambda_t)} \left[ \left\| s_\theta(\lambda_{t+h}, t+h) - \boldsymbol{\nabla}_{\lambda_{t+h}} \log p_{t+h|t}(\lambda_{t+h}|\lambda_t) \right\|_2^2 \right], \tag{6}
$$

where we will approximate the intractable $p_{t+h|t}$ with the Gaussian transition of the Euler-Maruyama approximation (see Appendix A) for details.

**Handling singularities.** Numerical solutions of Dyson-BM with a fixed step size are not practical, since Dyson-BM is singular at the boundary of the Weyl Chamber. A fixed step size leads to inaccuracies at the boundaries and may overshoot the singularity, leaving the Weyl Chamber. To overcome this, we implement an adaptive step-size algorithm which conditions on an event of probability 1 (non-crossing) and hence does not change marginal densities.[5] The step-size controller is described in Algorithm 2 and in Appendix F.1 in detail.

**Schedule.** Dyson's conjecture states that for $\alpha = \frac{1}{n}$ and $\beta = \frac{1}{2}$ Dyson-BM converges to the invariant distribution in time $\Theta(1)$, while the majority of eigenvalues mixes *locally* already in time $\Theta(1/n)$ (Yang, 2022). Hence, the fine-grained structure will be mixed in the time interval $(0, \Theta(1/n))$. We show in Appendix E through time change that this conjecture can be applied to any choice of coefficients $\alpha, \beta$ as. It is thus sensible to choose an exponential schedule. We specify the particular choices in Appendix K.1.

**Equilibrium shooting mechanism.** During inference, we require access to the learned score. In the forward diffusion, solely numerical errors due to the time discretisation may lead to crossings of singularities. On the one hand, when going backwards in time, due to inconsistencies in the learned score, the repulsion in eq. (5) might be too weak and the sample path may leave the Weyl Chamber for any sensible step size. Since the score is not defined outside the Weyl Chamber, this is problematic. On the other hand, the step size obtained by conditioning on not leaving the Weyl Chamber in this ill-trained point would be so small that the numerical solver would get stuck. To overcome this, we incorporate a shooting mechanism: If the repulsion force is too weak to prevent crossing of the singularity, resulting in a microscopic step size upon conditioning to remain in the Weyl Chamber, we repel with the invariant-state drift, i.e., we replace the learned score with the score in the invariant state (see Appendix G). This mechanism ensures that we stay in the Weyl Chamber, while minimizing its impact on the distribution. With well-tuned parameters, the shooting

---

[5]Note that the adaptive step-size is only used in the forward simulation, whereas the objective is evaluated always on a **fixed** grid to ensure correctness.

---

**Algorithm 1** DyDM training

---

1: **Input:** spectral samples $\lambda^{(1)}, \ldots, \lambda^{(N)} \in \mathbb{R}^n$, schedule $\mathcal{T} = \{t_j\}$
2: **for** each sample $i \in [N]$ **do**
3:      $t \leftarrow 0$
4:      $\lambda(t) \leftarrow \lambda^{(i)}$
5:      **while** $t < T$ **do**                                            ▷ Diffuse sample
6:          Let $u \sim \mathcal{N}(0, I_n)$
7:          $\delta t \leftarrow$ ForwardStepsizeController$(\lambda(t), u)$     ▷ Conditions on non-intersection, see F.1.
8:          **if** $\delta t < \delta t_{\min}$ **then**
9:              Skip step                                 ▷ Rare event: Skip tiny steps
10:          **end if**
11:          $\delta t \leftarrow \min\{\delta_t, t_{\text{fix}} - t\}$ where $t_{\text{fix}} = \min\{t_j \in \mathcal{T} : t_j > t\}$
12:          $\lambda(t + \delta t) \leftarrow$ Euler-Maruyama step of Dyson-BM with step size $\delta t$ and noise $u$
13:          $t \leftarrow t + \delta t$
14:      **end while**
15: **end for**
16: Update $s_\theta$ using loss $\tilde{L}(\theta)$ along paths $\lambda^{(1)}(t), \ldots, \lambda^{(N)}(t)$ on schedule $\mathcal{T}$ using eq. (20)
17: **Output:** $s_\theta$.

---

Figure 3: Dyson Diffusion Model (training): The Dyson-BM is evolved forward in time with an adaptive step size ensuring that the paths remain in the Weyl Chamber. The step-size controller conditions on the probability 1 event of non-crossing as detailed in Appendix F.1 and Algorithm 2.

gets rarely triggered (less than $0.5\%$ of steps) but is essential, as already a single event would cause getting stuck (upon conditioning) or leaving the Weyl Chamber.

**Sampling from Invariant Distribution** To sample from the invariant distribution of $\lambda(t)$ given by eq. (4) we exploit the connection between $\lambda(t)$ and $M(t)$: We first sample from the Gaussian invariant distribution of $M(t)$ and then perform an eigendecomposition.

**Comparison to direct simulation.** One may wonder why we do not sample from the matrix-valued OU process at any time $t$ directly, then perform an eigendecomposition to get $\lambda(t)$, determine $\lambda(t + dt)$, and learn the score network from the increment using eq. 6. This way, learning would be simulation-free and the Dyson SDE would only be needed for the derivation of the loss function as well as the backwards dynamics. The reason for *not* pursuing this is efficiency and precision. We found that the above way takes 150 times longer for graphs of size $n = 10$, since (accurate) eigendecomposition is computationally costly. Doing so at every step explodes the costs. Moreover, note that we train on the sample of increments, so that from a simulation until time $t$, we learn *on the entire generated path*. Through our efficient implementation of the SDE, the learning process has running time (up to smaller time steps performed by the adaptive step size controller) on the order of a simulation-free diffusion model.

## 4 EXPERIMENTS

We empirically evaluate DyDM against several state-of-the-art graph diffusion models. First, we compare them on a simple bimodal distribution between two WL-equivalent graphs, illustrating the struggle of purely GNN-based and graph-transformer-based methods (Section 4.1). Next, we carry out a comparison on a standard graph benchmark datasets (Section 4.1) and demonstrate scalability on a larger dataset ($15'000$ graphs).

### 4.1 METHODOLOGY

We compare to the GNN-based models EDP-GNN (Niu et al., 2020) and GDSS (Jo et al., 2022), as well as graph-transformer-based ConGress and DiGress (Vignac et al., 2022), where among all the models only DiGress uses a data-augmentation trick: it adds certain graph features, including cycle

Table 1: Statistical distances of DyDM compared to standard models: DyDM learns the spectrum better in both the $n$-dimensional mean and the marginal Wasserstein sense, without requiring ad hoc data augmentation. Results are rounded to two decimal places, and results ($*$) are equal until the fourth decimal place, and ($**$) until the third decimal place. Exact numbers are provided in Appendix Q.6.

| Dataset | WL-Bimodal | | Community Small | | Brain | |
|---|---|---|---|---|---|---|
| Distance | $\mu$ | $\mathcal{W}_{\mathrm{marg}}$ | $\mu$ | $\mathcal{W}_{\mathrm{marg}}$ | $\mu$ | $\mathcal{W}_{\mathrm{marg}}$ |
| DyDM (ours) | **0.02** | **0.01**$^*$ | **0.07** | **0.02** | **0.05** | **0.03**$^{**}$ |
| EDP-GNN | 0.13 | 0.08 | 0.42 | 0.14 | 0.07 | 0.03$^{**}$ |
| GDSS | 0.23 | 0.13 | 0.42 | 0.14 | 0.33 | 0.12 |
| ConGress | 0.38 | 0.16 | 0.27 | 0.11 | 0.13 | 0.03$^{**}$ |
| DiGress (no trick) | 1.06 | 0.29 | 2.51 | 0.45 | 0.57 | 0.17 |
| DiGress (trick) | 0.03 | **0.01**$^*$ | 0.09 | 0.03 | 0.12 | **0.03**$^{**}$ |

counts and the first 6 eigenvalues (for details, see Section 5), but this *trick* is supposedly inessential for building a good model (Vignac et al., 2022). The trick, however, improves the learning of certain features, but not necessarily other subgraph structures (Wang et al., 2025). We thus compare to both the Digress model "without the trick", i.e., just a graph (Markov chain) diffusion model, and with the data-augmentation trick.

As we evaluated the full spectrum and the published experimental results only showed partial or no information about the spectrum, we need explicit access to the samples. Since not all models (Vignac et al., 2022; Niu et al., 2020) report snapshots, we had to retrain those also on the standard datasets. We report our code and all the samples of all models in Github[6].

**WL-Bimodal.** Here we train the simple bimodal distribution in Fig. 2, consisting of 80% graph A and 20% graph B, which are WL equivalent. The dataset has $N = 5'000$ random permutations of A and B, and we follow the standard test/train split procedure (Jo et al., 2022; You et al., 2018; Niu et al., 2020) using 80% of the data as train data and the remaining 20% as test data. We performed hyperparameter tuning of the comparison models as described in Appendix Q.

**Community.** Being a standard benchmark (Niu et al., 2020; Jo et al., 2022; You et al., 2018), we include it for comparison. However, due to heavy undersampling, we only test for memorization. From our perspective, memorization is the best that can be tested with said benchmark, and we elaborate on this in Appendix Q.1

**Brain.** From the human connectome graph we drew $15'000$ ego-graphs (i.e. the induced subgraph of neighborhoods) of size 5 to 10 vertices Amunts et al. (2013); Rossi & Ahmed (2015). Crucially, this dataset demonstrates scalability to a graph number large enough for high statistical fidelity (faithful representation of the underlying distribution) in test and train set. For details, see Appendix P.

## 4.2 RESULTS

In Table 1 we report mean distances and marginal Wasserstein distances of the spectra. Explicitly, if $\nu_{\mathrm{test}}$ is the distribution of the spectrum of the test dataset and $\nu_{\mathrm{samp}}$ the distribution of our samples, then the distance between the means in $\mathbb{R}^n$ is $\mu(\nu_{\mathrm{samp}}, \nu_{\mathrm{test}}) = \left\| \mathbb{E}_{\lambda \sim \nu_{\mathrm{samp}}}[\lambda] - \mathbb{E}_{\lambda \sim \nu_{\mathrm{test}}}[\lambda] \right\|_2$. For statistical feasibility, instead of calculating the full Wasserstein distance we use the averaged marginal Wasserstein distance given by $\mathcal{W}_{\mathrm{marg}}(\nu_{\mathrm{samp}}, \nu_{\mathrm{test}}) = \frac{1}{n} \sum_{k=1}^{n} \mathcal{W}((\nu_{\mathrm{samp}})_k, (\nu_{\mathrm{test}})_k)$, for $(\nu_{\mathrm{samp}})_k$ the marginal distribution in dimension $k$ and $\mathcal{W}$ the Wasserstein distance between two one-dimensional distributions. Using these metrics we can evaluate both marginal and high-dimensional effects.

These metrics also reveal the limitations of GNN- and Graph-Transformer-based models on the simple WL-Bimodal dataset as described in Figure 2, we also see that this extends to the real-world benchmark dataset Community Small. DyDM, on the other hand, consistently overcomes these issues. Even if we compare to the model with ad hoc feature augmentation (Digress with "trick", on which we elaborate in Section 5), DyDM – which does not employ feature augmentation – either

---

[6]See https://anonymous.4open.science/r/DyDM-C854/

improves on or is on par with the feature augmented DiGress. We demonstrate that this performance still holds when working on large datasets, such as the $15'000$ ego-graphs from the Brain dataset.

# 5 RELATED WORK

The spectrum carries key features of graphs (Brouwer & Haemers, 2012), so that spectral methods have generally proven fruitful in graph research, exploiting information encoded in the spectrum, e.g. for graph comparison (Wilson & Zhu, 2008), or in dominant eigenvectors, such as in clustering, community detection (Shi & Malik, 2000; Newman, 2013), and network embedding (Belkin & Niyogi, 2003). For graph diffusion models, however, the spectrum has been used only as either auxiliary features for data augmentation (Vignac et al., 2022), or in a way which makes any remaining $\Theta(n^2)$ degrees of freedom inaccessible, as we outline below.

**Data augmentation.** Vignac et al. (2022) acknowledge the importance of spectra of graphs and add plenty of auxiliary features, among them the first 6 eigenvalues of the Graph Laplacian as graph-level features and the first 2 non-zero eigenvectors as vertex-level features. This engineering trick is indeed helpful, yet it is a trick; as we show in Table 1 these auxiliary features are *necessary* for DiGress to provide a good model. This is in stark contrast to DyDM, where we build on an *analytical* expression of the evolution of *all* eigenvalues during diffusion of a symmetric matrix.

**Spectral methods.** Spectral methods for graph learning have been employed in a variety of contexts, for instance with GANs (Martinkus et al., 2022), which have been, however, outperformed by diffusion-based models (Vignac et al., 2022). Spectral information for graph *diffusion* models has been explored in recent work, however, none consider how a diffusion of the spectrum impacts the remaining information (eigenvectors), leading to either a model that cannot guarantee orthogonality of the eigenvectors (Minello et al., 2025), or a model *solely on the spectrum* as *if the eigenvectors were not impacted by diffusion* (Luo et al., 2024), hence making any recovery of remaining information in the form of the eigenvectors impossible. Luo et al. (2024) argue that the spectrum contains much information, so that upon sampling the spectrum from the OU-based model, the eigenvectors are simply sampled by taking the eigenvectors of a training sample chosen uniformly at random. However, this strategy is very limited, in that it allows only for a diffusion in $n$ parameters, losing $\Theta(n^2)$ degrees of freedom *irrecoverably*. Since we learn in DyDM the spectrum of an OU diffusion *on the entire graph*, the remaining information remains accessible, see Theorem 3.2. This allows learning the eigenvectors beyond sampling uniformly from training data (Luo et al., 2024).

# 6 EXTENSIONS

A key property of DyDM is that it learns the *spectrum* while retaining all remaining $\Theta(n^2)$ degrees of freedom accessible through Eigenvector-SDE. Since Dyson-BM decouples from Eigenvector-SDE, future work could implement a model of Eigenvector-SDE and generate entire graphs. To parameterize Dyson-BM, a graph-transformer model such as Jo et al. (2022) could be used; since Eigenvector-SDE is conditioned on the eigenvalue path which we learn without WL-blindness, we assume that this biases paths sufficiently far from each other to mitigate any WL-problems in classical graph-transformer based methods. Moreover, beyond learning the spectrum of the adjacency matrix $\lambda(A(G))$, our method supports learning the spectrum of the combinatorial graph Laplacian $\lambda(L(G))$ (see Appendix O). Finally, we considered graphs with real (or integer) valued weights, while Dyson-BM is well-defined in other algebras. Hence, generalizations to complex-weighted graphs (Tian & Lambiotte, 2024; Amado et al., 2025) are straightforward: Instead of working on $\mathrm{Sym}(\mathbb{R}^{n \times n})$, one works on Hermitian ensembles.

**Towards a Diffusion Model for Eigenvectors.** We now describe how to implement a diffusion model for the (Eigenvector-SDE), which requires novel methods that go beyond previous work (De Bortoli et al., 2022; Bertolini et al., 2025).

The key insight to implement an efficient diffusion model for Eigenvector-SDE is to rewrite the latter SDE as a Stratonovich SDE on the Lie group $O(n)$ (see Appendix S for definitions). To introduce notation, denote by $\mathfrak{o}(n) = \mathrm{Lie}(O(n)) = T_{\mathrm{Id}}O(n) = \{A \in \mathbb{R}^{n \times n} : A^T = -A\}$ the Lie algebra of $O(n)$, which is the tangent space of $O(n)$ at the identity matrix Id. Let $E_{(\ell,k)} \in \mathfrak{o}(n)$ for $1 \le \ell < k \le d$ be the matrix that is 1 at the $(\ell, k)$-entry and $-1$ at the $(k, \ell)$-entry and 0 otherwise.

Denoting the Eigenvector-SDE by $X(t) = (v_1(t), \dots, v_n(t)) \in \mathrm{O}(n)$, we prove in Proposition S.1

$$dX(t) = \sqrt{\alpha} \sum_{1 \le \ell < k \le n} \left( \frac{X(t) E_{(\ell,k)}}{\lambda_k(t) - \lambda_\ell(t)} \right) \circ W^{(\ell,k)}(t), \qquad (7)$$

viewed as a Stratonovich SDE on $\mathrm{O}(n)$. It is important to note that the latter equation means that for small time steps $h > 0$, $X(t+h)$ can be approximated as

$$X(t+h) \approx X(t) \exp_{\mathrm{O}(n)}(Z) \qquad \text{with} \qquad Z = \sqrt{\alpha h} \sum_{\ell < k} \frac{E_{(\ell,k)} \mathcal{N}^{(\ell,k)}(0,1)}{\lambda_k(t) - \lambda_\ell(t)} \in \mathfrak{o}(n),$$

where $\mathcal{N}^{(\ell,k)}(0,1)$ are independent samples of standard 1-dimensional Gaussians and $\exp_{\mathrm{O}(n)}(Z) = \sum_{i=0}^{\infty} \frac{A^i}{i!}$ is the matrix exponential. We use the latter numerical approximation scheme to generate sample paths for the given SDE.

While SDEs of the form (7) on $\mathrm{O}(n)$ have not been studied for diffusion models so far, we observe that the latter SDE is, as explained in (S.3), similar to Brownian motion on the Lie group $\mathrm{O}(n)$ as studied by De Bortoli et al. (2022). Indeed the Brownian motion $B^{\mathrm{O}(n)}(t)$ on $\mathrm{O}(n)$ written as a Stratonovich SDE is of the form $dB^{\mathrm{O}(n)}(t) = \sum_{\ell < k}(B^{\mathrm{O}(n)}(t) E_{(\ell,k)}) \circ W^{(\ell,k)}(t)$. This similarity with Brownian motion on $\mathrm{O}(n)$ allows us to deduce a time reversal formula analogously to De Bortoli et al. (2022). Indeed, denote by $\mathcal{X}(\mathrm{O}(n))$ the vector fields on $\mathrm{O}(n)$ and by $\nabla \log p_t \in \mathcal{X}(\mathrm{O}(n))$ the gradient vector field of $\log p_t$. We then need to transform the gradient vector field by the map $Q(s) : \mathcal{X}(\mathrm{O}(n)) \to \mathcal{X}(\mathrm{O}(n))$ given for $V \in \mathcal{X}(\mathrm{O}(n))$ and $X \in \mathcal{O}(n)$ as

$$(Q(s)V)(X) = \sum_{\ell < k} \frac{\langle V(X), X E_{(\ell,k)} \rangle}{(\lambda_k(t) - \lambda_\ell(t))^2} X E_{(\ell,k)}.$$

The resulting time reversal formula is then for $Y(s) = X(T - s)$ given by

$$dY(s) = \alpha(Q(T-s)\nabla \log p_{T-s})(Y(s))dt + \sqrt{\alpha} \sum_{\ell < k} \left( \frac{Y(s) E_{(\ell,k)}}{\lambda_k(T-s) - \lambda_\ell(T-s)} \right) \circ W^{(\ell,k)}(s). \quad (8)$$

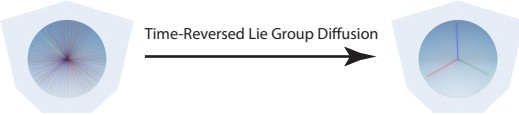

Figure 4: Numerical demonstration on $SO(3)$: Starting from the invariant distribution (left), a learned distribution is being generated (right) using Equation (8).

**Beyond graphs.** Since the Dyson SDE is defined on the domain of $\mathrm{Sym}(\mathbb{R}^{n \times n})$ (and even Hermitian matrices), it could be applied beyond graphs to other data. For instance, if correlations between $n$ points is measured in form of covariance matrices, DyDM could learn the spectrum and thereby quantify how strong the data clusters into (low-dimensional) principal components (Chen et al., 2015; Estavoyer & François, 2022; Hess, 2000).

## 7 CONCLUSION

Leveraging the analytical insights offered by Dyson's Brownian Motion, we introduced DyDM, a diffusion model for spectral learning. Using techniques from Random Matrix Theory, we derived an evolution equation for the corresponding eigenvectors, which renders the remaining information about the underlying matrix available. On the domain of graphs, we demonstrated the struggle of existing learning architectures, e.g., GNN- and graph-transformer-based models. Building on the analytical insights, we decompose the dynamics, such that the spectral part is not constrained by inductive bias, thereby expanding the scope of suitable learning architectures. This way, DyDM can learn the spectrum (even of challenging graph families) without constraining to permutation-equivariant networks. This eliminates the hallucination, so that DyDM learns the distributions without struggle and without any need for data augmentation, as demonstrated experimentally. We hope that this approach opens a new direction in enforcing the inductive bias beyond the learning architecture in graph diffusion models.

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

# APPENDIX

The Appendix is structured as follows. We first derive the tractable loss $\tilde{L}(\theta)$ in Appendix A. We then provide proofs of the Dyson-BM SDE in Appendix B as well as for the Eigenvector-SDE in Appendix C. We then explain the applicability of Anderson's time reversal (Appendix D), and provide a time-rescaling of Dyson-BM in Appendix E. In Appendix F, we provide the adaptive step size controller, followed by an explanation of the shooting mechanism (Appendix G). We provide a theoretical analysis of the complexity of the numerical update steps for Dyson-BM and Eigenvector-SDE in dependence of the graph size $n$ in Appendix H, followed by a discussion of empirical resource considerations in Appendix I. We then explain the sampling procedure (Appendix J) and give engineering details (Appendix K). We explain the challenges of Dyson's Brownian Motion which we overcame with DyDM in Appendix L. We present our theoretical argument for using inductive bias in Appendix M and prove the Lemma on WL equivalence of $k$-regular graphs in Appendix N. In Appendix O, we demonstrate DyDM on the Graph Laplacian, showing its applicability beyond adjacency spectra and show how it can learn properties such as the algebraic connectivity (Fiedler value). For the experiments, we first elaborate on the datasets (Appendix P) followed by an explanation of our extensive benchmarking in Appendix Q, including a comment about undersampling in some benchmark datasets in Appendix Q.1. We show the learning dynamics of 4 different runs of EDP-GNN in Appendix R. We give mathematical details for the extension to Eigenvector-SDE and its time-reversal in Appendix S.

## A  MAKING THE LOSS TRACTABLE

In this section, we deduce a general loss formula for an SDE on $\mathbb{R}$ that will be applied to (Dyson-BM). We consider the process $X = (X(t))_{t \geq 0}$ determined by the SDE

$$\mathrm{d}X(t) = a(t, X(t))\mathrm{d}t + b(t, X(t))\mathrm{d}W_t$$

with initial condition $X(0)$. We assume throughout this section that $X(t)$ is absolutely continuous for all $t \geq 0$ and denote by $p_t$ the density of $X(t)$. We furthermore assume that the joint densities of $X(t)$ and $X(s)$ also have density for all $t, s \geq 0$ that we write as $p_{t,s}$. Observe that by Bayes formula for $t \geq s \geq 0$ we have for $x, y \in \mathbb{R}^d$ that

$$p_{t,s}(y, x) = p_{t|s}(y|x)p_s(x), \tag{9}$$

where $p_{t|s}(y|x)$ is the conditional density of $y$ given $x$.

The canonical loss arising from $s(y, t) = \boldsymbol{\nabla}_{M(t)} \log p_t(M)$ is

$$L'(\theta) = \mathbb{E}_{t\sim\mathrm{Unif}[0,T],M(0)\sim p_0,M(t)\sim p_{t|0}(\cdot|M(0))} \left[ \left\| s_\theta(M(t),t) - \boldsymbol{\nabla}_{M(t)} \log p_t(M(t)) \right\|_2^2 \right]. \tag{10}$$

Note that the difference between $L'(\theta)$ and $L(\theta)$ is that we use the gradient of the *conditional* density $\boldsymbol{\nabla}_{M(t)} \log p_{t|0}(M(t)|M(0))$ in $L(\theta)$. It is a well-known fact that $L'(\theta) = L(\theta) + \mathrm{const}(\theta)$. We will first explain how the loss $L'(\theta)$ from (10) and $\tilde{L}(\theta)$ from (6) are the same up to a constant, that is $L'(\theta) = \tilde{L}(\theta) + \mathrm{const}(\theta)$, which therefore results in the same gradient descent as with $L(\theta)$.

We generalize the loss from (10) to weighing the time by a function $\eta$. So we consider the following generalized loss

$$L'(\theta) := \frac{1}{T} \int_0^T \eta(t) \int_{\mathbb{R}^n} \int_{\mathbb{R}^n} \|s_\theta(y,t) - \boldsymbol{\nabla}_y \log(p_t(y))\|_2^2 \, p_{t,0}(y,x) \, \mathrm{d}y\mathrm{d}x\mathrm{d}t$$

$$= \frac{1}{T} \int_0^T \eta(t) \int_{\mathbb{R}^n} \int_{\mathbb{R}^n} \|s_\theta(y,t) - \boldsymbol{\nabla}_y \log(p_t(y))\|_2^2 \, p_{t|0}(y \mid x)\mathrm{d}yp_0(x)\mathrm{d}x\mathrm{d}t, \tag{11}$$

where $\eta(t)$ is some weighting function with $\int_0^T \eta(t) = 1$. We wrote it in the above form with $p_{t|0}$ denoting the conditional density, since the three integrals can be replaced by $\mathbb{E}_{t\sim U[0,T]} \left[ \cdots \mathbb{E}_{x\sim p_0} \left[ \mathbb{E}_{y\sim p_{t|0}(\cdot|x)} \left[ \|\cdots\|_2^2 \right] \right] \right]$, which we can sample from if we assume (1) sample access to $p_0$, (2) known density at any $t$ given a dirac-delta $p_0$, (3) the term in the norm is tractable. (1) is assumed by the problem definition, (2) is known for an Ornstein-Uhlenbeck forward

SDE, (3) can be solved by realizing that there is an equivalent loss function $\hat{L}(\theta) = L(\theta) + \text{const}(\theta)$ where $p_t$ inside the norm is converted to a $p_{t|0}$, which is known (gaussian density) in an OU setting.

Here, steps (2) and (3) fail. Note that by the polarization identity

$$\|s_\theta(y,t) - \boldsymbol{\nabla}_y \log(p_t(y))\|_2^2 = \|s_\theta(y,t)\|_2^2 + \|\boldsymbol{\nabla}_y \log(p_t(y))\|_2^2 - 2 s_\theta(y,t)^T \boldsymbol{\nabla}_y \log(p_t(y))$$

We will rewrite the mixed term $s_\theta(y,t)^T \boldsymbol{\nabla}_y \log(p_t(y))$ of the $L^2$ norm in such a way that the first quadratic term $\|s_\theta(y,t)\|_2^2$ remains unchanged and the second quadratic term $\|\boldsymbol{\nabla}_y \log(p_t(y))\|_2^2$ is constant in $\theta$. First, we rewrite

$$\frac{1}{T} \int_0^T \eta(t) \int_{\mathbb{R}^n \times \mathbb{R}^n} s_\theta(y,t)^T \boldsymbol{\nabla}_y \log(p_t(y)) p_{t,0}(y,x) \, \mathrm{d}x \, \mathrm{d}y \, \mathrm{d}t \tag{12}$$

$$= \frac{1}{T} \int_0^T \eta(t) \int_{\mathbb{R}^n} s_\theta(y,t)^T \boldsymbol{\nabla}_y \log(p_t(y)) p_t(y) \, \mathrm{d}y \, \mathrm{d}t$$

$$= \frac{1}{T} \int_0^T \eta(t) \int_{\mathbb{R}^n} s_\theta(y,t)^T \boldsymbol{\nabla}_y \left[ p_t(y) \right] \mathrm{d}y \, \mathrm{d}t. \tag{13}$$

We next observe that for $0 \le s' \le s$ we have the following:

$$\boldsymbol{\nabla}_y \left[ p_s(y) \right] = \boldsymbol{\nabla}_y \left[ \int_{\mathbb{R}^n} p_{s|s'}(y|z) p_{s'}(z) \, \mathrm{d}z \right]$$

$$= \int_{\mathbb{R}^n} \boldsymbol{\nabla}_y p_{s|s'}(y|z) p_{s'}(z) \, \mathrm{d}z$$

$$= \int_{\mathbb{R}^n} \boldsymbol{\nabla}_y p_{s|s'}(y|z) \frac{p_{s,s'}(y,z)}{p_{s|s'}(y|z)} \, \mathrm{d}z$$

$$= \int_{\mathbb{R}^n} \boldsymbol{\nabla}_y \log p_{s|s'}(y|z) p_{s,s'}(y,z) \, \mathrm{d}z.$$

So we now perform in (13) for a small $h > 0$ a change of variables to $t + h$ and apply the latter equality with $s = t + h$ and $s' = t$. Then up to ignoring the boundary at $0$ and $T$, and using by (9) that $p_{t+h,t}(y,z) = p_{t+h|t}(y|z) p_t(z)$ it follows that the loss from (6)

$$(13) = \frac{1}{T} \int_0^T \eta(t+h) \int_{\mathbb{R}^n} \int_{\mathbb{R}^n} s_\theta(y,t+h)^T \boldsymbol{\nabla}_y \log p_{t+h|t}(y|z) p_{t+h|t}(y|z) \, \mathrm{d}y \, p_t(z) \, \mathrm{d}z \, \mathrm{d}t. \tag{14}$$

So it follows that

$$\tilde{L}(\theta) = \frac{1}{T} \int_0^T \eta(t+h) \int_{\mathbb{R}^n} \int_{\mathbb{R}^n} \left\| s_\theta(y,t+h) - \boldsymbol{\nabla}_y \log\left( p_{t+h|t}(y|z) \right) \right\|_2^2 p_{t+h|t}(y|z) \, \mathrm{d}y \, p_t(z) \, \mathrm{d}z \, \mathrm{d}t \tag{15}$$

is equal to $L(\theta)$ up to a constant term in $\theta$ (ignoring the boundary terms at $0$ and $T$).

We will now make a series of approximations to calculate the loss $\tilde{L}(\theta)$. The first one is to approximate the integral $\int_0^T$ over $t$ by a sum $\sum_{i=1}^k$ over the time points $t_0 < t_1 < \ldots < t_k$ with $t_0 = 0$ and $t_k = T$. The second approximation we make is that we approximate the latter integral $\int_{\mathbb{R}^n} \int_{\mathbb{R}^n}$ by sampling a path from $p_t$ at the time steps $t_i$. Indeed, we denote for each integer $1 \le r \le N$ by $x_{t_i}^{(r)}$ the sample path of $p_t$. Moreover, we actually make the time grid also dependent on our sample path. So for each $1 \le r \le N$, let $t_0^{(r)} < t_1^{(r)} < \ldots < t_{k^{(r)}}^{(r)}$ with $t_0^{(r)} = 0$ and $t_{k^{(r)}}^{(r)} = T$ be the discretization of $[0,T]$. Thus, the overall loss can be approximated as

$$\tilde{L}(\theta) \approx \frac{1}{N} \sum_{r=1}^N \sum_{i=1}^{k^{(r)}} \frac{t_i^{(r)} - t_{i-1}^{(r)}}{T} \eta(t_i^{(r)}) \left\| s_\theta(x_i^{(r)}, t_i^{(r)}) - \boldsymbol{\nabla}_{x_i^{(r)}} \left[ \log\left( p_{t_i^{(r)}|t_{i-1}^{(r)}}(x_i^{(r)} \mid x_{i-1}^{(r)}) \right) \right] \right\|_2^2$$

$$+ \text{const}(\theta). \tag{16}$$

We finally approximate the incremental score function $\boldsymbol{\nabla}_{x_i^{(r)}} \left[ \log\left( p_{t_i^{(r)}|t_{i-1}^{(r)}}(x_i^{(r)} \mid x_{i-1}^{(r)}) \right) \right]$ as follows. If $h > 0$ is a small time step, we can approximate the conditional random variable $X_{t+h} \mid \mathcal{F}_t$

with $x := X_t$ by

$$X_{t+h} \mid \mathcal{F}_t \approx x + a(t,x)h + b(t,x)N(0,h)$$
$$\text{(where } N(0,h) \text{ is a centered Gaussian RV with variance } h)$$
$$\sim \mathcal{N}\left(x + a(t,x)h,\, b^2(t,x)h\right), \tag{17}$$

so that we have for the density

$$p_{t+h|t}(y|x) \approx \frac{1}{\sqrt{2\pi b^2(t,x)h}} \exp\left(-\frac{(y - x - a(t,x)h)^2}{2b^2(t,x)h}\right), \tag{18}$$

which means for the scores

$$\boldsymbol{\nabla}_y \log\left(p_{t+h|t}(y|x)\right) \approx -\frac{y - x - a(t,x)h}{b^2(t,x)h}. \tag{19}$$

Thus, combining (16) and (19), the loss $\tilde{L}(\theta)$ can be approximated by the following as we use in our model:

$$\frac{1}{N} \sum_{r=1}^{N} \sum_{i=1}^{k^{(r)}} \frac{t_i^{(r)} - t_{i-1}^{(r)}}{T} \eta(t_i^{(r)}) \left\| s_\theta(x_i^{(r)}, t_i^{(r)}) - \frac{a(t_{i-1}^{(r)}, x_{i-1}^{(r)}) \cdot (t_i^{(r)} - t_{i-1}^{(r)}) - (x_i^{(r)} - x_{i-1}^{(r)})}{b^2(t_{i-1}^{(r)}, x_{i-1}^{(r)}) \cdot (t_i^{(r)} - t_{i-1}^{(r)})} \right\|_2^2 \tag{20}$$

## B   SPECTRAL DYSON SDE

**Theorem 3.1 (restated).** *Denote by $\lambda(t) = (\lambda_1(t), \ldots, \lambda_n(t))$ the ordered spectrum of the matrix-valued Ornstein-Uhlenbeck process $M(t)$ of eq. (1). Then assuming that the initial matrix $M(0)$ has simple spectrum, $\lambda(t)$ satisfies for all $1 \le k \le n$ the stochastic differential equation*

$$\mathrm{d}\lambda_k(t) = \left(\alpha \sum_{\ell \neq k} \frac{1}{\lambda_k(t) - \lambda_\ell(t)} - \beta\lambda_k(t)\right) \mathrm{d}t + \sqrt{2\alpha}\,\mathrm{d}W_k(t), \tag{Dyson-BM}$$

*for $W_1, \ldots, W_n$ independent standard Brownian motions. Moreover the unique stationary distribution of (Dyson-BM) has density*

$$p_{\mathrm{inv}}(\lambda) = \frac{1}{Z} \exp(-U(\lambda)) \qquad \text{for} \qquad U(\lambda) = \frac{\beta}{2\alpha} \sum_k \lambda_k^2 - \sum_{k<\ell} \ln|\lambda_k - \lambda_\ell|,$$

*for $\lambda \in C_n$ and $Z$ a normalizing constant so that $p_{\mathrm{inv}}$ corresponds to a probability measure.*

*Proof.* We prove the theorem in two parts.

**Dyson SDE.**   We first prove Dyson-BM. This part of the proof is based on Keating (2023) but generalizes it to arbitrary coefficients. For mathematical details on the $\alpha = \frac{1}{n}$, $\beta = 0$ case, see also Anderson et al. (2009). Suppose $M(t)$ satisfies the SDE eq. (1). That is, with $M(0) \in \mathrm{Sym}(\mathbb{R}^{n \times n})$ we have

$$\mathrm{d}M_{ij}(t) = -\beta M_{ij}\mathrm{d}t + D_{ij}\mathrm{d}B_{ij}$$

with $D_{ij} := \sqrt{(1 + \delta_{ij})\alpha}$ for any constants $\alpha, \beta \in \mathbb{R}^+$ with $B_{ij}(t) = B_{ji}(t) \; \forall t > 0$. Let $\lambda_1 \ge \ldots \ge \lambda_n$ be the eigenvalues. We choose as $v_1, \ldots, v_n$ an orthonormal basis of eigenvectors $(Mv_k = \lambda_k v_k)$, which exists by the spectral theorem for symmetric matrices.

Due to symmetry, we may constrain the set of indices $(i,j)$ to $\mathcal{I} := \{(i,j) \mid 1 \le i \le j \le n\}$. For any $k \in [n]$, the eigenvalue $\lambda_k$ can thus be seen as a function of the set of Itô processes $\{M_\eta \mid \eta \in \mathcal{I}\}$. Hence, we have by Itô's Lemma,

$$\mathrm{d}\lambda_k = \underbrace{\frac{\partial \lambda_k}{\partial t}\mathrm{d}t}_{\equiv 0} + \sum_{\eta \in \mathcal{I}} \frac{\partial \lambda_k}{\partial M_\eta}\mathrm{d}M_\eta + \frac{1}{2} \sum_{\eta, \xi \in \mathcal{I}} \frac{(\partial \lambda_k)^2}{\partial M_\eta \partial M_\xi}\mathrm{d}M_\eta \mathrm{d}M_\xi, \tag{21}$$

where the first part is 0 since $\lambda_k(t)$ is only a function of the $M_\eta(t)$, not of time. By eq. (1), we get

$$= \sum_{\eta \in \mathcal{I}} \left( -\beta M_\eta \frac{\partial \lambda_k}{\partial M_\eta} + \frac{1}{2} D_\eta^2 \frac{\partial^2 \lambda_k}{(\partial M_\eta)^2} \right) \mathrm{d}t + D_\eta \frac{\partial \lambda_k}{\partial M_\eta} \mathrm{d}B_\eta. \tag{22}$$

It remains to calculate the partial derivatives. In what follows, we successively apply properties of the spectrum in order to obtain equations for the partial derivatives. We have

$$M v_k = \lambda_k v_k \tag{23}$$

taking the partial derivative with respect to $M_{ij}$ for $(i, j) \in \mathcal{I}$ on both sides and applying the product rule yields

$$\frac{\partial M}{\partial M_{ij}} v_k + M \frac{\partial v_k}{\partial M_{ij}} = \frac{\partial \lambda_k}{\partial M_{ij}} v_k + \lambda_k \frac{\partial v_k}{\partial M_{ij}}. \tag{24}$$

Further, by orthogonality of the eigenvectors we have for any $k, l \in [n]$

$$v_k^T v_l = \delta_{kl} \tag{25}$$

$$\frac{\partial v_k^T}{\partial M_{ij}} v_l + v_k^T \frac{\partial v_l}{\partial M_{ij}} = 0. \tag{26}$$

which, taking the transpose, implies for $l = k$

$$\frac{\partial v_k^T}{\partial M_{ij}} v_k = v_k^T \frac{\partial v_k}{\partial M_{ij}} = 0. \tag{27}$$

Multiplying eq. (24) by $v_l^T$ from the left yields

$$v_l^T \frac{\partial M}{\partial M_{ij}} v_k + \underbrace{v_l^T M \frac{\partial v_k}{\partial M_{ij}}}_{(I)} = \underbrace{\frac{\partial \lambda_k}{\partial M_{ij}} v_l^T v_k}_{(II)} + \underbrace{\lambda_k v_l^T \frac{\partial v_k}{\partial M_{ij}}}_{(III)} \tag{28}$$

where the terms simplify as follows.

$$(I) = (1 - \delta_{kl}) \lambda_l v_l^T \frac{\partial v_k}{\partial M_{ij}} \qquad \text{(by eq. (27))}$$

$$(II) = \delta_{lk} \frac{\partial \lambda_k}{\partial M_{ij}} \tag{29}$$

$$(III) = (1 - \delta_{kl}) \lambda_k v_l^T \frac{\partial v_k}{\partial M_{ij}}, \qquad \text{(by eq. (27))}$$

resulting in the following equation for $l = k$

$$v_k^T \frac{\partial M}{\partial M_{ij}} v_k = \frac{\partial \lambda_k}{\partial M_{ij}} \tag{30}$$

and for $l \neq k$:

$$v_l^T \frac{\partial M}{\partial M_{ij}} v_k + \lambda_l v_l^T \frac{\partial v_k}{\partial M_{ij}} = \lambda_k v_l^T \frac{\partial v_k}{\partial M_{ij}} \tag{31}$$

$$v_l^T \frac{\partial M}{\partial M_{ij}} v_k = (\lambda_k - \lambda_l) v_l^T \frac{\partial v_k}{\partial M_{ij}}. \tag{32}$$

The matrix derivative is, trivially,

$$\left( \frac{\partial M}{\partial M_{ij}} \right)_{kl} = \begin{cases} 1 & \text{for } (k, l) = (i, j) \text{ or } (l, k) = (i, j) \\ 0 & \text{else} \end{cases}, \tag{33}$$

so that we can simplify eq. (30) to

$$\frac{\partial \lambda_k}{\partial M_{ij}} = (v_k)_i (v_k)_j (2 - \delta_{ij}) \tag{34}$$

and eq. (32) to

$$(\lambda_k - \lambda_l) v_l^T \frac{\partial v_k}{\partial M_{ij}} = (v_k)_i (v_l)_j + (1 - \delta_{ij})(v_k)_j (v_l)_i \tag{35}$$

$$= (v_l)_i (v_k)_j + (1 - \delta_{ij})(v_l)_j (v_k)_i. \tag{36}$$

Using that the $v_k$ are an ONB of eigenvectors, we can conclude with eq. (34) that the first summand of eq. (22) is

$$\sum_{\eta \in \mathcal{I}} -\beta M_\eta \frac{\partial \lambda_k}{\partial M_\eta} = -\beta \lambda_k. \tag{37}$$

Since $\{v_1, \ldots, v_n\}$ is an orthonormal basis, we may project on the basis vectors as follows

$$\frac{\partial v_k}{\partial M_{ij}} = \sum_{l \in [n]} v_l^T \frac{\partial v_k}{\partial M_{ij}} v_l \qquad \text{(projection into ONB basis)}$$

applying eq. (27) gives

$$= \sum_{l \in [n] \setminus \{k\}} v_l^T \frac{\partial v_k}{\partial M_{ij}} v_l \tag{38}$$

which yields with eq. (35)

$$= \sum_{l \in [n] \setminus \{k\}} \frac{1}{\lambda_k - \lambda_l} \left( (v_k)_i (v_l)_j + (1 - \delta_{ij})(v_k)_j (v_l)_i \right) v_l. \tag{39}$$

To obtain the second order partial derivative of $\lambda_k$, we observe

$$\frac{\partial^2 \lambda_k}{\partial M_{ij} \partial M_{ij}} = \frac{\partial}{\partial M_{ij}} \left( (v_k)_i (v_k)_j (2 - \delta_{ij}) \right) \qquad \text{(by eq. (34))}$$

$$= (2 - \delta_{ij}) \left( \frac{\partial (v_k)_i}{\partial M_{ij}} (v_k)_j + (v_k)_i \frac{\partial (v_k)_j}{\partial M_{ij}} \right)$$

$$= (2 - \delta_{ij}) \sum_{l \in [n] \setminus \{k\}} \frac{1}{\lambda_k - \lambda_l} \left( (v_l)_i (v_k)_j (v_l)_i (v_k)_j + (v_l)_j (v_k)_i (1 - \delta_{ij})(v_l)_i (v_k)_j \right.$$

$$\left. + (v_l)_i (v_k)_j (v_l)_j (v_k)_i + (v_l)_j (v_k)_i (1 - \delta_{ij})(v_l)_j (v_k)_i \right) \qquad \text{(by eq. (39))}$$

$$= (2 - \delta_{ij}) \sum_{l \in [n] \setminus \{k\}} \frac{1}{\lambda_k - \lambda_l} \left( (v_l)_i^2 (v_k)_j^2 + (v_l)_j^2 (v_k)_i^2 (1 - \delta_{ij}) \right.$$

$$\left. + (v_l)_i (v_l)_j (v_k)_i (v_k)_j (2 - \delta_{ij}) \right). \tag{40}$$

These second order partial derivative are summed over $\mathcal{I}$ in eq. (22), so that combining eq. (40) with the definition of $D_{ij}$ yields

$$\sum_{(i,j) \in \mathcal{I}} D_{ij}^2 \frac{\partial^2 \lambda_k}{(\partial M_{ij})^2} = \sum_{j=1}^n \sum_{i=1}^j \alpha (1 + \delta_{ij})(2 - \delta_{ij}) \sum_{l \in [n] \setminus \{k\}} \frac{1}{\lambda_k - \lambda_l} \left( (v_l)_i^2 (v_k)_j^2 \right.$$

$$\left. + (v_l)_j^2 (v_k)_i^2 (1 - \delta_{ij}) + (v_l)_i (v_l)_j (v_k)_i (v_k)_j (2 - \delta_{ij}) \right) \tag{41}$$

we reorder and note that since the summand is symmetric in $i, j$, we may change the range of summation and absorb the coefficient $2 - \delta_{ij}$,

$$= \alpha \sum_{l \in [n] \setminus \{k\}} \frac{1}{\lambda_k - \lambda_l} \sum_{j=1}^{n} \sum_{i=1}^{n} (1 + \delta_{ij}) \Big( (v_l)_i^2 (v_k)_j^2$$

$$+ (v_l)_j^2 (v_k)_i^2 (1 - \delta_{ij}) + (v_l)_i (v_l)_j (v_k)_i (v_k)_j (2 - \delta_{ij}) \Big) \tag{42}$$

where we realize that upon accounting for all $\delta_{ij}$, the first two summands are simply $\|v_l\|_2^2 \|v_k\|_2^2$, while the third summand may we written an inner product

$$= \alpha \sum_{l \in [n] \setminus \{k\}} \frac{1}{\lambda_k - \lambda_l} \left( 2\|v_l\|_2^2 \|v_k\|_2^2 + 2\underbrace{(v_l^T v_k)^2}_{0} \right) \tag{43}$$

since we chose an ONB, this simplifies to

$$= 2\alpha \sum_{l \in [n] \setminus \{k\}} \frac{1}{\lambda_k - \lambda_l}. \tag{44}$$

It remains to determine an explicit expression of the Brownian motion in eq. (22). We have

$$\sum_{\eta \in \mathcal{I}} D_\eta \frac{\partial \lambda_k}{\partial M_\eta} \mathrm{d}B_\eta = \sum_{(i,j) \in \mathcal{I}} \sqrt{(1 + \delta_{ij}) \alpha (2 - \delta_{ij})} (v_k)_i (v_k)_j \mathrm{d}B_{ij}$$

using that $B_{ij}(t) = B_{ji}(t)$, we obtain

$$= \sqrt{2\alpha} \sum_{i=1}^{n} \sum_{j=1}^{n} \sqrt{\frac{1 + \delta_{ij}}{2}} (v_k)_i (v_k)_j \mathrm{d}B_{ij}$$

we may now define $\mathrm{d}\tilde{B}_k$ as follows

$$= \sqrt{2\alpha} \mathrm{d}\tilde{B}_k. \tag{45}$$

Indeed, the set $\{\tilde{B}_k \mid k \in [n]\}$ is in distribution equal $n$ independent standard Brownian motions, since $\mathbb{E}\left[\mathrm{d}\tilde{B}_k\right] = 0$ and for $k, l \in [n]$

$$\mathbb{E}\left[\mathrm{d}\tilde{B}_k \mathrm{d}\tilde{B}_l\right] = \mathbb{E}\left[ \frac{1}{2} \sum_{ij} \sum_{st} \sqrt{1 + \delta_{ij}} \sqrt{1 + \delta_{st}} (v_k)_i (v_k)_j (v_l)_s (v_l)_t \mathrm{d}B_{ij} \mathrm{d}B_{st} \right]$$

where we realize that the product of the differentials is 0 except for $(i, j) = (s, t)$ and $(i, j) = (t, s)$

$$= \mathbb{E}\left[ v_k^T v_l v_k^T v_l \right] \mathrm{d}t$$
$$= \delta_{kl} \mathrm{d}t. \tag{46}$$

We can thus conclude that by eq. (22) we have,

$$\mathrm{d}\lambda_k = \left( -\beta \lambda_k + \alpha \sum_{l \in [n] \setminus \{k\}} \frac{1}{\lambda_k - \lambda_l} \right) \mathrm{d}t + \sqrt{2\alpha} \mathrm{d}W_k \tag{47}$$

where $\{W_k \mid k \in [n]\}$ are $n$ independent Brownian motions.

**Invariant Distribution.**    We now show that

$$p_{\text{inv}}(\lambda) = \frac{1}{Z} \exp(-U(\lambda)) \qquad \text{for} \qquad U(\lambda) = \frac{\beta}{2\alpha} \sum_k \lambda_k^2 - \sum_{k<\ell} \ln |\lambda_k - \lambda_\ell|, \qquad (48)$$

for $\lambda \in C_n$ and $Z$ a normalizing constant so that $p_{\text{inv}}$ is a probability measure, is the invariant distribution. We use the Fokker-Plank equation. Indeed, recall that if we have an SDE

$$\mathrm{d}\lambda_t = f(\lambda_t)\mathrm{d}t + L(\lambda_t)\mathrm{d}B_t$$

in dimension $n$, where $f(\lambda) = (f_1(\lambda), \ldots, f_n(\lambda))$ is a $C^2$-function, $L(\lambda)$ is matrix-valued $C^2$-function and $B_t$ is a $n$-dimensional Brownian motion, then an invariant distribution $p(x)$ satisfies

$$\sum_{i=1}^n \frac{\partial}{\partial \lambda_i}[f_i(\lambda)p(\lambda)] = \frac{1}{2} \sum_{i,j=1}^n \frac{\partial^2}{\partial \lambda_i \partial \lambda_j}[L(\lambda)L(\lambda)^T]_{ij}p(\lambda). \qquad (49)$$

In our case, for the $\lambda(\alpha, \beta)$-SDE we have that $L(\lambda, t) = \sqrt{2\alpha}1_n$ and therefore $[L(\lambda, t)L(\lambda, t)^T]_{ij} = 2\alpha\delta_{ij}$. So the right hand side of (49) equals

$$\alpha \sum_{i=1}^n \frac{\partial^2}{(\partial \lambda_i)^2}p(\lambda).$$

Assume for now that $\lambda = (\lambda_1, \ldots, \lambda_n)$ satisfies $\lambda_1 > \lambda_2 > \ldots > \lambda_n$ and

$$U(\lambda) = c \sum_i \lambda_i^2 - \sum_{i<j} \ln(\lambda_i - \lambda_j)$$

for some constant $c > 0$ to be determined. We first calculate for a fixed $i$,

$$\frac{\partial}{\partial \lambda_i}p(\lambda) = -\frac{1}{Z} \exp(-U(\lambda))\frac{\partial}{\partial \lambda_i}U(\lambda)$$

$$= -\frac{1}{Z} \exp(-U(\lambda))\left(2c\lambda_i - \sum_{i<j} \frac{1}{\lambda_i - \lambda_j} + \sum_{j<i} \frac{1}{\lambda_j - \lambda_i}\right)$$

$$= -\frac{1}{Z} \exp(-U(\lambda))\left(2c\lambda_i - \sum_{j\neq i} \frac{1}{\lambda_i - \lambda_j}\right).$$

Therefore,

$$\frac{\partial^2}{(\partial \lambda_i)^2}p(\lambda) = \frac{1}{Z} \exp(-U(\lambda))\left(2c\lambda_i - \sum_{j\neq i} \frac{1}{\lambda_i - \lambda_j}\right)^2 - \frac{1}{Z} \exp(-U(\lambda))\left(2c + \sum_{j\neq i} \frac{1}{(\lambda_i - \lambda_j)^2}\right)$$

$$= \left(\sum_{j\neq i} \frac{1}{\lambda_i - \lambda_j} - 2c\lambda_i\right)\frac{\partial}{\partial \lambda_i}p(\lambda) - p(\lambda)\left(2c + \sum_{j\neq i} \frac{1}{(\lambda_i - \lambda_j)^2}\right)$$

and so the right hand side of (49) is equal to

$$\alpha \sum_{i=1}^n \frac{\partial^2}{(\partial \lambda_i)^2}p(\lambda) = \sum_{i=1}^n \left(\alpha \sum_{j\neq i} \frac{1}{\lambda_i - \lambda_j} - 2\alpha c\lambda_i\right)\frac{\partial}{\partial \lambda_i}p(\lambda) - \sum_{i=1}^n p(\lambda)\left(2\alpha c + \alpha \sum_{j\neq i} \frac{1}{(\lambda_i - \lambda_j)^2}\right)$$

Now the left hand side of (49) is equal to

$$\sum_{i=1}^n \frac{\partial}{\partial \lambda_i}\left[\left(\alpha \sum_{i\neq j} \frac{1}{\lambda_i - \lambda_j} - \beta\lambda_i\right)p(\lambda)\right]$$

$$= \sum_{i=1}^n \left(\alpha \sum_{i\neq j} \frac{1}{\lambda_i - \lambda_j} - \beta\lambda_i\right)\frac{\partial}{\partial \lambda_i}p(\lambda) + \sum_{i=1}^n \left(-\alpha \sum_{i\neq j} \frac{1}{(\lambda_i - \lambda_j)^2} - \beta\right)p(\lambda)$$

So it follows that in (49) the left-hand side is equal to the right-hand side if and only if $\beta = 2\alpha c$ or equivalently $c = \beta/2\alpha$, concluding the proof. $\qquad\square$

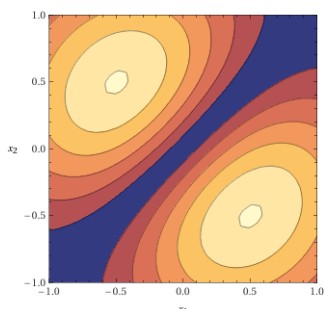

Figure 5: Plot of the invariant density of Dyson-BM for $d = 2$ and $\alpha = \beta = 1$

## C   INFERRING THE EIGENVECTOR DYNAMICS

**Theorem 3.2 (restated).** *Denote by $(v_1(t), \ldots, v_n(t))$ a tuple of orthonormal eigenvectors associated to the eigenvalues of Theorem 3.1. Assuming that the initial matrix $M(0)$ has simple spectrum, $v_k(t)$ satisfies for $k \in [n]$ the stochastic differential equation*

$$\mathrm{d}v_k(t) = -\frac{\alpha}{2} \sum_{\ell \neq k} \frac{1}{(\lambda_k(t) - \lambda_\ell(t))^2} v_k(t)\mathrm{d}t + \sqrt{\alpha} \sum_{\ell \neq k} \frac{1}{\lambda_k(t) - \lambda_\ell(t)} v_\ell(t)\mathrm{d}w_{\ell k}(t)$$

(Eigenvector-SDE)

*for $\{w_{ij:i\neq j}\}$ standard Brownian motions also independent of the eigenvalue trajectories, with $w_{ji} = w_{ij}$.*

*Proof.* Analogously to the proof of Theorem 3.1, we may view the $v_k$ for $k \in [n]$ as a function of the matrix components $M_{ij}$ for $(i,j) \in \mathcal{I} := \{(i,j) : 1 \leq i \leq j \leq n\}$. We thus have by Itô's lemma

$$\mathrm{d}v_k = \sum_{\eta \in \mathcal{I}} \frac{\partial v_k}{\partial M_\eta} \mathrm{d}M_\eta + \frac{1}{2} \sum_{\eta, \xi \in \mathcal{I}} \frac{\partial^2 v_k}{\partial M_\eta \partial M_\xi} \mathrm{d}M_\eta \mathrm{d}M_\xi$$

$$= \sum_{\eta \in \mathcal{I}} \left( -\beta M_\eta \frac{\partial v_k}{\partial M_\eta} + \frac{1}{2} D_\eta^2 \frac{\partial^2 v_k}{(\partial M_\eta)^2} \right) \mathrm{d}t + D_\eta \frac{\partial v_k}{\partial M_\eta} \mathrm{d}B_\eta. \tag{50}$$

For the first summand of eq. (50), we observe, using eq. (38), that

$$\sum_{(i,j)\in\mathcal{I}} M_{ij} \frac{\partial v_k}{\partial M_{ij}} \mathrm{d}t = \sum_{l \neq k} \sum_{(i,j)\in\mathcal{I}} v_l^T \frac{\partial v_k}{\partial M_{ij}} v_l M_{ij} \mathrm{d}t \tag{51}$$

and further use eq. (32)

$$= \sum_{l \neq k} \frac{1}{\lambda_k - \lambda_l} \left( v_l^T \underbrace{\sum_{(i,j)\in\mathcal{I}} \frac{\partial M}{\partial M_{ij}} M_{ij}}_{= M \text{ by eq. (33)}} v_k \right) v_l \mathrm{d}t$$

$$= \sum_{l \neq k} \frac{\lambda_k}{\lambda_k - \lambda_l} (v_l^T v_k) v_l \mathrm{d}t. \tag{52}$$

Since $l \neq k$ and the $\{v_l \mid l \in [n]\}$ are orthogonal, we get

$$= 0. \tag{53}$$

For the second summand of eq. (50), we observe

$$\frac{\partial^2 v_k}{(\partial M_\eta)^2} = \frac{\partial}{\partial M_\eta} \frac{\partial v_k}{\partial M_\eta} \tag{54}$$

and using Equation (32) and Equation (38) we get

$$= \sum_{m \neq k} \frac{\partial}{\partial M_\eta} \left[ \frac{1}{\lambda_k - \lambda_m} v_m^T \frac{\partial M}{\partial M_\eta} v_k v_m \right] \tag{55}$$

by noting $\frac{\partial}{\partial M_\eta} \frac{\partial M}{\partial M_\eta} = 0$, we apply the chain rule to obtain

$$= \sum_{m \neq k} \left\{ \underbrace{- \left( \frac{1}{\lambda_k - \lambda_m} \right)^2 \left( \frac{\partial \lambda_k}{\partial M_\eta} - \frac{\partial \lambda_m}{\partial M_\eta} \right) v_m^T \frac{\partial M}{\partial M_\eta} v_k v_m}_{(i)} + \underbrace{\frac{1}{\lambda_k - \lambda_m} \frac{\partial v_m^T}{\partial M_\eta} \frac{\partial M}{\partial M_\eta} v_k v_m}_{(ii)} \right. \tag{56}$$

$$\left. + \underbrace{\frac{1}{\lambda_k - \lambda_m} v_m^T \frac{\partial M}{\partial M_\eta} \frac{\partial v_k}{\partial M_\eta} v_m}_{(iii)} + \underbrace{\frac{1}{\lambda_k - \lambda_m} v_m^T \frac{\partial M}{\partial M_\eta} v_k \frac{\partial v_m}{\partial M_\eta}}_{(iv)} \right\}. \tag{57}$$

We now analyze each term.

**Term (i)**  We recall Equation (34)

$$\frac{\partial \lambda_k}{\partial M_{ij}} = (v_k)_i (v_k)_j (2 - \delta_{ij}),$$

as well as that the partial derivative $\frac{\partial M}{\partial M_{ij}}$ is 0 except at $i, j$ and $j, i$, where it is 1 (using symmetry).

Pulling the summation over $\eta$ and all $\eta$-dependent terms in, we have

$$\sum_{\eta \in \mathcal{I}} (1 + \delta_\eta) \left( \frac{\partial \lambda_k}{\partial M_\eta} - \frac{\partial \lambda_m}{\partial M_\eta} \right) v_m^T \frac{\partial M}{\partial M_\eta} v_k$$

$$= \sum_{ij \in \mathcal{I}} (1 + \delta_{ij}) \left\{ \left[ (v_k)_i (v_k)_j (2 - \delta_{ij}) - (v_m)_i (v_m)_j (2 - \delta_{ij}) \right] \right.$$

$$\left. \left[ (v_m)_i (v_k)_j + (v_m)_j (v_k)_i (1 - \delta_{ij}) \right] \right\}$$

$$= 2 \left[ v_k v_k^T v_k^T v_m - v_m^T v_m v_m^T v_k \right]$$

$$= 2 (\delta_{km} - \delta_{km})$$

$$= 0.$$

Thus, the overall contribution of term $(i)$ is 0.

**Term (ii)**  For term (ii) we have by Equation (32) and Equation (38)

$$(ii) = \frac{1}{\lambda_k - \lambda_m} \sum_{s \neq m} \frac{1}{\lambda_m - \lambda_s} v_m^T \frac{\partial M}{\partial M_\eta} v_s v_s^T \frac{\partial M}{\partial M_\eta} v_k v_m. \tag{58}$$

Using

$$v_m^T \frac{\partial M}{\partial M_{ij}} v_s = (v_m)_i (v_s)_j + (v_m)_j (v_s)_i (1 - \delta_{ij}), \tag{59}$$

we note that

$$\sum_{\eta \in \mathcal{I}} D_\eta^2 v_m^T \frac{\partial M}{\partial M_\eta} v_s v_s^T \frac{\partial M}{\partial M_\eta} v_k$$

$$= \alpha \sum_{i,j \in \mathcal{I}} (1 + \delta_{ij}) \left[ (v_m)_i (v_s)_j + (v_m)_j (v_s)_i (1 - \delta_{ij}) \right] \left[ (v_k)_i (v_s)_j + (v_k)_j (v_s)_i (1 - \delta_{ij}) \right]$$

$$= \alpha \left\{ \left[ \sum_i (v_m)_i (v_k) i \right] \left[ \sum_i (v_s)_i^2 \right] + v_m^T v_s v_s^T v_k \right\}$$

$$= \alpha \delta_{km} + \alpha \delta_{ms} \delta_{sk} \tag{60}$$

$$= 0.$$

Thus, the contribution of term $(ii)$ vanishes.

**Term (iii)**  Analogous to term $(ii)$.

**Term (iv)**  By using Equation (32) and Equation (38), we get

$$
\begin{aligned}
(iv) =& \frac{1}{\lambda_k - \lambda_m} v_m^T \frac{\partial M}{\partial M_\eta} v_k \frac{\partial v_m}{\partial M_\eta} \\
=& \frac{1}{\lambda_k - \lambda_m} v_m^T \frac{\partial M}{\partial M_\eta} v_k \sum_{p \neq m} \frac{1}{\lambda_m - \lambda_p} v_p^T \frac{\partial M}{\partial M_\eta} v_m v_p \\
=& \frac{1}{\lambda_k - \lambda_m} \sum_{p \neq m} \frac{1}{\lambda_m - \lambda_p} \left( v_m^T \frac{\partial M}{\partial M_\eta} v_k \right) \left( v_p^T \frac{\partial M}{\partial M_\eta} v_m \right) v_p.
\end{aligned}
$$

By pulling in the $\eta$-dependent terms from Equation (50) which affect term $(iv)$, and then by subsequently pulling the sum over $\eta$ in, we get

$$
\sum_{\eta \in \mathcal{I}} \frac{1}{2} D_\eta^2 v_m^T \frac{\partial M}{\partial M_\eta} v_k v_p^T \frac{\partial M}{\partial M_\eta} v_m \tag{61}
$$

which, using eq. (60), becomes

$$
\begin{aligned}
=& \frac{\alpha}{2} \left( \delta_{kp} + \delta_{km} \delta_{mp} \right) \\
=& \frac{\alpha}{2} \delta_{kp}.
\end{aligned}
$$

Moreover, using that $p \neq m$, we can conclude that the total contribution of term $(iv)$ is

$$
\begin{aligned}
& \sum_{\eta \in \mathcal{I}} \frac{1}{2} D_\eta^2 \sum_{m \neq k} (iv) \\
=& -\frac{\alpha}{2} \sum_{m \neq k} \frac{1}{(\lambda_k - \lambda_m)^2} v_k.
\end{aligned}
$$

Before proceeding, we prove the following lemma.

**Lemma C.1.** *Let $v^{(k)}$ be a set of orthonormal vectors and $B_{ij}$ a set of independent standard Brownian motions with $B_{ij} = B_{ji}$, for $i, j, k \in [n]$. We have that*

$$
\tilde{B}_{lk} := (v^{(l)})^T \begin{pmatrix} \sqrt{2} B_{11} & B_{12} & B_{13} & \dots & B_{1n} \\ B_{21} & \sqrt{2} B_{22} & B_{23} & \dots & B_{2n} \\ \vdots & & \vdots & & \vdots \\ B_{n1} & B_{n2} & B_{n3} & \dots & \sqrt{2} B_{nn} \end{pmatrix} v^{(k)}
$$

*is a Brownian motion with variance $1 + \delta_{lk}$, that is $\tilde{B}_{lk} \stackrel{\text{distr}}{=} \sqrt{1 + \delta_{lk}} B$ for $B$ a standard Brownian motion. $\tilde{B}_{lk}$ is independent of $\tilde{B}_{ab}$ for $(l, k) \neq (a, b)$ and $(l, k) \neq (b, a)$.*

*Proof.* We have for any $a, b \in [n]$ that

$$
\mathrm{d}\tilde{B}_{ab} = \sum_{i,j} (v^{(a)})_i (v^{(b)})_j \sqrt{1 + \delta_{ij}} \mathrm{d}B_{ij}.
$$

Thus, $\mathbb{E}\left[\mathrm{d}\tilde{B}_{ab}\right] = 0$. We further have for $c, d \in [n]$:

$$
\begin{aligned}
\mathbb{E}\left[\mathrm{d}\tilde{B}_{ab}\mathrm{d}\tilde{B}_{cd}\right] &= \sum_{ijkl}(v^{(a)})_i(v^{(b)})_j(v^{(c)})_k(v^{(d)})_l\sqrt{1+\delta_{ij}}\sqrt{1+\delta_{kl}}\,\mathbb{E}\left[\mathrm{d}B_{ij}\mathrm{d}B_{kl}\right] \\
&= \sum_{ijkl}(v^{(a)})_i(v^{(b)})_j(v^{(c)})_k(v^{(d)})_l\sqrt{1+\delta_{ij}}\sqrt{1+\delta_{kl}}\mathbb{1}_{(i,j)=(k,l)\vee(i,j)=(l,k)}\mathrm{d}t \\
&= \sum_{ij}(v^{(a)})_i(v^{(b)})_j\left\{\mathbb{1}_{i\neq j}\left[(v^{(c)})_i(v^{(d)})_j+(v^{(c)})_j(v^{(d)})_i\right]+\mathbb{1}_{i=j}(v^{(c)})_i(v^{(d)})_i\right\}(1+\delta_{ij})\mathrm{d}t \\
&= \sum_{ij}(v^{(a)})_i(v^{(b)})_j\left[(v^{(c)})_i(v^{(d)})_j+(v^{(c)})_j(v^{(d)})_i\right]\mathrm{d}t \\
&= \sum_{ij}(v^{(a)})_i(v^{(b)})_j(v^{(c)})_i(v^{(d)})_j\mathrm{d}t+\sum_{ij}(v^{(a)})_i(v^{(b)})_j(v^{(c)})_j(v^{(d)})_i\mathrm{d}t \\
&= (v^{(a)})^T v^{(c)}(v^{(b)})^T v^{(d)}\mathrm{d}t+(v^{(a)})^T v^{(d)}(v^{(b)})^T v^{(c)}\mathrm{d}t \\
&= \mathbb{1}_{(a,b)=(c,d)}\mathrm{d}t+\mathbb{1}_{(a,b)=(d,c)}\mathrm{d}t
\end{aligned}
\tag{62}
$$

Thus, in particular the process $\tilde{B}_{ab}$ has variance $t$ for $a \neq b$ and variance $2t$ for $a = b$.

Since the linear combination of independent Brownian Motions is jointly normal, we see from the Covariance property in eq. (62) that $\tilde{B}_{lk}$ is independent of $\tilde{B}_{ab}$ for $(l, k) \neq (a, b)$ and $(l, k) \neq (b, a)$. $\qquad\square$

We can now turn to the third summand of eq. (50).

We have

$$
\begin{aligned}
&\sum_{\eta\in\mathcal{I}}D_\eta\frac{\partial v_k}{\partial M_\eta}\mathrm{d}B_\eta \\
&=\sqrt{\alpha}\sum_{l\neq k}\frac{1}{\lambda_k-\lambda_l}v_l^T\sum_{\eta\in\mathcal{I}}\frac{\partial M}{\partial M_\eta}v_k\sqrt{1+\delta_\eta}\mathrm{d}B_\eta v_l \\
&=\sqrt{\alpha}\sum_{l\neq k}\frac{1}{\lambda_k-\lambda_l}v_l^T\begin{pmatrix}\sqrt{2}\mathrm{d}B_{11} & \mathrm{d}B_{12} & \mathrm{d}B_{13} & \ldots & \mathrm{d}B_{1n} \\ \mathrm{d}B_{21} & \sqrt{2}\mathrm{d}B_{22} & \mathrm{d}B_{23} & \ldots & \mathrm{d}B_{2n} \\ \vdots & & \vdots & & \vdots \\ \mathrm{d}B_{n1} & \mathrm{d}B_{n2} & \mathrm{d}B_{n3} & \ldots & \sqrt{2}\mathrm{d}B_{nn}\end{pmatrix}v_k v_l \\
&=\sqrt{\alpha}\sum_{l\neq k}\frac{1}{\lambda_k-\lambda_l}v_l\mathrm{d}\tilde{B}_{lk}
\end{aligned}
\tag{63}
$$

where we used Lemma C.1 in the last step so that $\mathrm{d}\tilde{B}_{lk}$ are symmetric Brownian motions with variance $1 + \delta_{lk}$.

Having established all the terms in the SDE in Theorem 3.2, we check that this dynamics of the eigenvectors gives rise to normalized vectors, assuming that the initial vectors $(v_1(0), \ldots, v_n(0))$ are normalized. To this end, it suffices to note that $v_k(t + \mathrm{d}t) = v_k(t) + dv_k(t)$ is normalized given that $v_k(t)$ is normalized, that is $(v_k(t))^2 = 1$ for any $t$. By continuity of $t \mapsto v_k(t)$, it suffices to show that $v_k(\mathrm{d}t) = v_k(0) + dv_k(0)$ remains normalized. We use the SDE from Theorem 3.2 to

compute that up terms of the order of $\mathrm{d}t^{3/2}$ or higher we have (omitting "(0)" in the notation)

$$
\begin{aligned}
(v_k(0) + \mathrm{d}v_k(0))^2 &= 1 + 2v_k^T \mathrm{d}v_k + (\mathrm{d}v_k)^2 \\
&= 1 + 2\left(\frac{-\alpha}{2}\right)\sum_{l \neq k}\frac{\mathrm{d}t}{(\lambda_k - \lambda_l)^2} + 2\sqrt{\alpha}\sum_{l \neq k}\frac{1}{\lambda_l - \lambda_k}\underbrace{v_k^T v_l}_{=0}\mathrm{d}w_{lk} \\
&\quad + \alpha \sum_{l \neq k}\sum_{j \neq k}\frac{\mathrm{d}w_{lk}\mathrm{d}w_{jk}}{(\lambda_l - \lambda_k)(\lambda_j - \lambda_k)} + \mathcal{O}(\mathrm{d}t^{3/2}) \\
&= 1 - \alpha\sum_{l \neq k}\frac{\mathrm{d}t}{(\lambda_k - \lambda_l)^2} + \alpha\sum_{l \neq k}\sum_{j \neq k}\frac{\delta_{jl}\mathrm{d}t}{(\lambda_k - \lambda_l)(\lambda_k - \lambda_j)} + \mathcal{O}(\mathrm{d}t^{3/2}) \\
&= 1 + \mathcal{O}(\mathrm{d}t^{3/2}).
\end{aligned}
\tag{64}
$$

Since contributions of $\mathcal{O}(\mathrm{d}t^{3/2})$ do not contribute to the dynamics for $\mathrm{d}t \to 0$, this implies that the $v_k(\mathrm{d}t)$ remain normalized. The argument may be iterated, i.e., $(v_k(2\mathrm{d}t))^2 = (v_k(\mathrm{d}t) + \mathrm{d}v_k(\mathrm{d}t))^2 = 1 + \mathcal{O}(\mathrm{d}t^{3/2})$ and so on, and thus the $v_k(t)$ remain normalized for all $t > 0$.

Finally, as an instructive consistency check, we also show that the eigenvectors remain orthogonal to each other (as they must, since $M$ is symmetric for all $t \geq 0$). This can be checked from the SDE in an analogous manner using again the continuity of $t \mapsto v_k(t)$. Note that for $k \neq l$ and given $v_k^T(0)v_l(0) = 0$, it suffices to show that $v_k^T(\mathrm{d}t)v_l(\mathrm{d}t) = 0$. Indeed, we have (omitting "(0)" in the notation)

$$
\begin{aligned}
(v_k(0) + \mathrm{d}v_k(0))^T(v_l(0) + \mathrm{d}v_l(0)) &= 0 + v_k^T dv_l + v_l^T dv_k + \mathrm{d}v_k^T dv_l \\
&= \sqrt{\alpha}\sum_{i \neq l}\frac{1}{\lambda_l - \lambda_i}v_k^T v_i \mathrm{d}w_{il} + \sqrt{\alpha}\sum_{j \neq k}\frac{1}{\lambda_k - \lambda_j}v_l^T v_j \mathrm{d}w_{jk} \\
&\quad + \alpha\sum_{j \neq k}\sum_{i \neq l}\frac{v_j^T v_i \mathrm{d}w_{il}\mathrm{d}w_{jk}}{(\lambda_k - \lambda_j)(\lambda_l - \lambda_i)} + \mathcal{O}(\mathrm{d}t^{3/2}) \\
&= \frac{\sqrt{\alpha}}{\lambda_l - \lambda_k}\underbrace{(\mathrm{d}w_{kl} - \mathrm{d}w_{lk})}_{=0} + \alpha\sum_{j \neq k}\sum_{i \neq l}\frac{\delta_{ij}\delta_{ik}\delta_{jl}\mathrm{d}t}{(\lambda_k - \lambda_j)(\lambda_l - \lambda_i)} + \mathcal{O}(\mathrm{d}t^{3/2}) \\
&= 0 + \mathcal{O}(\mathrm{d}t^{3/2}).
\end{aligned}
\tag{65}
$$

The argument may be iterated and thus the claim hold for all $t > 0$.

$\square$

## D  TIME REVERSAL: EXISTENCE AND UNIQUENESS

The time-reversal of the Dyson-BM in the sense of Ref. Anderson (1982) is given in Eq. (5). Since the drift coefficient in the Dyson-BM is not locally Lipschitz continuous, the existence and uniqueness of strong solutions to Dyson-BM and Eq. (5), and the applicability of Ref. Anderson (1982) are not obvious. While the existence and uniqueness of a strong solution to the Dyson-BM is well established (see, e.g., Lemma 4.3.3 in Ref. Anderson et al. (2009)), we here sketch how to ensure the other two points.

To ensure existence and uniqueness of a strong solution to Eq. (5), repeat the arguments in Lemma 4.3.3 in Ref. Anderson et al. (2009), where the divergent $x^{-1}$-term in the drift is replaced by the locally Lipschitz continuous approximation $\phi(x) = x^{-1}$ for $|x| \geq R^{-1}$ and $\phi(x) = xR^2$ otherwise. For any $R > 0$, the desired statements follow from the local Lipschitz continuity of the drift, and details on the limit $R \to 0$ can be found in Ref. Anderson et al. (2009). The only difference to the forward motion is the additional drift term containing the score function. However, since the dynamics is almost surely contained in the interior of the Weyl chamber (Katori & Tanemura, 2003), the propagator in the score-contribution is, as usual, dominated by white noise as $\mathrm{d}t \to 0$. Therefore, this term will not cause complications as $R \to 0$ and the arguments from Ref. Anderson et al. (2009) imply uniqueness and existence of a strong solution to Eq. (5).

Using the same regularization of the divergent term, for any $R > 0$ the statements of Ref. Anderson (1982) are directly applicable. Since this regularization is piecewise continuous, we can take the limit $R \to 0$ under time reversal. Since we just established the existence of the solution to Eq. (5), the limit $R \to 0$ converges. Furthermore, the arguments are expected to generalize (regularize $\phi(x) = x^{-2}$) to the eigenvector equations and their time reversal.

# E    TIME RESCALING

We rescale time for Dyson-BM. Let $T(s)$ be a continuous, differentiable rescaling of time, monotonically increasing, with $T(0) = 0$ (for convenience).

Let $\gamma_k(t) := \lambda(T(t))$. With this definition, the Ito SDE eq. (Dyson-BM) corresponds to the Ito integral

$$\gamma_k(t) := \lambda_k(T(t)) = \lambda_k(0) + \int_0^{T(t)} \left( \alpha \sum_{i \neq k} \frac{1}{\lambda_k(s) - \lambda_i(s)} - \beta \lambda_k(s) \right) \mathrm{d}s + \int_0^{T(t)} \sqrt{2\alpha} \mathrm{d}B_k(s)$$

$$= \int_0^t \left( \alpha \sum_{i \neq k} \frac{1}{\lambda_k(T(s)) - \lambda_i(T(s))} - \beta \lambda_k(T(s)) \right) T'(s) \mathrm{d}s + \int_0^t \sqrt{2\alpha T'(s)} \mathrm{d}B_k(s)$$

(using Thm. 8.5.7 in Øksendal (2003))

$$= \int_0^t \left( \alpha \sum_{i \neq k} \frac{1}{\gamma_k(s) - \gamma_i(s)} - \beta \gamma_k(s) \right) T'(s) \mathrm{d}s + \int_0^t \sqrt{2\alpha T'(s)} \mathrm{d}B_k(s) \tag{66}$$

which we can write in SDE notation

$$\mathrm{d}\gamma_k(t) = \left( \alpha \sum_{i \neq k} \frac{1}{\gamma_k(t) - \gamma_i(t)} - \beta \gamma_k(t) \right) T'(t) \mathrm{d}t + \sqrt{2\alpha T'(t)} \mathrm{d}B_k(t). \tag{67}$$

By using $T(t) := \frac{1}{\alpha} t$, we obtain

$$\mathrm{d}\gamma_k(t) = \left( \sum_{i \neq k} \frac{1}{\gamma_k(t) - \gamma_i(t)} - \frac{\beta}{\alpha} \gamma_k(t) \right) \mathrm{d}t + \sqrt{2} \mathrm{d}B_k(t), \tag{68}$$

so that we can summarize the two parameters to $\eta := \frac{\beta}{\alpha}$:

$$\mathrm{d}\gamma_k(t) = \left( \sum_{i \neq k} \frac{1}{\gamma_k(t) - \gamma_i(t)} - \eta \gamma_k(t) \right) \mathrm{d}t + \sqrt{2} \mathrm{d}B_k(t) \tag{69}$$

which we call "$\gamma(\eta)$-SDE".

Dyson's conjecture says that the $\lambda(\frac{1}{N}, \frac{1}{2})$-SDE converges to global equilibrium in time $\Theta(1)$ (see Yang (2022)). Running $\lambda(\frac{1}{N}, \frac{1}{2})$ until time 1 is the same as running the $\gamma(\frac{N}{2})$-SDE until time $T(1) = \frac{1}{N}$.

# F    STEPSIZE CONTROLLER

Dyson Brownian Motion almost surely never crosses the singularities. Hence, conditioning on non-crossing corresponds to conditioning on a probability 1 event, which does not change the dynamics. Given the noise, we can thus calculate the maximal step size, beyond which we would cross the singularity. This is a very useful upper bound, which we employ in practice to get the numerical scheme working. It has two effects: (1) close to the boundary of the Weyl Chamber, it avoids numerically stepping over the singularities and (2) far from the boundary, it allows for larger step size, increasing efficiency.

---

**Algorithm 2** Forward stepsize controller for Dyson SDE

---

1: **Input:** position $\lambda \in \mathbb{R}^n$, time $t \in \mathbb{R}^+$, independent normal samples $u \sim \mathcal{N}^n$.
2: **for** $k \in [n-1]$ **do**
3:     $\delta t_k \leftarrow$ maximal step size based on $\lambda_{k+1} - \lambda_k$ and samples $u_k$, $u_{k+1}$ as described in Appendix F.1
4: **end for**
5: **Output:** stepsize $\min_i \delta t_i$.

---

Figure 6: Forward step size controller which exploits that non-crossing of paths happens with probability 1. The exact calculations are carried out in Appendix F.1.

### F.1 FORWARD IN TIME

The difference between components $k$ and $k+1$ of $\lambda$ after a first-order discretization step of size $\mathrm{d}t$ reads for any time $t \in \mathbb{R}^+$

$$\Delta_k(t + \mathrm{d}t) \coloneqq \lambda_k(t + \mathrm{d}t) - \lambda_{k+1}(t + \mathrm{d}t) \tag{70}$$

$$= \lambda_k(t) - \lambda_{k+1}(t) + \mathrm{d}t f_k(t) + \sqrt{\mathrm{d}t} g_k \tag{71}$$

with functions $f_k(t) \coloneqq \alpha \left( \sum_{i \neq k} \frac{1}{\lambda_k(t) - \lambda_i(t)} - \sum_{i \neq k+1} \frac{1}{\lambda_{k+1}(t) - \lambda_i(t)} \right) - \beta \left( \lambda_k(t) - \lambda_{k+1}(t) \right)$,

$g_k \coloneqq \sqrt{2\alpha} \left( X_k - X_{k+1} \right)$ where the $X_j \overset{iid}{\sim} \mathcal{N}(0,1)$ are the independent standard gaussian random variables taken in the update step of the numerical SDE scheme. Note that $\Delta_k$ is a random variable, but if the step size is sufficiently small we must have almost surely

$$\Delta_k(t + \mathrm{d}t) > 0.$$

To find the maximal step size, we observe that the equation above is a quadratic function in the substituted $\tau \coloneqq \sqrt{\mathrm{d}t}$ yielding the inequality that

$$\tau^2 + \frac{g_k}{f_k(t)}\tau + \frac{\Delta_k(t)}{f_k(t)} \qquad \text{is} \qquad \begin{cases} > 0 & \text{if } f_k(t) > 0, \\ < 0 & \text{if } f_k(t) < 0. \end{cases} \tag{72}$$

We treat first the case $f_k(t) > 0$. The inequality is equivalent to

$$\left( \tau + \frac{g_k}{2 f_k(t)} \right)^2 > \left( \frac{g_k}{2 f_k(t)} \right)^2 - \frac{\Delta_k(t)}{f_k(t)}, \tag{73}$$

where we know that $\frac{\Delta_k(t)}{f_k(t)} > 0$ must hold. Hence, if $g_k > 0$, any stepsize $\mathrm{d}t > 0$ works. Otherwise, if $g_k < 0$, the quadratic formula shows immediately that both roots will be in the $\tau > 0$ regime. Since the parabola is convex in $\tau$, for inequality (73) to be fulfilled, we take $\tau$ in the range from $0$ to the smallest root. For $\mathrm{d}t$, that means

$$\mathrm{d}t \in \left( 0, \frac{1}{4} \left( -\frac{g_k}{f_k(t)} - \sqrt{\frac{g_k^2}{f_k(t)^2} - 4\frac{\Delta_k(t)}{f_k(t)}} \right)^2 \right). \tag{74}$$

If $f_k(t) < 0$, we have

$$\left( \tau + \frac{g_k}{2 f_k(t)} \right)^2 < \left( \frac{g_k}{2 f_k(t)} \right)^2 - \frac{\Delta_k(t)}{f_k(t)}. \tag{75}$$

Since $\frac{\Delta_k(t)}{f_k(t)} < 0$, we know that the inequality is satisfied at $\tau = 0$, and that the largest root $\tau_2$ will be positive:

$$\tau_{1,2} = \frac{1}{2} \left( -\frac{g_k}{f_k(t)} \mp \sqrt{\frac{g_k^2}{f_k(t)^2} - 4\frac{\Delta_k(t)}{f_k(t)}} \right).$$

Thus, any stepsize in the following interval is valid

$$\mathrm{d}t \in \left(0, \frac{1}{4}\left(-\frac{g_k}{f_k(t)} + \sqrt{\frac{g_k^2}{f_k(t)^2} - 4\frac{\Delta_k(t)}{f_k(t)}}\right)^2\right)$$

Note that $f_k(t) \neq 0$ and $g_k \neq 0$ almost surely.

### F.2 Backward in Time

Backwards in time, we carry out the analogous computation for the more involved backwards SDE in eq. (5). In essence, this boils again down to solving a quadratic equation and considering all edge cases.

## G Shooting mechanism

In the forward dynamics, if a certain step size would lead to the probability-0 event of leaving the Weyl-Chamber, we know that the source of the error is the finite-time-step discrete approximation, so that decreasing the step size will always provide a fix. In the backward dynamics, however, this is not guaranteed: As described in Section 3.2, a possible reason for this probability-0 event in the backwards dynamics is that the score $s_\theta$ might be not perfectly learned: $s_\theta(\lambda, t) \neq s(\lambda, t)$ for some $\lambda, t$. To avoid this probability-0 event, we use the following "shooting mechanism": If we were to leave the Weyl-Chamber, we replace the learned score with the analytically known score in the invariant state, eliminating the use of the neural network at that point and leading effectively to a repulsion with the negative forward drift. This mechanism is not expected to change the dynamics in any unfavorable way since it is only applied very rarely, and only in cases where the actual learned dynamics is a much worse approximation (since it would give rise to measure zero events).

## H Complexity dependence on graph size $n$ of numerical step for Dyson-BM and Eigenvector-SDE

When simulating the Dyson-BM numerically, for a fixed eigenvalue $\lambda_k$, we need to calculate its distance to any $\lambda_\ell$ for $\ell \in [n] \setminus \{k\}$ in order to obtain the repulsion force entering the drift of the Dyson-BM. Since we need to do this for each component, we have for a fixed time step a complexity of $\mathcal{O}\left(n^2\right)$.

When simulating Eigenvector-SDE numerically, for one time step, we update the entire set of $n$ eigenvectors together. Each of the $n(n-1)/2$ entries of the Lie algebra can be computed in time $\Theta\left(1\right)$ (note that $E_{(l,k)}$ is a basis element of the Lie algebra). This update requires thus $\mathcal{O}\left(n^2\right)$. Accounting for the projection from the Lie algebra in the Lie group, this yields a complexity of $\mathcal{O}\left(n^3\right)$ for an eigenvector update.

Since Dyson-BM decouples from Eigenvector-SDE, we thus have a complexity for an update step of $\mathcal{O}\left(n^2\right)$ if one is interested in only the spectrum, and a complexity of $\mathcal{O}\left(n^3\right)$ if one is interested in the spectrum and eigenvectors. Note that this modularity allows the practitioner to first learn the spectrum in isolation and do any hyperparameter tuning steps on the (smaller) spectral model, and then use this already tuned model to train and optimize for the remaining eigenvector dynamics.

In comparison, an OU-based approach on the entire graph with a GNN-based algorithm requires for a fixed time point an update of the SDE in $n^2$ entries in time $\mathcal{O}\left(n^2\right)$ followed by $C$ message passing steps (a GNN is trained using message passing, where $C$ is a constant defining how many hops a message is being passed), with each message passing taking time $\mathcal{O}\left(n^3\right)$, leading to an overall complexity of $\mathcal{O}\left(n^3\right)$ per time step, assuming that $C$ is constant.

## I Empirical Considerations

While we give a thorough analysis of the computational complexity of each update step, the number of time steps is determined by the adaptive step size algorithm. For practical reasons, our method

can be configured with a minimal (and maximal) step size. We illustrate the distribution of the number of time steps on the Brain dataset in Figure 7, where it can be observed that the number of time steps taken concentrates very well. Note that this number of time steps perfectly suffices even for a dataset as large as Brain, which contains $15'000$ graphs. Note further that while these are the number of steps the numerical integrator takes forward, learning has to happen only on much fewer steps, through the described pre-defined schedule. In the case of Brain, learning happens only on $651$ time points. The numerical generation of $105'000$ paths with step number depicted in Figure 7 took in total $5$ seconds. For memory considerations, note that we require for training only a single H100 GPU ($80$ GB Video Random Access Memory).

We further note that the skip event (see Algorithm 1) is indeed a rare event: In the Brain setting, the empirical probability of a step being a skip is $10^{-5}$. Similarly, shootings in the analogous backwards runs happen rarely. Note that skips and shoots are needed for correctness and stability, but have due to their rarity no negative impact on runtime.

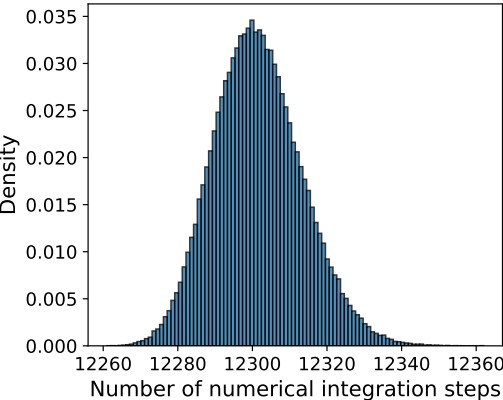

Figure 7: Concentration of time steps of the numerical integrator: Within $5$ second, we generated $105'000$ forward paths using our numerical scheme, where the total number of steps per path are distributed as depicted in this histogram. As can be seen, the distribution concentrates well.

## J    INFERENCE

We describe in Algorithm 3 the sampling procedure with the shooting mechanism (see Appendix G for details) incorporated, ensuring that we remain in the Weyl Chamber.

## K    ENGINEERING

We follow Karras et al. (2024) by using SiLU activations. We further use EMA to average over multiple runs (Song & Ermon, 2020).

A key strength of DyDM is that the score $s_\theta(\lambda, t)$ can be parameterized with any learning architecture, without being constrained to GNNs or graph transformers. We demonstrate this by parameterizing the score with a simple MLP. The size of layers varies by application, and we document it for each dataset in Github. For instance, for the large dataset of $15'000$ brain ego graphs, our model consists of a hidden MLP of depth $4$, where the input and output layer have width $64$ and the hidden layers have width $256$. The space + time data is first scaled up with a linear layer from size $n + 1 = 11$ followed by a batch norm to feed into the hidden MLP, and upon processing through the hidden MLP, it gets scaled down through a simple linear layer to size $n = 10$. In the hidden MLP, we employ as nonlinearities scaled SiLU functions, as argued by Karras et al. (2024).

To make full use of the GPU memory, in one epoch, we sample $N_{paths}$ paths in parallel. If the dataset is too small to fill the GPU memory, we sample multiple, independent, paths of the same data points in parallel. From these samples, we update the score network $s_\theta$ in smaller batches. Proceeding in

---

**Algorithm 3** DyDM sampling

---

1: **Input:** dimension $n$, schedule $\mathcal{T} = \{t_j\}$, trained score network $s_\theta$
2: $\lambda(T) \leftarrow$ Spectrum of a random GOE matrix.         ▷ Sampling from invariant distribution.
3: $t \leftarrow T$
4: **while** $t > 0$ **do**
5:      Let $u \sim \mathcal{N}(0, I_n)$
6:      $\delta t \leftarrow$ BackwardStepsizeController$(\lambda(t), u)$         ▷ Step size as described F.2.
7:      $\hat{s}_\theta \leftarrow$ Interpolation of $s_\theta(\lambda, t_j)$ and $s_\theta(\lambda, t_{j+1})$ for $t_j, t_{j+1} \in \mathcal{T}$ closest points in schedule
8:      BackwardsDrift $\leftarrow$ drift of eq. (5) at time $t$ and point $\lambda(t)$ with score $s_\theta(\lambda, t_{j+1})$
9:      **if** $\lambda_t -$ BackwardsDrift $\cdot \delta t - u\sqrt{2\alpha\delta t} \in$ Weyl Chamber **then**
10:          $\lambda_{t-\delta t} \leftarrow \lambda_t -$ BackwardsDrift $\cdot \delta t - u\sqrt{2\alpha\delta t}$
11:      **else**         ▷ Otherwise – in a rare event – shooting mechanism triggers (see Appendix G)
12:          $\lambda_{t-\delta t} \leftarrow \lambda_t - (-$ForwardDrift$) \cdot \delta t - u\sqrt{2\alpha\delta t}$         ▷ ForwardDrift as in 3.1
13:      **end if**
14:      $t \leftarrow t - \delta t$
15: **end while**
16: **Output:** spectral sample $\lambda(0) \in \mathbb{R}^n$

---

Figure 8: Sampling from the Dyson Diffusion Model. The eq. (5) is evolved backward in time with the shooting mechanism and an adaptive step size, ensuring that the paths remain in the Weyl Chamber.

this way ensures that JAX uses the full potential of the GPU. These parameters ($N_{paths}$ and batch size) can be specified in the configuration file.

We implement the model in Jax and Equinox (Kidger & Garcia, 2021).

### K.1   CHOICE OF TIME GRID

As outlined in Section 3.2, we choose an exponential time grid on which the objective is learned. This is due to the mixing behavior of Dyson's Brownian Motion. For instance, for the Brain dataset, we use exponential time grid detailed in Table 2.

Table 2: Example of exponential time grid, here for the brain dataset which contains in total $15'000$ graphs. $dt$ is $0.05$, and the final time is $T = 12.0$.

| from | to | stepsize |
|------|------|----------|
| 0 | 1/8 | $1/64 \cdot dt$ |
| 1/8 | 1/4 | $1/32 \cdot dt$ |
| 1/4 | 1/2 | $1/16 \cdot dt$ |
| 1/2 | 1 | $1/8 \cdot dt$ |
| 1 | 2 | $1/4 \cdot dt$ |
| 2 | 3 | $1/2 \cdot dt$ |
| 3 | 7 | $1 \cdot dt$ |
| 7 | T | $2 \cdot dt$ |

### K.2   PREPROCESSING AND DEALING WITH DIFFERENT DIMENSIONS

As is the case for an Ornstein-Uhlenbeck diffusion, the speed of convergence depends for Dyson-BM on (1) the coefficients and (2) the initial condition. We can thus choose to significantly vary the final time $T$ or rescale the initial condition. We choose the latter (although both options are feasible). To that end, we rescale the spectra with an affine transformation in a preprocessing step. In more detail: For a given set of graphs, we perform an eigendecomposition, and rescale so that among the entire dataset the largest eigenvalue is at most $\lambda_{\max}$ and at least $\lambda_{\min}$. For instance, on the

benchmark models we chose $\lambda_{\max} = 5$, $\lambda_{\min} = -5$. If a spectrum has eigenvalues of multiplicities greater than 1, we perform an $\epsilon$-perturbation, where the $\epsilon$ depends on the distance of the closest eigenvalues in the dataset. In postprocessing, the spectra are scaled back and the perturbation is undone for eigenvalues that are at most $\epsilon$ apart after generation. Note that with this preprocessing, only one eigendecomposition per training graph is necessary.

## L    Challenges of Dyson's Brownian Motion

Using Dyson's Brownian Motion for a diffusion presents several challenges, all of which we overcame in this paper. First, the Dyson SDE is not an OU process, but instead an SDE with singularities of order $\mathcal{O}(1/(\lambda_k - \lambda_l))$ in the drift, posing both theoretical and numerical challenges. Second, the conditional density $p(x \mid x_0)$ is non-Gaussian and challenging to obtain, as with any non-OU diffusion process. Therefore, we do not have access to a canonical loss function. Finally, the non-availability of conditional distributions means that training is not simulation-free.

We overcome the obstacles mentioned above and provide a diffusion model for the spectra of graphs based on the Dyson SDE (Fig. 1). The model is not only efficient but is also able to distinguish between spectra of graphs that GNNs are blind to (Fig. 2). In addition, with DyDM, no ad hoc data augmentation is necessary. Further, through Eigenvector-SDE, we give the dynamics of the remaining information in form of conditioned eigenvector dynamics, hence making them accessible for future work devoted to eigenvector diffusion.

### L.1    Why not log-transform the SDE?

One idea could be to transform the spectral SDE into terms $\lambda_1, \lambda_2 - \lambda_3, \ldots, \lambda_{n-1} - \lambda_n$ and take logarithms, to avoid singularities. However, this is no desirable for multiple reasons: First, upon applying Ito to this SDE, we (i) lose the $\log$ and obtain singularities again and (ii) get higher order singularities $d \log(x_t) = \frac{1}{x_t} dx_t - \frac{1}{2x_t^2} (dx_t)^2$. Second, the space that would need to be sampled would certainly not decrease, since now the transformed domain reaches from $-\infty$ to $+\infty$. Hence, we choose the method described in the main part.

## M    Why not learn on all $n!$ many graph representations?

In short, learning $n!$ more data is much harder. This point has been mentioned by previous literature, e.g. Niu et al. (2020). However, if we go deeper, the interested reader might wonder why exactly.

### M.1    Rigorous argument

We give here a toy example, where the challenge can be phrased rigorously. Suppose we have a binary matrix $X \in \{0,1\}^{m,k}$ with independent entries, which are for $i \in [m], j \in [k]$ distributed as $X_{ij} \sim \text{Bernoulli}(p_j)$ for some unknown $p_j \in [0,1]$. The task is to estimate the $p_j$. The motivation for this example stems from the following setting: We want to learn the probability of $k$ objects (for instance, graphs), each having $m$ representations (for instance, representations of graphs such as adjacency matrices). Each column of the matrix $X$ thus consists of all representations of the same object. We define two estimators, with $\text{est}_A$ using the inductive bias and $\text{est}_B$ not using it.

To that end, suppose we have $N$ uniformly at random obtained samples $Z_1, \ldots, Z_N$. In more detail, that means that we sample for each $\ell \in [N]$ a pair of indices $(i_\ell, j_\ell) \in [m] \times [k]$ uniformly at random, and describe the obtained sample by $Z_\ell \sim X_{i_\ell, j_\ell}$. Estimator $\text{est}_A$ makes use of the inductive bias. That is, for $u \in [m], v \in [k]$ we define

$$\text{est}_A^{(u,v)} := \frac{k}{N} \sum_{\ell \in [N]} Z_\ell \mathbb{1}_{j_\ell = v}.$$

Estimator $\text{est}_B$ does not make use of the inductive bias. That is, for $u \in [m], v \in [k]$, we define

$$\text{est}_B^{(u,v)} := \frac{k \cdot m}{N} \sum_{\ell \in [N]} Z_\ell \mathbb{1}_{i_\ell = u, j_\ell = v}.$$

Clearly, both estimators are unbiased: $\mathbb{E}\left[\text{est}_{\text{A}}^{(\text{u,v})}\right] = \mathbb{E}\left[\text{est}_{\text{B}}^{(\text{u,v})}\right] = p_v$. However, their mean squared error, defined for $X \in \{A, B\}$ as

$$\text{MSE}(\text{est}_{\text{X}}) := \frac{1}{m \cdot k} \sum_{u \in [m], v \in [k]} \mathbb{E}\left[\left(\text{est}_{\text{X}}^{(\text{u,v})} - p_v\right)^2\right],$$

varies significantly between $\text{est}_{\text{A}}$ and $\text{est}_{\text{B}}$.

**Corollary M.1** (Mean square error). *In dependence of the problem size $m$ and number of samples $N$, the mean square error of $\text{est}_{\text{A}}$ is of order $\Theta(1/N)$, while the mean square error of $\text{est}_{\text{B}}$ is of order $\Theta(m/N)$.*

To prove Corollary M.1, we derive the mean squared error for both estimators.

**Lemma M.2** (MSE of $\text{est}_{\text{A}}$). *The mean squared error of estimator $\text{est}_{\text{A}}$ is*

$$\text{MSE}\left(\text{est}_{\text{A}}\right) = \frac{1}{N} \sum_{v \in [k]} \left(p_v - \frac{1}{k}p_v^2\right).$$

*Proof.* We have for $u \in [m], v \in [k]$

$$\mathbb{E}\left[\left(\text{est}_{\text{A}}^{(\text{u,v})} - p_v\right)^2\right]$$

$$= \mathbb{E}\left[\left(\frac{k}{N}\sum_{\ell \in [N]} Z_\ell \mathbb{1}_{j_\ell = v} - p_v\right)^2\right]$$

$$= \frac{k^2}{N^2} \sum_{\ell \in [N]} \mathbb{E}\left[\sum_{o \in [N]} Z_\ell \mathbb{1}_{j_\ell = v} X_o \mathbb{1}_{j_o = v}\right] - 2\frac{k}{N}\mathbb{E}\left[\sum_{\ell \in [N]} Z_\ell \mathbb{1}_{j_\ell = v}\right] p_v + p_v^2$$

$$= \left(1 - \frac{1}{N}\right)p_v^2 + \frac{k}{N}p_v - 2p_v^2 + p_v^2$$

$$= \frac{k}{N}p_v - \frac{1}{N}p_v^2.$$

Averaging over all $u \in [m], v \in [k]$, we obtain the desired result. $\square$

**Lemma M.3** (MSE of $\text{est}_{\text{B}}$). *The mean squared error of estimators $\text{est}_{\text{B}}^{(\text{u,v})}$ for $u \in [m], v \in [k]$ is*

$$\text{MSE}\left(\text{est}_{\text{B}}\right) = \frac{1}{N} \sum_{v \in [k]} \left(m \cdot p_v - \frac{1}{k}p_v^2\right).$$

*Proof.* We have for $u \in [m], v \in [k]$

$$\mathbb{E}\left[\left(\text{est}_{\text{B}}^{(\text{u,v})} - p_v\right)^2\right]$$

$$= \mathbb{E}\left[\left(\frac{k \cdot m}{N}\sum_{\ell \in [N]} Z_\ell \mathbb{1}_{i_\ell = u, j_\ell = v} - p_v\right)^2\right]$$

$$= \frac{k^2 m^2}{N^2} \sum_{\ell \in [N]} \mathbb{E}\left[\sum_{o \in [N]} Z_\ell \mathbb{1}_{i_\ell = u, j_\ell = v} Z_o \mathbb{1}_{i_o = u, j_o = v}\right] - 2\frac{k\,m}{N}\mathbb{E}\left[\sum_{\ell \in [N]} Z_\ell \mathbb{1}_{i_\ell = u, j_\ell = v}\right] p_v + p_v^2$$

$$= \left(1 - \frac{1}{N}\right)p_v^2 + \frac{k\,m}{N}p_v - 2p_v^2 + p_v^2$$

$$= \frac{k\,m}{N}p_v - \frac{1}{N}p_v^2.$$

Summing over all estimators for $u \in [m], v \in [k]$ gives the desired result. $\square$

By considering the estimation problem as a problem of parameters $m$, $N$, Corollary M.1 follows directly from Lemma M.2, Lemma M.3.

The same argument may be carried out with Normal instead of Bernoulli random variables.

## N  WL-EQUIVALENCE OF REGULAR GRAPHS

We now prove Lemma 2.1.

**Lemma 2.1 (restated).** *For every fixed $n, k \in \mathbb{N}$, all $k$-regular graphs $G \in \mathcal{G}^n$ are WL equivalent. Moreover, every graph $G \in \mathcal{G}^n$ that is WL equivalent to a $k$-regular graph is $k$-regular.*

*Proof.* Recall that the 1-Weisfeiler-Leman (WL) algorithm tests graph equivalence by iteratively updating vertex *colors*. Initially, all vertices share one color. In each step, a vertex's new color is determined by the multiset (a set allowing for duplicates) of its own color and its neighbors' current colors. This continues until the coloring stabilizes. Two graphs are WL-equivalent if this process generates identical color counts (histograms) at every step.

For a $k$-regular graph, since every vertex has exactly $k$-neighbors, in each iteration of the WL-algorithm all vertices have the same color. Thus in particular the histograms are always the same and therefore any two $k$-regular graphs are WL-equivalent.

A $k$-regular graph is not WL-equivalent to a $\ell$-regular graph for $k \neq \ell$ since the colors after the first iteration are distinct as the number of neighbors is different. Moreover, if a graph is not regular, then the first iteration must assign a different new color to at least two vertices in the first iteration. Therefore, the color histogram is not the same as the color histogram of a regular graph and thus they are not WL-equivalent. This shows that the WL-equivalence class of a $k$-regular graph is exactly the set $k$-regular graphs. □

In particular, it follows from Morris (Morris et al., 2019) that GNNs cannot distinguish $k$-regular graphs.

## O  LAPLACIAN SPECTRUM

The Laplacian spectrum captures key properties of a graph, such as the Fiedler Value encoding the algebraic connectivity of the graph. Our model works perfectly well on the Laplacian spectrum, for which we plot the marginal densities on the datasets "Brain" and "Community Small". Note that the model perfectly learns key properties such as the fact that the smallest eigenvalue is $0$ and learns the distributions well.

For Brain (Figure 9), we plot for each eigenvalue $\lambda_i$ a histogram of the DyDM-generated marginal (blue) compared to a histogram of the Ground Truth (note that brain test for generalization). For the commonly used "Community Small" dataset, we show in Figure 10 the first 12 Eigenvectors, since only 73 graphs have more than 12 nodes.

Note that in this Section, we use the convention $\lambda_1 \leq \cdots \leq \lambda_n$ to stay consistent with Graph Laplacian literature.

## P  DATASETS

**WL-bimodal**  The WL-Bimodal graph consists of $80\%$ graph $A$ and $20\%$ graph B (see Fig. 2) adjacency matrices. We drew among all permutations $5'000$ permutations uniformly at random and shuffled the graphs. Hence, we have $5'000$ matrices, of which $4'000$ represent graph $A$ and $1'000$ represent graph $B$. Adjacency matrices representing the same graph are not identical matrices, but rather permutations of each other, making this dataset relevant to *graph* generative models. The first $80\%$ of this dataset are used for training, the remaining $20\%$ are used for testing.

**Community-small**  This standard benchmark dataset (Niu et al., 2020; Jo et al., 2022; You et al., 2018) consists of 100 graphs of size up to 20 vertices. We comment in Appendix Q.1 on the small

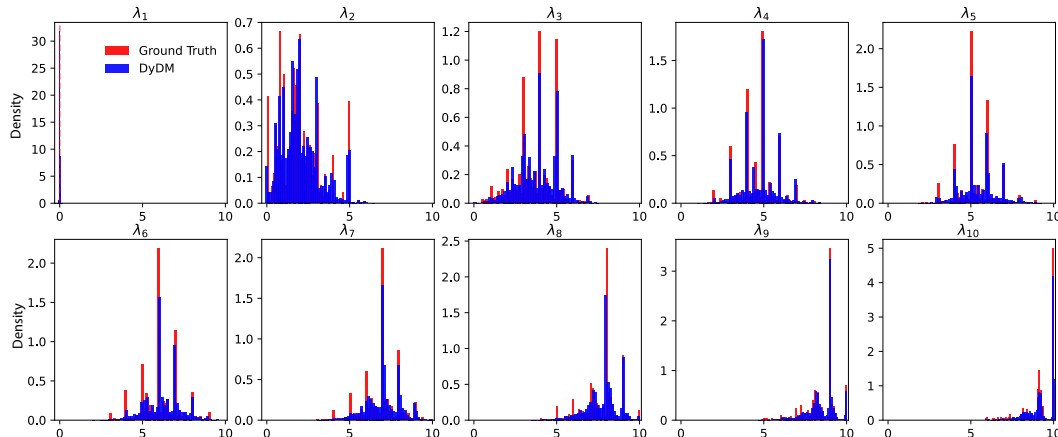

Figure 9: Performance of DyDM on the Laplacian of the Brain dataset compared to the underlying distribution: DyDM perfectly learns the (rough) distribution and key properties such as the smallest eigenvalue always being 0 (faint line, since it is very concentrated) or the distribution of the Fiedler Value ($\lambda_2$).

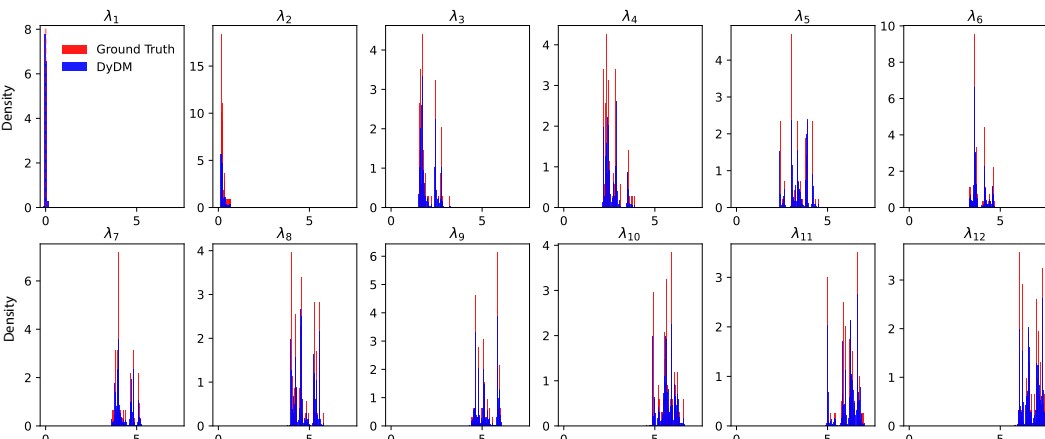

Figure 10: Performance of DyDM on the Laplacian of the Community Small dataset: DyDM learns the distribution and key properties very well.

dataset (100 graphs) compared to the big dimension (up to 20 vertices) and the thereby induced effect of undersampling.

**Brain** We report this dataset in our repository. In detail, we construct from the brain graph Amunts et al. (2013); Rossi & Ahmed (2015) so-called ego-graphs. That is, we take the (distance 1) neighborhoods of vertices, and consider the induced subgraph. From those, we generate $15'000$ graphs of size $n = 5$ to $n = 10$ vertices with eigenvalue multiplicity up to 3, with the closest eigenvalues – which are not multiplicities – having distance 0.036. We take 70% as train graphs, 15% as validation, and the remaining 15% as test graphs.

## Q COMPARING TO BENCHMARK MODELS

### Q.1 ON UNDERSAMPLING

For the "bimodal" case, we have sufficient statistics for the 10-dimensional space $C_{10}$ ($N = 5'000$ graph samples, each isomorphic to one of two graphs) and know in addition the underlying distribution; hence, an extensive interpretation of this result is appropriate. For a fair comparison, we thus

follow the standard test/train split procedure as reported in (Jo et al., 2022; You et al., 2018; Niu et al., 2020) using 80% of the data as train data and the remaining 20% as test data.

Conversely, the standard benchmark set "community small" (Niu et al., 2020; Jo et al., 2022; You et al., 2018) contains *only* 100 graphs, and each has a size of up to $n = 20$ vertices. Thus, a comparison from the learned distribution based on (few) training samples to (very few) test samples suffers from undersampling. This becomes very stark if one considers the following issue: If one would extract 80% of the dataset for training, *where* they are taken from already matters a lot: Whether they are taken from the front or back changes the maximal graph size in the training set. Depending on whether the training data is taken from the front or back of the dataset, a model trained on the training set might thus have no possibility to learn the correct maximal graph size. More generally, poorness of benchmarks in graph generative learning has been recently addressed by Bechler-Speicher et al. (2025). To offer some consistent comparison, we do include the standard benchmark "community small", but focus on *memorization* rather than the (on those benchmarks untestable) generalization.

To overcome the issue of undersampling, we construct a set of $15'000$ ego-graphs from the brain dataset Amunts et al. (2013) as described in Appendix P, which is sufficiently large to *not* suffer from undersampling. We train both our model and DiGress on 70% ($= 10'500$ graphs), perform hyperparameter tuning (see below for details on the DiGress hyperparameter tuning) on a validation set of 15%, and test on the remaining 15%.

## Q.2 GDSS

To compare to the GDSS model Jo et al. (2022), we take the following approach to ensure a fair comparison: The GDSS model has been trained and optimised on the ego small and community small dataset, so that we take for these datasets the snapshots and hyperparameters given by the original paper Jo et al. (2022). For the remaining datasets, we start with the settings from the community-small dataset since that has similar size, adapt the maximal number of vertices to the dataset, and then perform hyperparameter tuning as described in Appendix C of the GDSS paper: We form a grid search on the model's following parameters: The scale coefficient in $\{0.1, 0.2, 0.3, 0.4, 0.5, 0.6, 0.7, 0.8, 0.9, 1.0\}$, the signal-to-noise ratio in $\{0.05, 0.1, 0.15, 0.2\}$, and in addition to the GDSS paper, we also try different $\beta_{max}$ in $\{1, 10, 20\}$, and try different batch sizes $\{128, 4096\}$. We test with and without EMA. The motivation for the additional parameters we hyper tune is that we observed that they further improve the GDSS model. This gives full fairness to the model. The training time on one H100 GPU per job was about 24 hours for each of the configurations with batch size 128 and approximately 1 hour and 45 minutes for each of the configurations with batch size 4096.

On the Two-WL graph case, our hyperparameter tuning resulted in 280 models that we trained and sampled from. From each of those 280 models, we generated $1'000$ graphs. Most of those models worked fine, that is 269 models did not contain NaNs in their output. From those, we select the best model based on the following relative error: From the generated samples, we calculate the share of spectra $\epsilon$-close to the spectrum of graph $A$, say $\hat{p}_A$ and the share of spectra $\epsilon$-close to graph $\hat{p}_B$ (we choose the $l_2$ distance with $\epsilon = 0.2$). Recall that in the ground truth, we have $p_A = 0.8$, $p_B = 0.2$. We then selected the best model based on the relative error

$$\frac{|\hat{p}_A - p_A|}{p_A} + \frac{|\hat{p}_B - p_B|}{p_B}.$$

In summary, we invested a lot of resources in following both the hyperparameter tuning given in the GDSS paper and, in addition, tried new hyperparameters, leading to the 280 models that we trained and sampled from. This ensures maximal fairness.

## Q.3 EDP-GNN

We proceeded analogously to GDSS: We used the given configurations for community small. We observe that the model already performs hyperparameter tuning of the noise scales during sampling. For the other datasets, we have used the given configurations for community small and in addition tried the learning rates $\{0.001, 0.0002\}$, number of diffusion steps $\{1'000, 2'000\}$ and number of layers $\{4, 6\}$. In the two graph case, for example, the optimal configuration was with learning rate

0.0002, number of diffusion steps being equal to $2'000$ and 6 layers. The training time for $5'000$ epochs on a H100 GPU was approximately 10 hours.

### Q.4    DIGRESS

As with the previous models, we used the given configurations for community small. For the other datasets, we have used the given configurations for community small and in addition tried the learning rates $\{0.001, 0.0002\}$, weight decay parameters $\{10^{-2}, 10^{-12}\}$, number of diffusion steps $\{500, 1'000\}$ and number of layers $\{5, 8\}$. The model was quite robust to hyperparameter tuning and in the two graph example, for all parameters the model sampled the $80\%$-graph with likelihood between $75\%$ and $82\%$. The training time for $1'000$ epochs on a H100 GPU was approximately 3 hours.

### Q.5    CONGRESS

Since Congress and DiGress come from the same paper, we have proceeded almost exactly as in DiGress. We tried the learning rates $\{0.001, 0.0002\}$, weight decay parameters $\{10^{-2}, 10^{-12}\}$, number of diffusion steps $\{500, 1'000\}$ and number of layers $\{6, 8\}$. The model was again rather robust to hyperparameter tuning, yet not as much as Digress. In the two graph example, for all parameters the model sampled the $80\%$-graph with likelihood between $10\%$ and $25\%$. The default learning rate from the community-small configuration was 0.0002, yet we have observed that the results were significantly better with learning rate 0.001. The training time for $1'000$ epochs on a H100 GPU was approximately 3 hours.

### Q.6    COMPARISON TABLE WITH MORE DIGITS

We provide here the table shown in the main part Table 1 but with more digits (*not* implying that all are statistically significant): This rationalizes which entries in Table 1 are dark green and which ones are light green. Note that this result is included only for transparency, since the reported digits here go beyond the significant digits.

Table 3: Statistical distances of DyDM compared to standard models, as in Table 1 but with more digits.

| Dataset | WL-Bimodal | | Community Small | | Brain | |
|---|---|---|---|---|---|---|
| Distance | $\mu$ | $\mathcal{W}_{\mathrm{marg}}$ | $\mu$ | $\mathcal{W}_{\mathrm{marg}}$ | $\mu$ | $\mathcal{W}_{\mathrm{marg}}$ |
| DyDM (ours) | **0.0166** | **0.0076** | **0.0671** | **0.0172** | **0.0455** | **0.0275** |
| EDP-GNN | 0.1342 | 0.0750 | 0.4164 | 0.1356 | 0.0723 | 0.0321 |
| GDSS | 0.2289 | 0.1252 | 0.4180 | 0.1444 | 0.3276 | 0.1191 |
| ConGress | 0.3802 | 0.1590 | 0.2741 | 0.1138 | 0.1315 | 0.0296 |
| DiGress (no trick) | 1.0568 | 0.2852 | 2.5088 | 0.4481 | 0.5686 | 0.1749 |
| DiGress (trick) | 0.0302 | **0.0073** | 0.0934 | 0.0254 | 0.1208 | **0.0285** |

## R    LEARNING DYNAMICS OF EDP-GNN

We report in Figure 11 the learning progress of EDP-GNN on the WL-bimodal dataset. We average over 4 different training and sampling runs of EDP-GNN. After $5'000$ epochs, we observe the result of EDP-GNN reported in Figure 2.

We observe that the model quickly learns the WL equivalence class (the pink line is from epoch 500 onward close to 1). The share of graph $A$ and graph $B$ samples initially increase until epoch $1'500$, but then remain low and significantly different from the ground truth (blue and green dashed lines). Importantly, a significant share of the samples are $WL$-equivalent but neither isomorphic to graph $A$ nor to graph $B$.

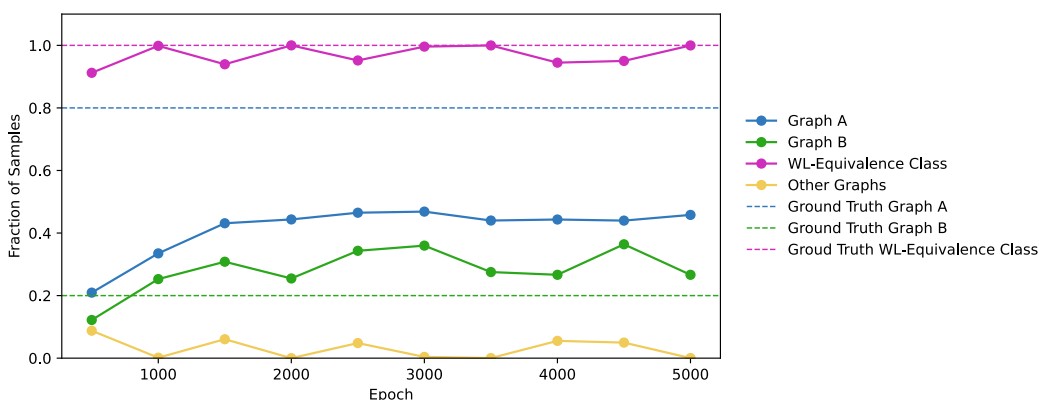

Figure 11: Learning dynamics of EDP-GNN on the WL-bimodal dataset (ground truth = $80\%$ graph A, $20\%$ graph B): The model learns very quickly (after less than 500 epochs) the WL-equivalence class, but struggles to learn graphs $A$ and $B$.

## S  TOWARDS A DIFFUSION MODEL FOR EIGENVECTOR SDE

In this appendix, we discuss a diffusion model for the Eigenvector SDE. First, in section S.1 we present some preliminaries on SDEs on $\mathrm{O}(n)$. In section S.2 we show that the Stratonovich SDE in (7) agrees with the (Eigenvector-SDE). In section S.3 we establish the time-reversal formula for the eigenvector SDE, and in section S.4 we explain the corresponding loss function.

Foundational results on diffusion models on Riemannian manifolds were established by De Bortoli et al. (2022), treating the case of Brownian motions with a drift. Moreover, recently Bertolini et al. (2025) discussed diffusion models for Lie groups acting on $\mathbb{R}^n$. However, a diffusion model for SDEs of the type of the (Eigenvector-SDE) has not been studied in the literature so far, and in this appendix we develop initial results towards establishing diffusion models for more general SDEs on Lie groups.

As before, we denote by $\mathrm{O}(n)$ the orthogonal group

$$\mathrm{O}(n) = \{X \in \mathrm{GL}_d(\mathbb{R}) \,:\, X^T = X^{-1}\},$$

where $\mathrm{GL}_d(\mathbb{R}) = \{X \in \mathbb{R}^{n \times n} \,:\, \det(X) \neq 0\}$ is the group of invertible $n \times n$ matrices. The Lie algebra of $\mathrm{O}(n)$ is the tangent space at the identity and can be calculated to be

$$\mathfrak{o}(n) = T_e \mathrm{O}(n) = \{Z \in \mathbb{R}^{n \times n} \,:\, Z^T = -Z\}.$$

Moreover, it is then easily seen that the tangent space at $X \in \mathrm{O}(n)$ is then given

$$T_X \mathrm{O}(n) = X\mathfrak{o}(n) = \{XZ \,:\, Z \in \mathfrak{o}(n)\}$$

### S.1  PRELIMINARIES ON SDEs ON $\mathrm{O}(n)$

We first briefly recall some preliminary material on SDEs, particularly Stratonovich SDEs, on Lie groups. An excellent reference for this topic is Hsu (2002), as well as the appendix in De Bortoli et al. (2022).

**Vector Fields and Stratonovich SDEs on $\mathbb{R}^n$.**  Recall that a smooth vector field $V$ on $\mathbb{R}^d$ is a map that smoothly assigns to each point a vector. Formally, we can therefore view $V$ as a smooth map

$$V : \mathbb{R}^n \to \mathbb{R}^n, \qquad x \mapsto (V_1(x), \ldots, V_n(x)).$$

Given two smooth vector fields $V$ and $U$ on $\mathbb{R}^d$, we define the covariant derivative $\nabla_U V$ of $V$ with respect to $U$ to be the vector field

$$(\nabla_U V)(x) = \frac{d}{dt}\bigg|_{t=0} V(x + tU(x))$$

$$= \left(\sum_{j=1}^n U_j(x)\frac{\partial V_1}{\partial x_j}(x), \sum_{j=1}^n U_j(x)\frac{\partial V_2}{\partial x_j}(x), \dots, \sum_{j=1}^n U_j(x)\frac{\partial V_n}{\partial x_j}(x)\right).$$

Given $d \geq 1$, denote by $W^1, \dots, W^d$ independent one-dimensional Brownian motions and let $V_1, \dots, V_d$ be vector fields on $\mathbb{R}^d$. Then the Stratonovich SDE

$$dX(t) = \sum_i V_i(X_t) \circ dW^i(t)$$

denotes the Ito SDE

$$dX(t) = \frac{1}{2}\left(\sum_i (\nabla_{V_i} V_i)(X(t))\right) dt + \sum_i V_i(X(t)) dW^i(t).$$

Just to have a concrete example in mind: The $n$-dimensional Brownian motion can be written by taking $d = n$ and considering the constant vector fields $V_i(x) = e_i$ for $e_i$ the standard basis vectors.

**Vector Fields and Stratonovich SDEs on** $\mathrm{O}(n)$**.**    In this subsection, we consider the concrete case of a Stratononvich SDE driven by left invariant vector fields on $\mathrm{O}(n)$. Indeed, a left invariant vector field $V$ is determined by a Lie algebra element $z \in \mathfrak{o}(n)$ and given by

$$V(X) = Xz \in T_X\mathrm{O}(n).$$

We know exploit that $\mathrm{O}(n)$ is embedded in $\mathbb{R}^{n \times n}$ so we can write a Lie group SDE as an SDE on $\mathbb{R}^{n \times n}$. Indeed, let $V_1, \dots, V_d$ be left invariant vector fields $V_i(X) = Xz_i$ and consider the Lie group SDE

$$dX(t) = \sum_i (X(t)z_i) \circ dW^i(t).$$

Based on (Hsu, 2002, section 1.2), we can write the above SDE as an Ito SDE on $\mathbb{R}^{n \times n}$ with the terms

$$dX(t) = \frac{1}{2}\sum_i (X(t)z_i^2)dt + \sum_i (X(t)z_i)dW^i(t). \tag{76}$$

## S.2    Writing the Eigenvector SDE as a $\mathrm{O}(n)$ Stratonovich SDE

**Proposition S.1.** *The* $\mathrm{O}(n)$ *Stratonovich SDE from* (7) *is the same as* (Eigenvector-SDE) *viewed as an SDE on* $\mathbb{R}^{n \times n}$*.*

*Proof.* This follows by a direct calculation using equation (76) for time-dependent vector fields. Recall that $E_{(\ell,k)} = e_\ell e_k^T - e_k e_\ell^T$ for $\ell < k$, where $e_i$ is the standard basis vector of $\mathbb{R}^n$ viewed as a row vector. Then denoting by $e_{ii}$ the diagonal matrix that is 1 at the $(i,i)$ entry and zero everywhere else, it holds that

$$E_{(\ell,k)}^2 = -(e_{\ell\ell} + e_{kk}).$$

Therefore the $\mathrm{O}(n)$ Stratonovich SDE from (7) is the Ito SDE on $\mathbb{R}^{n \times n}$ given by

$$dX(t) = -\frac{\alpha}{2}\sum_{\ell<k}\frac{X(t)(e_{\ell\ell}+e_{kk})}{(\lambda_k(t)-\lambda_\ell(t))^2} + \sqrt{\alpha}\sum_{\ell<k}\frac{X(t)E_{(\ell,k)}}{\lambda_k(t)-\lambda_\ell(t)}dW_{(\ell,k)}(t).$$

By writing $X(t) = (v_1(t) \dots v_n(t))$ as row vectors, the last equation is easily seen to be exactly the (Eigenvector-SDE). $\qquad\square$

## S.3 Deduction of Time Reversal Formula

Recall that the Brownian motion on $O(n)$ is given by the $O(n)$ Stratonovich SDE

$$dB^{O(n)}(t) = \sum_{\ell < k} (B^{O(n)}(t) E_{(\ell,k)}) \circ dW^{(l,k)}(t).$$

We next consider the time-scaled Brownian motion on $O(n)$ given by

$$X(t) = g(t) B^{O(n)}(t)$$

for some scalar function $g(t)$.

As in De Bortoli et al. (2022) we denote by $\nabla \log p_{T-s}$ the gradient vector field of $\log p_{T-s}$. It is shown (or rather stated that it can be shown) in (De Bortoli et al., 2022, Theorem 1) that the time reversal of the Brownian motion $Y(s) = X(T - s)$ satisfies the SDE

$$dY(s) = g(T - s)^2 \nabla \log p_{T-s}(Y(s)) \, dt + g(T - s) dB^{O(n)}(s).$$

The time reversal formula (8) is obtained by adapting the above result to our SDE (7), where the vector fields are scaled by the factors $\frac{\sqrt{\alpha}}{\lambda_k(t) - \lambda_\ell(t)}$. The final result is stated in (8).

## S.4 Loss Function

As for the eigenvalue SDE, we have no direct access to the probability measures $p_t(x_t|x_0)$. We therefore employ a loss analogous to the one used to train the eigenvalues and explained in Appendix A.

To approximate the score, we observe that similarly to (De Bortoli et al., 2022, Equation (8)),

$$\lim_{t \to s} (t-s)(\nabla \log p_{t|s})(X_t|X_s) \approx -\Sigma(t,s) \cdot \log_{O(n)}(X_t^T X_s) \quad \text{for} \quad \Sigma(t,s) = \text{diag}\left(\frac{\alpha(t-s)}{(\lambda_k(s) - \lambda_\ell(s))^2}\right),$$

where $\log_{O(n)} : O(n) \to \mathfrak{o}(n)$ is the matrix logarithm given for $X \in O(n)$ by $\log_G(X) = \sum_{k=1}^\infty \frac{(-1)^{k+1}}{k}(X - \text{Id})^k$ and we view $\Sigma(t,s)$ as a diagonal matrix with respect to the standard basis $E_{(\ell,k)}$ of $\mathfrak{o}(n)$.

As discussed in the main part, for a small $h > 0$, we have the approximation

$$X(t + h) \approx X(t) \exp_{O(n)}(Z) \quad \text{with} \quad Z = \sqrt{\alpha h} \sum_{\ell < k} \frac{E_{(\ell,k)} \mathcal{N}^{(\ell,k)}(0,1)}{\lambda_k(t) - \lambda_\ell(t)} \in \mathfrak{o}(n),$$

where $\mathcal{N}^{(\ell,k)}(0,1)$ are independent samples of standard 1-dimensional Gaussians and $\exp_{O(n)}(Z) = \sum_{i=0}^\infty \frac{Z^i}{i!}$ is the matrix exponential.

Assume now that we have $N$ discretizations of the interval $[0, T]$ for some $T > 0$ that we denote for $1 \le r \le N$ as

$$0 = t_0^{(r)} < t_1^{(r)} < t_2^{(r)} < \ldots < t_{k^{(r)}-1}^{(r)} < t_{k^{(r)}}^{(r)} = T.$$

For those we have $N$ sampled paths of our SDE from (7) denoted for $1 \le r \le N$ as

$$X_0^{(r)}, X_{t_1^{(r)}}^{(r)}, X_{t_2^{(r)}}^{(r)}, \ldots X_{t_{k^{(r)}-1}^{(r)}}^{(r)}, X_T^{(r)} \quad \text{with Lie algebra increments} \quad Z_0^{(r)}, Z_{t_1^{(r)}}^{(r)}, Z_{t_2^{(r)}}^{(r)} \ldots Z_{t_{k^{(r)}-1}^{(r)}}^{(r)}$$

so that for all $0 \le i \le k^{(r)} - 1$ it holds that

$$X_{t_{i+1}^{(r)}}^{(r)} = X_{t_i^{(r)}}^{(r)} \exp_{O(n)}(Z_{t_i^{(r)}}^{(r)}) \quad \text{and therefore} \quad -Z_{t_i^{(r)}}^{(r)} = \log_{O(n)}((X_{t_{i+1}^{(r)}}^{(r)})^T X_{t_i^{(r)}}^{(r)}).$$

By the last equation, it therefore follows that

$$\nabla \log p_{t|s}(X_{t_{i+1}^{(r)}}^{(r)} | X_{t_i^{(r)}}^{(r)}) \approx \frac{\Sigma(t_{i+1}^{(r)}, t_i^{(r)}) Z_{t_i^{(r)}}^{(r)}}{t_{i+1}^{(r)} - t_i^{(r)}}.$$

Analogously to Appendix A, we therefore want to optimize the loss

$$\frac{1}{N} \sum_{i=1}^N \sum_{i=1}^{k^{(r)}-1} \frac{t_{i+1}^{(r)} - t_i^{(r)}}{T} \left\| s(X_{t_{i+1}^{(r)}}^{(r)}, t_{i+1}^{(r)}) - \frac{\Sigma(t_{i+1}^{(r)}, t_i^{(r)}) Z_{t_i^{(r)}}^{(r)}}{t_{i+1}^{(r)} - t_i^{(r)}} \right\|,$$

where the norm is the $L^2$-norm on the Lie algebra $\mathfrak{o}(n)$.

