# OpenReview forum: "Permutation-Invariant Spectral Learning via Dyson Diffusion"
_ICLR.cc/2026/Conference — ICLR 2026 Conference Withdrawn Submission_

### Official Review · Reviewer_zqUH · 2025-10-31

**Soundness:** 2
**Presentation:** 3
**Contribution:** 3
**Rating:** 6
**Confidence:** 2

**Summary:**

This paper proposes a new way to do graph diffusion by moving permutation invariance from the neural architecture to the diffusion dynamics. Instead of diffusing adjacency matrices with GNN-based architectures, the authors use random matrix theory and Dyson Brownian Motion to directly model and generate the eigenvalue trajectories of graphs.

**Strengths:**

**1. Novel problem formulation**

The paper introduces a new perspective on graph diffusion by shifting permutation invariance from neural architecture to stochastic dynamics. Leveraging Dyson Brownian Motion to model spectral evolution is grounded in random matrix theory, offering a principled alternative to the WL-limited message-passing paradigm.

**2. Addresses a fundamental limitation in graph generative models**

The work directly tackles the expressivity constraints of GNN-based diffusion models and clearly demonstrates how current architectures fail on WL-equivalent graphs. The proposed DyDM avoids these blind spots and captures spectral distributions that traditional models struggle with.

**Weaknesses:**

**1. Limited scope: generates spectra, not full graphs**

Experiments merely demonstrates spectral generation. No adjacency-level reconstruction or topology sampling is shown, so the model currently functions as a spectral generator, not a full graph generator. This may limit perceived practical impact unless downstream graph construction is demonstrated.

**2. Scalability and efficiency unclear**

The Dyson SDE requires adaptive step control and an equilibrium shooting rescue mechanism. The paper lacks of computational complexity analysis, training/inference time comparison and memory usage reports.

**3. Evaluation narrowly focused on spectrum fidelity**

Evaluaition mainly focus on spectral statistics (i.e., mean and marginal Wasserstein distance). More broader evaluation could be better. e.g., dowmstream task performance (i.e., graph generation), visual or qualitative graph samples.

**4. Limited benchmarks**

Experiments emphasize synthetic WL-equivalent cases and small community/brain datasets (i.e., for Brain datasets of size 5 to 10 vertices). No results on widely used graph-gen benchmarks (e.g., ZINC, Planetoid citation graphs, Proteins). This makes generalization claims harder to assess.

**Questions:**

1. Is full graph reconstruction via the eigenvector SDE currently feasible, or is DyDM’s scope limited to spectral generation at this stage?

2. Dyson dynamics have singular drift near eigenvalue collisions. While adaptive control handles this, how robust is the proposed solver under near-degenerate spectra, such as graphs with high symmetry or repeated eigenvalues?

---

> ### Author Response · Authors · 2025-11-21
> **Official Comment to Reviewer (1/2)**
>
> We thank the Reviewer for their constructive, positive feedback, and for highlighting the novelty and contribution of our work. We address in the revised version of the manuscript the questions raised by the Reviewer, as detailed below.
>
> > [Q1]: Is full graph reconstruction via the eigenvector SDE currently feasible, or is DyDM’s scope limited to spectral generation at this stage?
>
> This is related to the Reviewer’s first and third weakness point.
>
>
> As the Reviewer highlighted, in this work, we propose a novel way to dissect the stochastic dynamics underlying a graph diffusion model into two parts, one for the spectrum and one of the eigenvectors, which gives numerous benefits as outlined in the paper. For the implementation, we focused on the Dyson Diffusion (the spectrum) since (1) this is the more challenging SDE with a repulsive drift forcing the spectrum towards the outside of the Weyl chamber domain *unless* proper numerical care is taken, and (2) the spectrum encodes key features of the graph (as underlined, among others, by the references suggested by Reviewer 54zp).
>
> We provided in the first submission the Eigenvector SDE but did not go into further detail. While the focus of the present work remains the spectrum and how it decouples from the Eigenvectors, in the revised version, we (A) develop new theory towards a diffusion model for the eigenvectors – including a numerical scheme, an accessible loss function, and a time reversal formula – and (B) illustrate the result on $\mathrm{SO}(3)$ (see Section 6 and Appendix S).
>
> We hope that providing all those missing parts for a diffusion model on the eigenvectors shows that by using DyDM, Eigenvectors remain indeed recoverable and that we answer the Review’s’ concerns.
>
>
> > [Q2] Dyson dynamics have singular drift near eigenvalue collisions. While adaptive control handles this, how robust is the proposed solver under near-degenerate spectra, such as graphs with high symmetry or repeated eigenvalues?
>
>
> This is a very good numerical question, we can handle both, and demonstrating this was one of the motivations for the Brain dataset: This dataset contains graphs with eigenvalue multiplicity of up to three, and also has graphs with very narrow eigenvalues, the closest distinct eigenvalues have distance $0.036$ (see our description in Appendix P).
> First, the vector field for close eigenvalues is very large (in Dyson’s physical context, this corresponds to two electrons being very close together, thus having large repulsion) so that during training, this will naturally get high importance. In addition, we weigh smaller times higher for the loss. With the problem of close eigenvalues solved, we handle identical eigenvalues through perturbation (see Appendix K.2 for details).
>
> > [W2] The Dyson SDE requires adaptive step control and an equilibrium shooting rescue mechanism. The paper lacks of computational complexity analysis, training/inference time comparison and memory usage reports.
>
> We thank for this constructive critique, which we implemented now with two entirely new Sections addressing those concerns: Generally, the shooting mechanism and the skipping mechanism are needed for correctness (imagine one numerical error happening leading to leaving the Weyl Chamber; this would lead to the dynamics staying out of the Weyl Chamber for a long time until the next numerical error happens – a phenomenon that can be observed when a simple Euler Maruyama approach is taken towards the Dyson SDE) and adaptive timesteps help with accuracy. However, we note that skipping and shooting are very rare events – they are needed for correctness but do not degrade performance (we give details in the new Section I). Further, the time steps – while adaptive in nature – concentrate well (see the new Fig. 7). We also include in the revised version the resources used (in particular, we use an H100 GPU with 80GB Video RAM).
>
> To address dependency of the computational cost of an update step in terms of the graph size $n$, we added another new Section (Section H: Complexity Dependence on graph size $n$ of numerical step of Dyson-BM and Eigenvector-SDE) and compare the derived complexity to a GNN-based OU-approach.

---

> > ### Author Response · Authors · 2025-11-21
> > **Official Comment to Reviewer (2/2)**
> >
> > > [W4] Experiments emphasize synthetic WL-equivalent cases and small community/brain datasets (i.e., for Brain datasets of size 5 to 10 vertices). No results on widely used graph-gen benchmarks (e.g., ZINC, Planetoid citation graphs, Proteins). This makes generalization claims harder to assess.
> >
> > For the benchmarking, we want to stress that benchmarking in this area was addressed in a position paper by key experts in the field as challenging due to poor benchmarks, focusing on “overfitting rather than fostering generalizable insights” and “favoring narrow domains like two-dimensional molecular graphs over broader, impactful areas such as combinatorial optimization” [1]. This critique comes from a recent position paper by experts in the field. In particular, the use of  “ZINC250K” mentioned by the Reviewer is criticized. Instead, it is proposed that one should focus on more combinatorial data.
> >
> > We want to demonstrate to the Reviewer how we did our best to follow these expert recommendations: First, we point out particular challenges of common datasets (“Appendix O.1: On Undersamling”) and how we overcame them in our paper. That meant, unfortunately, that we had to re-run existing models either because they were not reporting Snapshots of their trained models [e.g. Digress and Congress], or because they chose a training-test split of a set of 100 graphs  of size up to 20 vertices each, which they trained on 80 graphs and tested on the remaining 20. The domain of a distribution on 20-vertex-graphs is huge  (1 graph = n*(n-1)/2 = 190 many possible variables), and training on only 80 of them and comparing to 20 others leads to clear overfitting and undersampling, hence we could not faithfully use such a work.
> >
> > Consequently, we retrained each model on each dataset: The Reviewer can see that the benchmarking we report (Tables 1 and 3) are not just copied values, but actually manually benchmarked results, including hyperaparameter tuning. We thus hope that the significant resources that went into the Benchmarking of both, our model and other work, in line with the expert guidelines from [2] are another valuable contribution of our work.
> >
> > None of our datasets are 2d molecular datasets which contain additional (e.g. location) information – since this information would distract from the task of *graph* learning, hence in line with [2] our datasets all are of combinatorial nature.
> >
> > The particular choice of datasets is as follows: the WL-Bimodal dataset is a very simple (two non-isomorphic graphs only) dataset, which contains the graph nature of the problem (i.e. many *permutations* of adjacency matrices representing graphs A and B) and which contains the challenge of Weisfeiler-Leman equivalence. This is thus a “simple” but synthetic dataset, where, however, results can be very well quantified (how many samples are close to either A or B?). The “Community Small” dataset we provide due to its popularity in graph diffusion model works [2,3,4], however, with the undersampling fixed by transforming this dataset into a memorization dataset. We then pick the Brain data to demonstrate scalability: Brain does not contain particularly challenging graphs, however, it contains $15,000$ graphs. Thus, we (1) test beyond memorization how good a model generalises and (2) that our model can scale to large quantities of graphs ($15,000$ many).
> >
> > In response to Reviewers’ requests, we continued this benchmark by now reporting two more models on the Brain dataset (see Table 1 and 3).
> >
> > [1] Maya Bechler-Speicher, Ben Finkelshtein, Fabrizio Frasca, Luis M¨uller, Jan T¨onshoff, Antoine Siraudin, Viktor Zaverkin, Michael M. Bronstein, Mathias Niepert, Bryan Perozzi, Mikhail Galkin, and Christopher Morris. Position: Graph Learning Will Lose Relevance Due To Poor Benchmarks. In International Conference on Machine Learning Position Paper Track, 2025
> >
> > [2] Chenhao Niu, Yang Song, Jiaming Song, Shengjia Zhao, Aditya Grover, and Stefano Ermon. Permutation Invariant Graph Generation via Score-Based Generative Modeling. In International Conference on Artificial Intelligence and Statistics, 2020.
> >
> > [3] Jaehyeong Jo, Seul Lee, and Sung Ju Hwang. Score-based Generative Modeling of Graphs via the System of Stochastic Differential Equations. In International Conference on Machine Learning, 2022
> >
> > [4] Jiaxuan You, Rex Ying, Xiang Ren, William Hamilton, and Jure Leskovec. GraphRNN: Generating Realistic Graphs with Deep Auto-regressive Models. In International Conference on Machine Learning, 2018.

---

> > > ### Comment · Reviewer_zqUH · 2025-11-28
> > >
> > > Thanks for the authors' rebuttal. My concerns have been resolved and no additional inquiries arise. I wish to keep my score.

---

### Official Review · Reviewer_STxX · 2025-10-31

**Soundness:** 3
**Presentation:** 3
**Contribution:** 3
**Rating:** 6
**Confidence:** 4

**Summary:**

The paper proposes the Dyson Diffusion Model (DyDM), which analytically decomposes an Ornstein–Uhlenbeck diffusion on graph adjacency matrices into eigenvalue (spectrum) and eigenvector dynamics via random matrix theory: eigenvalues follow Dyson Brownian Motion (permutation-invariant), so the score can be learned with any architecture while preserving non-spectral information through an eigenvector SDE—yielding more accurate spectrum learning than GNN/transformer graph diffusion and mitigating GI/WL expressivity blind spots.

**Strengths:**

1. Principled permutation invariance via dynamics: Shifts the inductive bias from architecture to the SDE itself (DBM), enabling architecture-agnostic spectral learning and avoiding GI/WL limitations.

2. Information-preserving and effective: Retains non-spectral content (theorem-backed) and empirically outperforms GNN/transformer baselines on spectrum predict

**Weaknesses:**

1. Scope limited to spectral generation: The current experiments focus on generating graph spectra, without demonstrating adjacency-level reconstruction or topology sampling. As a result, the method operates primarily as a spectral generator rather than a full graph generative model. Demonstrating downstream graph construction would strengthen the practical impact.

2. Scalability and computational efficiency not established The proposed approach involves adaptive step-size control and a shooting mechanism for stability, yet the paper does not report computational complexity, runtime comparisons, or memory usage. Without such analysis, it remains unclear how well the method scales to larger graphs or real-world workloads.

**Questions:**

See the weakness

---

> ### Author Response · Authors · 2025-11-21
> **Official Response to Reviewer**
>
> We thank the Reviewer for their positive feedback and for highlighting the strengths of the newly proposed method. With the revised manuscript, we would like to address the Reviewer’s remaining weakness points:
>
> > [W1] Scope limited to spectral generation: The current experiments focus on generating graph spectra, without demonstrating adjacency-level reconstruction or topology sampling. As a result, the method operates primarily as a spectral generator rather than a full graph generative model. Demonstrating downstream graph construction would strengthen the practical impact.
>
> As the Reviewer highlighted, in this work, we propose a novel way to dissect the stochastic dynamics underlying a graph diffusion model into two parts, one for the spectrum and one of the eigenvectors, which gives numerous benefits as outlined in the paper. For the implementation, we focused on the Dyson Diffusion (the spectrum) since (1) this is the more challenging SDE with a repulsive drift forcing the spectrum towards the outside of the Weyl chamber domain *unless* proper numerical care is taken, and (2) the spectrum encodes key features of the graph (as underlined, among others, by the references suggested by Reviewer 54zp).
>
> We provided in the first submission the Eigenvector SDE but did not go into further detail. While the focus of the present work remains the spectrum and how it decouples from the Eigenvectors, in the revised version, we (A) develop new theory towards a diffusion model for the eigenvectors – including a numerical scheme, an accessible loss function, and a time reversal formula – and (B) illustrate the result on $\mathrm{SO}(3)$ (see Section 6 and Appendix S).
>
> We hope that providing all those missing parts for a diffusion model on the eigenvectors shows that by using DyDM, Eigenvectors remain indeed recoverable and that we answer the Review’s’ concerns.
>
>
> > [W2] Scalability and computational efficiency not established The proposed approach involves adaptive step-size control and a shooting mechanism for stability, yet the paper does not report computational complexity, runtime comparisons, or memory usage. Without such analysis, it remains unclear how well the method scales to larger graphs or real-world workloads.
>
>
> We thank the Reviewer for this comment, and we established in response two entire new Sections in the Appendix addressing this. Generally, the shooting mechanism and the skipping mechanism are needed for correctness (imagine one numerical error happening leading to leaving the Weyl Chamber; this would lead to the dynamics staying out of the Weyl Chamber for a long time until the next numerical error happens – a phenomenon that can be observed when a simple Euler Maruyama approach is taken towards the Dyson SDE) and adaptive timesteps help with accuracy. However, we note that skipping and shooting are very rare events – they are needed for correctness but do not degrade performance (we give details in the revised Section I). Further, the time steps – while adaptive in nature – concentrate well (see the new Fig. 7).
>
> To address dependency of the computational cost of an update step in terms of the graph size $n$, we added another new Section (Section H: Complexity Dependence on graph size $n$ of numerical step of Dyson-BM and Eigenvector-SDE) and compare the derived complexity to a GNN-based OU-approach.
>
> We thank the Reviewer for their constructive feedback and hope to have addressed all of the Reviewer's concerns with the major additions of the revised manuscript.

---

### Official Review · Reviewer_Lp4a · 2025-11-08

**Soundness:** 3
**Presentation:** 3
**Contribution:** 2
**Rating:** 4
**Confidence:** 3

**Summary:**

The paper proposes a new diffusion model for graph generation, called the Dyson Diffusion Model (DyDM), which aims to achieve a permutation-invariant diffusion process.
The model is based on the Dyson Brownian Motion, a modified version of the Ornstein–Uhlenbeck (OU) process that includes a repulsive force between eigenvalues, allowing the diffusion to evolve in the spectral domain in a physically coherent manner.
The central idea is to shift permutation invariance from the model architecture (as done in GNNs or transformers) to the diffusion dynamics itself, by operating directly on the spectrum of the graph adjacency matrix.

**Strengths:**

The paper is well written and clearly organized.
The proposed formulation is interesting and promising, as it represents a relevant step toward a better understanding of permutation-invariant graph generation based on spectral properties.
Moreover, the derivation grounded in the Dyson Brownian Motion provides the model with a solid theoretical foundation (although I did not verify every formula in detail).

**Weaknesses:**

The work introduces a theoretically elegant idea, but unfortunately, it falls short in terms of experimental validation (expected for this venue).
- The model operates only on eigenvalues and does not reconstruct graphs. This limits its practical relevance, as it only generates spectral distributions, not concrete graph structures.
- The experimental evaluation is limited, as it does not include standard benchmarks (e.g., QM9, ZINC, ENZYMES) nor significant competitors in spectral generation (e.g., SPECTRE, GGSD).
- A brief intuitive explanation of the Ornstein–Uhlenbeck process would improve clarity, since it represents the theoretical basis of the model.
- The advantage of using Dyson diffusion over other spectral models (e.g., those based on Transformers) should be better motivated.
In particular, both types of models are permutation-invariant, but differ only in how they treat the physics of the spectrum: the Dyson Diffusion Model does not introduce a new kind of invariance, but rather a more physically consistent formulation of diffusion. However, it does not demonstrate a clear empirical benefit over existing approaches.
- The related work section omits important recent studies on spectral reconstruction, such as:

  -Minello, Giorgia, Alessandro Bicciato, Luca Rossi, Andrea Torsello, and Luca Cosmo. Generating Graphs via Spectral Diffusion. Proceedings of the Thirteenth International Conference on Learning Representations (ICLR 2025).


  -Martinkus, Karolis, Andreas Loukas, Nathanaël Perraudin, and Roger Wattenhofer. Spectre: Spectral Conditioning Helps to Overcome the Expressivity Limits of One-Shot Graph Generators. ICML 2022.

**Questions:**

The authors mention that the approach could be extended to other matrices beyond the adjacency one (e.g., Laplacian or normalized Laplacian).
I was wondering whether the authors conducted any preliminary exploration to at least gain some intuition about how the method would behave with these alternative spectral representations.

*Minor* Line 77, page 2: “we show that an OU diffusion on the graph can be dissected into diffusion of the …”
The acronym OU, referring to the Ornstein–Uhlenbeck process, should be explicitly defined when first introduced.

---

> ### Author Response · Authors · 2025-11-21
> **Official Response to Reviewer (1/3)**
>
> We thank the Reviewer for their thorough reading of our paper and the constructive feedback. We appreciate the Reviewer’s valuing of our theoretically founded contribution, and in particular that the Reviewer finds that we provide a “relevant step toward a better understanding of permutation-invariant graph generation based on spectral properties”. We address the questions and weaknesses highlighted by the Reviewer in our revised version and describe them in detail below.
>
> > [Q1] The authors mention that the approach could be extended to other matrices beyond the adjacency one (e.g., Laplacian or normalized Laplacian). I was wondering whether the authors conducted any preliminary exploration to at least gain some intuition about how the method would behave with these alternative spectral representations.
>
> We thank the Reviewer for motivating us to pursue this further. We implemented this now on all three datasets and generated an entire new section “O. Laplacian Spectrum” in the appendix on this. Our model works perfectly well on the Laplacian and we report in the Section O detailed results on the distribution of the generated spectra. We observe that the distribution is well learned, and key features of the Laplacian can be seen: For instance, the smallest eigenvalue is always $0$, as it should be. Similarly, this approach allows us to observe with the Fiedler Value the algebraic connectivity of the graph.
> In summary, we thank the Reviewer for raising this question, since this demonstrates also numerically the applicability of our method beyond the adjacency matrix.
>
> > [Q2] Minor Line 77, page 2: “we show that an OU diffusion on the graph can be dissected into diffusion of the …” The acronym OU, referring to the Ornstein–Uhlenbeck process, should be explicitly defined when first introduced.
>
> We thank the Reviewer for highlighting this, which is now fixed.
>
> Related to that, the Reviewer raises the following weakness
>
> > [W3] A brief intuitive explanation of the Ornstein–Uhlenbeck process would improve clarity, since it represents the theoretical basis of the model.
>
> We thank the Reviewer for this remark and added a brief intuitive explanation following the introduction of the Ornstein-Uhlenbeck process in Equation (1), by describing explicitly the role of the drift (as a linear restoring drift), pushing the dynamics back to its mean.
>
>
> > [W1] The model operates only on eigenvalues and does not reconstruct graphs. This limits its practical relevance, as it only generates spectral distributions, not concrete graph structures.
>
> We acknowledge the Reviewer’s point that practical utility requires full graph reconstruction (both eigenvalues and eigenvectors). Our methodological contribution relies on dissecting the dynamics into these two components. We initially prioritized the spectrum (Dyson diffusion) because it presents the primary mathematical challenge—specifically, the singular repulsive drift that requires careful numerical treatment to remain within the Weyl chamber. However, to fully address the practical relevance and enable reconstruction, the revised manuscript now includes a comprehensive treatment of the Eigenvector SDE in the new Section 6. We have added: 1) A theoretical framework for the eigenvector diffusion. 2) A practical implementation including a numerical scheme and tractable loss function. 3) A time-reversal formula and experimental validation on $\mathrm{SO}(3)$. With these additions, DyDM provides a complete framework for a graph generative model.

---

> > ### Author Response · Authors · 2025-11-21
> > **Official Response to Reviewer (2/3)**
> >
> > > [W5] The related work section omits important recent studies on spectral reconstruction, such as: -Minello, Giorgia, Alessandro Bicciato, Luca Rossi, Andrea Torsello, and Luca Cosmo. Generating Graphs via Spectral Diffusion. Proceedings of the Thirteenth International Conference on Learning Representations (ICLR 2025).
> >  -Martinkus, Karolis, Andreas Loukas, Nathanaël Perraudin, and Roger Wattenhofer. Spectre: Spectral Conditioning Helps to Overcome the Expressivity Limits of One-Shot Graph Generators. ICML 2022.
> >
> > This is related to the following weakness point “The advantage of using Dyson diffusion over other spectral models (e.g., those based on Transformers) should be better motivated. In particular, both types of models are permutation-invariant, but differ only in how they treat the physics of the spectrum: the Dyson Diffusion Model does not introduce a new kind of invariance, but rather a more physically consistent formulation of diffusion.”.
> >
> > We thank the Reviewer for highlighting these relevant papers. We have updated the Related Work section to discuss both Generating Graphs via Spectral Diffusion (GGSD) [Minello et al., 2025] and SPECTRE [Martinkus et al., 2022]. We address the specific comparison and advantages of our Dyson-based approach below:
> >
> > 1. Comparison with GGSD (Minello et al., 2025): While GGSD also operates on the spectral decomposition at $t=0$, our methods diverge fundamentally in how they treat the dynamics.
> >
> >     - GGSD approach: Treats the spectrum and eigenvectors essentially as data features to be learned via standard diffusion. This ignores the intrinsic geometric constraints, forcing the model to 'learn' mathematical tautologies (e.g., that eigenvectors must be orthogonal). This results in a parameterization with $n^2 + n$ degrees of freedom trying to learn a symmetric matrix with only $n(n+1)/2$ degrees of freedom—effectively doubling the necessary learning complexity.
> >
> >     - Our approach (DyDM): We model the actual stochastic dynamics of the eigensystem. Our Eigenvector SDE is defined intrinsically on the Lie Group $\mathrm{O}(n)$, automatically preserving orthonormality and adhering to the correct $n(n-1)/2$ degrees of freedom. By respecting the geometry of the problem, our model does not waste capacity relearning basic linear algebra properties, unlike GGSD. We have clarified this distinction in the revised Related Work section.
> >
> > 2. Comparison with SPECTRE (Martinkus et al., 2022): We have added SPECTRE to our discussion as a key predecessor in spectral conditioning. However, we note that SPECTRE utilizes a GAN-based architecture rather than a diffusion framework. Recent benchmarks (e.g., in the DiGress/ConGress papers [1]) have demonstrated that diffusion-based approaches generally outperform GAN-based baselines like SPECTRE on these tasks. Thus, while related in spirit (using spectral properties of graphs), our work aligns with the more recent and effective diffusion-based paradigm.

---

> > > ### Author Response · Authors · 2025-11-21
> > > **Official Response to Reviewer (3/3)**
> > >
> > > > [W2] The experimental evaluation is limited, as it does not include standard benchmarks (e.g., QM9, ZINC, ENZYMES) nor significant competitors in spectral generation (e.g., SPECTRE, GGSD).
> > >
> > > We thank the Reviewer for raising the important issue of benchmarking standards. We acknowledge that QM9, ZINC, and ENZYMES are common in the literature. However, our experimental design was deliberately chosen to align with recent expert consensus [2], which argues that the field should shift away from label-heavy molecular graphs (like QM9/ZINC) towards tasks that rigorously test combinatorial structure and generalization.
> > >
> > > 1. Choice of Datasets: We focused on datasets that isolate topological challenges rather than node-attribute memorization:
> > >     - Combinatorial Focus: As noted in [2], 2D molecular graphs often favor models that overfit to local structures or auxiliary attributes (like location). To address the Reviewer's concern regarding practical relevance, we prioritized datasets that test pure structural generation.
> > >     - Addressing Undersampling (Appendix O.1): We avoided datasets where standard splits (e.g., training on 80 graphs, testing on 20) lead to severe undersampling given the dimensionality of the graph space ($2^{N(N-1)/2}$). Instead, we selected WL-Bimodal (A controlled synthetic environment to quantify the handling of structural equivalence (isomorphism)), Community Small (a standard benchmark, but modified to fix undersampling issues, turning it into a rigorous memorization task), Brain (selected specifically to demonstrate scalability to large populations ($N \approx 15,000$ graphs), testing generalization).
> > > 2. Rigor of Comparison (Baselines):Regarding the baselines, we prioritized DiGress/ConGress and GDSS as they represent the primary state-of-the-art diffusion baselines. We selected DiGress over SPECTRE because DiGress has been shown to empirically outperform SPECTRE (see [1]). By comparing against the stronger baseline, we believe we establish the necessary validity. Crucially, we did not simply copy numbers from prior papers. We retrained and hyperparameter-tuned every baseline on every dataset to ensure a fair comparison, overcoming the inconsistent splits often found in the literature.
> > >
> > > 3. Additional Benchmarks: To further address the Reviewer's request for a broader comparison, we have now run and added results for two additional models on the Brain dataset (see updated Tables 1 and 3). We believe this rigorous, first-principles approach to benchmarking provides a more faithful evaluation of structural generation than the standard molecular suites.
> > >
> > >
> > > [1] Clement Vignac, Igor Krawczuk, Antoine Siraudin, Bohan Wang, Volkan Cevher, and Pascal Frossard. DiGress: Discrete Denoising diffusion for graph generation. In International Conference on Learning Representations, 2022
> > >
> > > [2] Maya Bechler-Speicher, Ben Finkelshtein, Fabrizio Frasca, Luis M¨uller, Jan T¨onshoff, Antoine Siraudin, Viktor Zaverkin, Michael M. Bronstein, Mathias Niepert, Bryan Perozzi, Mikhail Galkin, and Christopher Morris. Position: Graph Learning Will Lose Relevance Due To Poor Benchmarks. In International Conference on Machine Learning Position Paper Track, 2025
> > >
> > > [3] Giorgia Minello, Alessandro Bicciato, Luca Rossi, Andrea Torsello, and Luca Cosmo. Generating graphs via spectral diffusion. In The Thirteenth International Conference on Learning Representations, ICLR 2025, Singapore, April 24-28, 2025
> > >
> > > [4] Tianze Luo, Zhanfeng Mo, and Sinno Jialin Pan. Fast Graph Generation via Spectral Diffusion.IEEE Transactions on Pattern Analysis and Machine Intelligence, 46:3496–3508, 2024
> > >
> > > [5] Chenhao Niu, Yang Song, Jiaming Song, Shengjia Zhao, Aditya Grover, and Stefano Ermon. Permutation Invariant Graph Generation via Score-Based Generative Modeling. In International Conference on Artificial Intelligence and Statistics, 2020.
> > >
> > > [6] Jaehyeong Jo, Seul Lee, and Sung Ju Hwang. Score-based Generative Modeling of Graphs via the System of Stochastic Differential Equations. In International Conference on Machine Learning, 2022
> > >
> > > [7] Jiaxuan You, Rex Ying, Xiang Ren, William Hamilton, and Jure Leskovec. GraphRNN: Generating Realistic Graphs with Deep Auto-regressive Models. In International Conference on Machine Learning, 2018.

---

### Official Review · Reviewer_54zp · 2025-11-08

**Soundness:** 2
**Presentation:** 3
**Contribution:** 2
**Rating:** 4
**Confidence:** 3

**Summary:**

This work introduces the Dyson Diffusion Model (DyDM), a novel approach that learns graph spectra through an Ornstein–Uhlenbeck-driven diffusion process without relying on GNNs or graph transformers. DyDM preserves full graph information and enables the formulation of an eigenvector SDE. Experiments show that DyDM outperforms existing spectral learning methods and highlights the limitations of GNN-based graph diffusion models.

**Strengths:**

1. The paper introduces a novel diffusion model that enables graph spectrum learning without relying on GNNs or Transformers.

2. The model demonstrates strong empirical performance on distinguishing WL-equivalent but non-isomorphic graphs

**Weaknesses:**

* As mentioned in paper, "work on the set of symmetricreal matrices". The method is limited applicability to general graph types, such as such as directed or attributed graphs.

* Lack of structural recovery evaluation makes it unclear how effectively the model can reconstruct full graph structures.

* The evaluation is limited to statistical fidelity of the spectrum. This limits the understanding of the model’s utility.

* Figure 2 claims that DyDM successfully distinguishes WL-equivalent graphs A and B. However, the paper does not clearly explain how these graphs are constructed. Whether types of A differ in their spectra.


* While the Brain dataset is used , most baseline models are not evaluated on it. The paper does not explain why these comparisons are missing.

* The paper overlooks foundational spectral methods in graph theory, such as spectral clustering and Laplacian-based analysis. And does not discuss the computational cost or applicability limitations of eigenvalue decomposition.

[1] Gallagher, Ian, Andrew Jones, Anna Bertiger, Carey E. Priebe, and Patrick Rubin-Delanchy. "Spectral embedding of weighted graphs." Journal of the American Statistical Association 119, no. 547 (2024): 1923-1932.
[2] Chung, Fan RK. Spectral graph theory. Vol. 92. American Mathematical Soc., 1997.

**Questions:**

* How exactly are graphs A and B constructed? Are the adjacency matrices of different versions of graph A identical or permutation variants of the same matrix?

* Why are baseline models missing on Brain?

* Can the model handle attributed or directed graphs?

* Can the model be used for downstream tasks?

* How does the runtime of the DyDM scale with graph size (n) compared to training a GNN-based model? A runtime/memory comparison would be valuable.

---

> ### Author Response · Authors · 2025-11-21
> **Official Response to Reviewer (1/2)**
>
> We thank the Reviewer for thoroughly reading our paper and for providing this constructive feedback. We address all of the Reviewer’s questions below and updated the paper with changes highlighted in blue.
>
> > [Q1]  How exactly are graphs A and B constructed? Are the adjacency matrices of different versions of graph A identical or permutation variants of the same matrix?
>
> Adjacency matrices representing the same graph are permutations of each other. Formally, if both $M_1$ and $M_2$ are adjacency matrices representing the same graph, then there exists a permutation matrix $P$ such that $M_1 = P^T M_2 P $. We describe this already in Appendix M (now appendix P), but thank the Reviewer for pointing out that it was not clear enough, so that we now improved the presentation in Appendix P (Datasets).
>
> > [Q2] Why are baseline models missing on Brain?
>
> We thank the Reviewer for this question, which highlights the need to better clarify our experimental design regarding the Brain dataset. Our primary motivation for using this dataset was to demonstrate scalability to large-scale data ($N \approx 15,000$ graphs) with a well-defined empirical distribution (see Appendix O.1), rather than to test expressivity on 'hard' combinatorial structures. Consequently, we initially compared against only the strongest state-of-the-art baseline (DiGress with augmentation) to establish validity.However, we agree that a broader comparison provides value to the community. We have initiated runs for the remaining baseline models (which require significant compute for fair hyperparameter tuning). We have already completed and added results for two additional models to Tables 1 and 3. We are currently finalizing the remaining baselines and will update the manuscript with the full results by December 3rd. We have revised Section 4.1 to explicitly state that the Brain dataset serves as a benchmark for scalability to large graph populations, distinct from the expressivity benchmarks.
>
> By answering [Q2], we believe that we have also addressed the following weakness:
>
> > [W5] While the Brain dataset is used , most baseline models are not evaluated on it. The paper does not explain why these comparisons are missing.
>
> See reply to [Q2].
>
> >[Q3] Can the model handle attributed or directed graphs?
>
> This is related to the following Weakness listed by the Reviewer:
>
> > [W1] As mentioned in paper, "work on the set of symmetric[al] matrices". The method has limited applicability to general graph types, such as directed or attributed graphs.
>
> We appreciate the Reviewer's comment on the scope of the method. Regarding the limitation to symmetric matrices, the Reviewer is correct that our current framework is strictly designed for undirected graphs. However, regarding 'attributed' graphs, we would like to clarify two key points:
>
> Edge Attributes (Weights): Our model is not limited here; it supports real-valued edge weights, including negative weights. Therefore, it is fully applicable to general weighted undirected graphs.
>
> Node Attributes: While extending our model to learn node features is possible (e.g., via a second network as in GDSS), we intentionally focused on the graph topology. As highlighted in the literature on WL-tests [2,3], structural generation poses a fundamental challenge: without node labels, GNNs struggle to distinguish structurally similar graphs. Adding node labels effectively simplifies the task by providing auxiliary information that bypasses this structural bottleneck. Consequently, recent expert consensus [4] argues for shifting research focus away from label-rich domains (like molecular graphs) back towards pure combinatorial structure to address these core limitations. Our work targets this foundational structural challenge.
>
> Finally, we thank the Reviewer for raising this point, as it highlights a unique strength of the spectral approach: Dyson’s Brownian Motion naturally generalizes to complex numbers [1]. This allows our method to be extended to complex-weighted graphs—a capability we have strengthened in the discussion as a promising direction for future work.

---

> ### Author Response · Authors · 2025-11-21
> **Official Response to Reviewer (2/2)**
>
> > [Q4] Can the model be used for downstream tasks?
>
> The Reviewer mentions in [W6] that we did not mention “Laplacian-based analysis”. The Reviewer is right that Laplacian-based analysis is important. So far, we had run DyDM only for the spectra of adjacency matrices. To make it useful for downstream Laplacian-based analysis, we now ran our model on the Laplacian and include an entire new section in the appendix (Section O), where for instance the algebraic connectivity through the Fiedler Value can be observed.
>
> For other downstream tasks beyond the Laplacian spectrum, the Reviewer raises the following weak point:
>
> > [W3] The evaluation is limited to statistical fidelity of the spectrum. This limits the understanding of the model’s utility.
>
> We understand the Reviewer’s concern that evaluating the spectrum alone does not demonstrate the model's capability to generate complete graphs (which requires both eigenvalues and eigenvectors).
>
> We agree that while the spectrum captures critical structural information – as highlighted by the references suggested by the Reviewer [6,7] – the model’s full utility relies on recovering the eigenvectors as well. To address this limitation, we have significantly expanded the manuscript to include Section 6. This new section provides the theoretical and practical components necessary for the Eigenvector SDE, including a tractable loss function, numerical scheme, and time reversal formula.
>
> By complementing our rigorous treatment of the Dyson diffusion (spectrum) with these new results for the Eigenvector SDE, we demonstrate that our framework provides a complete "recipe" for graph generation, rather than being limited to spectral analysis.
>
> > [Q5] How does the runtime of the DyDM scale with graph size (n) compared to training a GNN-based model? A runtime/memory comparison would be valuable.
>
> We thank the Reviewer for this comment. We now investigate how the cost varies per time step in dependence on $n$ in a new section (Appendix H). We compare the scaling of the numerical update steps in terms of the graph size $n$ to a GNN-based model.
>
> Further, the Reviewer criticised:
>
> > [W6] The paper overlooks foundational spectral methods in graph theory, such as spectral clustering and Laplacian-based analysis. And does not discuss the computational cost or applicability limitations of eigenvalue decomposition.
>
> We thank the Reviewer for this remark and address those points in the revised version as follows:
>
> (1) There is indeed literature underlining the importance of spectra in graphs in general – including for other tasks, such as clustering as mentioned by the Reviewer – so that we now  improved the section summarizing related works. These references further underline the importance of spectra for downstream tasks, and we are thankful for the Reviewer reminding us to include them.
>
> (2) For the downstream task of Laplacian-based analysis, we now ran our model on the Laplacian of the graph and include an entire section on this in Section O, where for instance the algebraic connectivity can be observed.
>
> (3) We now address the computational cost as addressed below [Q5] in the newly added Sections $H$ and $I$.
>
> We hope that our replies and revisions address the Reviewer’s comments.
>
> References:
>
> [1] Freeman J. Dyson. A Brownian-Motion Model for the Eigenvalues of a Random Matrix. Journal of Mathematical Physics, 3(6):1191–1198, 1962.
>
> [2] Keyulu Xu, Weihua Hu, Jure Leskovec, and Stefanie Jegelka. How Powerful are Graph Neural Networks? In International Conference on Learning Representations, 2018.
>
> [3] Christopher Morris, Martin Ritzert, Matthias Fey, William L. Hamilton, Jan Eric Lenssen, Gaurav Rattan, and Martin Grohe. Weisfeiler and Leman Go Neural: Higher-Order Graph Neural Networks. In AAAI Conference on Artificial Intelligence, 2019.
>
> [4] Maya Bechler-Speicher, Ben Finkelshtein, Fabrizio Frasca, Luis M¨uller, Jan T¨onshoff, Antoine Siraudin, Viktor Zaverkin, Michael M. Bronstein, Mathias Niepert, Bryan Perozzi, Mikhail Galkin, and Christopher Morris. Position: Graph Learning Will Lose Relevance Due To Poor Benchmarks. In International Conference on Machine Learning Position Paper Track, 2025.
>
> [5] Tianze Luo, Zhanfeng Mo, and Sinno Jialin Pan. Fast Graph Generation via Spectral Diffusion. IEEE Transactions on Pattern Analysis and Machine Intelligence, 46:3496–3508, 2024.
>
> [6] Gallagher, Ian, Andrew Jones, Anna Bertiger, Carey E. Priebe, and Patrick Rubin-Delanchy. "Spectral embedding of weighted graphs." Journal of the American Statistical Association 119, no. 547 (2024): 1923-1932.
>
> [7] Chung, Fan RK. Spectral graph theory. Vol. 92. American Mathematical Soc., 1997.
>
> [8] Andries E. Brouwer and Willem H. Haemers. Spectra of Graphs. Universitext. Springer New York, New York, NY, 2012. ISBN 978-1-4614-1938-9 978-1-4614-1939-6. doi: 10.1007/
> 978-1-4614-1939-6

---

> > ### Comment · Reviewer_54zp · 2025-11-26
> >
> > Thanks for your detailed response.
> >
> > I still have concern that the model's capability to reconstruct full graph structures is still a theoretical promise rather than experimental fact.
> >
> > Since the authors have addressed the core limitations of the submission, I will raise my score.

---

### Official Review · Reviewer_uLbz · 2025-11-13

**Soundness:** 2
**Presentation:** 3
**Contribution:** 3
**Rating:** 6
**Confidence:** 2

**Summary:**

This paper introduces the Dyson Diffusion Model (DyDM), a novel approach for permutation-invariant graph generation via spectral learning. The key idea is to leverage Dyson’s Brownian Motion (DBM) to model the evolution of graph spectra during an Ornstein–Uhlenbeck diffusion process on adjacency matrices. By analytically decoupling the eigenvalue dynamics (which are permutation-invariant) from the eigenvector dynamics, DyDM avoids the limitations of prior graph diffusion models that rely solely on permutation-equivariant architectures like GNNs or graph transformers. The authors demonstrate that DyDM outperforms existing methods in learning graph spectra, especially on challenging graph families where traditional models fail due to Weisfeiler–Leman (WL) equivalence.

**Strengths:**

1.The theoretical foundation is strong, building on well-established results from random matrix theory and stochastic differential equations.
2.Experimental results are comprehensive and compare against multiple state-of-the-art baselines (EDP-GNN, GDSS, ConGress, DiGress) across synthetic and real-world datasets.

**Weaknesses:**

1.The method currently focuses on spectral generation and does not fully address the generation of the entire graph (eigenvectors are not modeled in the generative process, though their dynamics are derived).
2.The numerical challenges of simulating Dyson-BM (e.g., singularities, adaptive step sizing) may limit scalability or ease of implementation.

**Questions:**

1.How does DyDM scale with graph size n, especially given the need for adaptive step sizing and the conditioning on non-crossing events?
2.Have the authors considered applying DyDM to other symmetric matrix data (e.g., covariance matrices) beyond graphs?
3.How sensitive is the model to the choice of hyperparameters α,β and the time schedule?

---

> ### Author Response · Authors · 2025-11-21
> **Official Response to Reviewer**
>
> We thank the Reviewer for their appreciation of both our theoretical foundation and our extensive benchmarking. We further thank for insightful and constructive questions, which we address in the new version, as detailed below.,
>
> > [Q1] How does DyDM scale with graph size n, especially given the need for adaptive step sizing and the conditioning on non-crossing events?
>
> We thank the Reviewer for this question. To address their comments, we provide two entirely new Sections in the Appendix (Section H: Complexity dependence on graph size $n$ of numerical update step for Dyson-BM and Eigenvector-SDE, and Section I: Empirical Considerations). Herein we first analyse the complexity per time step in dependence of $n$. We then describe in Section $I$ how the number of time steps behave empirically. We report explicit numbers for the Brain dataset, including the time to generate $105’000$ paths (only 5 seconds), and which hardware we used. While the shooting and skip mechanism are indeed needed to ensure correctness (for instance, if our method would due to one numerical error leave the Weyl Chamber, it would stay with high probability outside of the Weyl Chamber), these mechanisms are triggered very rarely, as we describe in this new Section.
>
> > [Q2] Have the authors considered applying DyDM to other symmetric matrix data (e.g., covariance matrices) beyond graphs?
>
> We thank the Reviewer for this question – yes, we have considered it and mention it in the Extensions section. We have now added new experiments for the Laplacian – rather than the adjacency – spectrum in Section O and Figures 9 and 10. However, this is yet another perspective on graphs and we have not had time to apply it to other data like Covariance matrices. We hope to do so in the future.
>
> > [Q3] 3.How sensitive is the model to the choice of hyperparameters α,β and the time schedule?
>
> We show in Appendix E (“Time Rescaling”) that actually $\alpha$ and $\beta$ – although they are naturally motivated by two different parameters in the OU process – are actually connected in the following precise way: The only quantity that matters is  their ratio, say $\eta:= \frac{\alpha}{\beta}$ (this property is proven in Section E). The remaining question would be: How are $\eta$ and the final time related? The final time should be chosen such that the distribution is sufficiently close to the invariant distribution – we provide a tool to plot this – and interestingly, “Dyson’s Conjecture” provides a connection from $\eta$ to the mixing time. Hence, $\alpha,\beta$ and the final time $T$ are intrinsically connected, so that it is possible to fix $\alpha$ and $\beta$ and then determine the variable $T$ through observing the mixing time. Hence, all these three parameters are readily given, so that no hyperparameter tuning is necessary for those.
> For the time schedule, we found the schedule documented in Appendix K.1 to be sufficient for very good benchmark results in all three datasets, even in Brain.
>
> > [W1] The method currently focuses on spectral generation and does not fully address the generation of the entire graph (eigenvectors are not modeled in the generative process, though their dynamics are derived)
>
> We thank the Reviewer for noting that we derive the *dynamics* of the eigenvectors already. We provided in the first submission this Eigenvector SDE but did not go into further detail. While the focus of the present work remains the spectrum and how it decouples from the Eigenvectors, in the revised version, we (A) develop new theory towards a diffusion model for the eigenvectors – including a numerical scheme, an accessible loss function, and a time reversal formula – and (B) illustrate the result on $\mathrm{SO}(3)$ (see Section 6 and Appendix S).
>
> We hope that providing all those missing parts for a diffusion model on the eigenvectors shows that by using DyDM, Eigenvectors remain indeed recoverable and that we answer the Review’s concerns.

---

### Author Response · Authors · 2025-11-21
**Commment on revised version to all Reviewers**

In this work, we propose a novel way to dissect the stochastic dynamics underlying a graph diffusion model into two parts: one for the spectrum and one for the eigenvectors. This decomposition offers numerous benefits as outlined in the paper. Regarding the implementation, we initially prioritized the Dyson Diffusion (the spectrum) because: (1) it presents the more challenging SDE, featuring a repulsive drift that forces the spectrum out of the Weyl chamber unless specific numerical precautions are taken; and (2) the spectrum encodes key structural features of the graph (a point highlighted by the references suggested by Reviewer 54zp).

In the revised version, we further demonstrate that DyDM works beyond adjacency data on the Laplacian and demonstrate in Section O with new numerical data (Figures 9 and 10) how it thereby learns key properties of the graphs (like the algebraic connectivity).
Further, we added a complexity analysis for the numerical update step in terms of the graph size (Section H). We complete this with an empirical description of the resources needed for the numerical integrator, providing insightful statistics and describing the resources used (Section I). In addition, we revised the Related Works section, highlighting better the novel contribution of DyDM and referencing further literature. Upon Reviewers' requests, we continued the benchmark on Brain (Tables 1 and 3), where we now report the performance of two further models, and will report the performance of the remaining models until the end of the discussion period.

While there is consensus on the value of the spectral analysis, several Reviewers requested further details on the Eigenvector SDE. In our initial submission, we provided the SDE without extensive elaboration. To address this, the revised manuscript significantly expands the scope. We now (A) develop new theory towards a diffusion model for the eigenvectors – including a numerical scheme, a tractable loss function, and a time reversal formula – and (B) illustrate these results on $\mathrm{SO}(3)$. We also carried out many further improvements addressing specific comments; please find a description of those under the particular Reviewer’s response.

We address the questions regarding the Eigenvector SDE and downstream graph generation in the newly added parts of Section 6. This section introduces a rigorous framework based on Lie Group theory that extends beyond existing models. While we provide all the necessary ingredients for implementation (numerical scheme, loss function, and time reversal), we have omitted a foundational introduction to the underlying Lie algebra concepts to preserve the manuscript's focus. Given the comprehensive theory already dedicated to the Dyson SDE, we prioritized the derivation of these new mathematical results over background exposition to keep the manuscript within reasonable limits.

---

### Author Response · Authors · 2025-12-03
**Summary for new AC (1/2)**

Dear new AC,

For your convenience, we summarise here the discussion until the OpenReview incident happened and our improvements. In summary, during the rebuttal period, we contributed the following:
1. We demonstrated that DyDM works beyond adjacency data on the Laplacian and illustrated in Appendix O with new experiments (Figures 9 and 10) how it thereby learns key properties of the graphs (like the algebraic connectivity).
2. We developed all necessary theory towards a diffusion model on the eigenvectors, which is the extension that most Reviewers asked about (see Section 6 and Appendix S). This required in-depth analysis of SDEs on Lie Groups which we successfully completed. We additionally presented a numerically tractable scheme and presented experiments on $SO(3)$
3. We completed all benchmarks, so that in particular the “brain” dataset of 15,000 graphs is now evaluated on all models.
4. We added a complexity analysis for the numerical update step in terms of the graph size (Section H).
5. We completed this with an empirical description of the resources needed for the numerical integrator, providing insightful statistics and describing the particular resources we used (Section I).
6. We revised the Related Works section, referencing further literature.

Our interactions with the individual Reviewers were as follows.

*Reviewer 54zp (Initial Rating: 4, had been raised to 6):*

We responded thoroughly to all questions and concerns by the Reviewer, leading to them **having already increased their score to 6**. As you can see by their comment below, this increase in rating happened multiple days before the OpenReview incident.

*Reviewer Lp4a (Initial Rating: 4, waiting for response to rebuttal):*

Although Reviewer Lp4a has not yet responded to our rebuttal, we have addressed **every** concern raised in our revision.
Specifically:
- As requested, we extended our model to the Graph Laplacian. This yielded key insights, specifically our model's strong capability in learning the algebraic connectivity of a graph (see the new Appendix O). We thus went far beyond the Reviewer’s request which asked us “to at least gain some intuition about how the method would behave with these alternative spectral representations”. We even gave detailed quantitative histograms.
- We resolved the Reviewer’s question about the Ornstein-Uhlenbeck process
- We incorporated the literature suggested by the Reviewer, highlighting our model’s novel approach and accuracy benefits in contrast to the existing literature.
- We noted that the datasets listed by the Reviewer are strongly discouraged by a recent position paper from experts in the field [1]. Instead, we demonstrated that our benchmarking meets the highest standards by implementing the recommendations from [1] (see our "Official Response 3/3" to the Reviewer for details).
- Prompted by the Reviewer's inquiry, we developed a new theoretical framework for an eigenvector diffusion model—including a numerical scheme, a tractable loss function, and a time reversal formula—and illustrated these results on $SO(3)$ (Section 6 and Appendix S).

Since the rebuttal, we have now completed in addition the brain benchmark on **all** other models (see Table 1 and Table 3). This demonstrates our model’s scalability and accuracy on a large-scale dataset (15,000 real-world graphs). Due to our detailed implementation of the Reviewer’s feedback, we are confident that Reviewer Lp4a would have raised their score.

*Reviewer STxX (Initial Rating: 6, waiting for response to rebuttal):*

Reviewer STxX was very positive and quite confident in their assessment. Nonetheless, we  addressed the only two weaknesses they raised in detail in our Revision. In detail, that means:
- We added a new complexity analysis both theoretically (Section H) and also an empirical discussion (Section I).
- We developed the new theory for a diffusion model on the eigenvectors and demonstrated our extension on $SO(3)$.

Since we addressed the few remaining questions thoroughly, we expected Reviewer STxX to maintain or increase their score.

*Reviewer zqUH (initial rating: 6, one response received):*

We addressed all of the Reviewer’s concerns in two detailed comments, however, this Reviewer was the only one who announced in the middle of the discussion period that they want to keep their score. As a response, we continued the extensive benchmarking, and note that in the meantime, we continued benchmarking with extensive resources allocated (that meant e.g. for GDSS training and sampling from 240 different hyperparameter configuration combinations). As a result, we now have the brain dataset and thereby all benchmarks complete (see Tables 1 and 3). We are confident that this helps to further convince the Reviewer.

---

> ### Author Response · Authors · 2025-12-03
> **Summary for new AC (2/2)**
>
> *Reviewer uLbz (Initial Rating: 6, waiting for response to rebuttal):*
>
> We addressed all of the Reviewer’s questions on (1) computational cost both theoretically and empirically with explicit run times in dependence on the graph size $n$ (2) we demonstrated applications beyond the adjacency matrix through the Laplacian and (3) showed that the parameters alpha, beta, and T can be brought down to one hyperparameter due to our proof in appendix E.
> With all of the Reviewer’s concerns addressed, we were waiting for a response and expected the Reviewer to maintain or increase their score.
>
> In summary, we believe to have addressed all of the Reviewers’ concerns, and Reviewer 54zp had already **raised the score multiple days before** the OpenReview incident happened. We were hoping with confidence that due to our **detailed implementation of the feedback**, others would have followed, in particular Reviewers Lp4a, uLbz, and STxX.
>
>
>
>
> ---
> References:
>
> [1] Maya Bechler-Speicher, Ben Finkelshtein, Fabrizio Frasca, Luis M¨uller, Jan T¨onshoff, Antoine Siraudin, Viktor Zaverkin, Michael M. Bronstein, Mathias Niepert, Bryan Perozzi, Mikhail Galkin, and Christopher Morris. Position: Graph Learning Will Lose Relevance Due To Poor Benchmarks. In International Conference on Machine Learning Position Paper Track, 2025

---

### Note · Authors · 2026-01-28

I have read and agree with the venue's withdrawal policy on behalf of myself and my co-authors.

---

### Meta-Review · Area_Chair_TDab · 2025-12-18

**Summary:**

This paper proposes the Dyson Diffusion Model (DyDM), a novel graph diffusion framework that shifts permutation invariance from the neural architecture into the stochastic dynamics themselves. By leveraging results from random matrix theory, the authors analytically decompose an Ornstein–Uhlenbeck diffusion on symmetric adjacency matrices into permutation-invariant eigenvalue dynamics governed by Dyson Brownian Motion and a complementary eigenvector SDE. This perspective is original and intellectually compelling, and it directly targets known expressivity limitations of GNN- and transformer-based graph diffusion models, particularly their failure modes on Weisfeiler–Leman–equivalent graphs. The theoretical development is substantial, technically sophisticated, and generally well presented, and the empirical results convincingly demonstrate that DyDM learns graph spectra more accurately than existing baselines on both synthetic and real datasets.

The main reason for my decision is a mismatch between the central claims of the paper and what is empirically validated at submission time. While the paper argues that pushing permutation invariance into the dynamics “preserves all remaining information” and enables full graph generation via an accompanying eigenvector diffusion, this claim is not empirically demonstrated on graph data. All experimental results focus on spectral learning, and no adjacency-level reconstruction, topology sampling, or end-to-end graph generation is shown. The eigenvector component is treated in depth at a theoretical level, and a numerical illustration on a Lie group is provided, but this does not constitute empirical validation of recoverability of full graph structure in the sense required to support the paper’s broader claims. This concern was raised consistently by multiple reviewers and remains unresolved despite the substantial rebuttal.

**Reviewer Concerns:**

Reviewers largely agree that the paper presents an original and intellectually appealing approach to graph diffusion by shifting permutation invariance from neural architectures to the stochastic dynamics themselves. The use of Dyson Brownian Motion to model spectral evolution is consistently viewed as principled and well grounded in random matrix theory, and the theoretical development is generally regarded as strong and nontrivial. Reviewers also agree that the paper clearly identifies limitations of GNN- and transformer-based graph diffusion models and convincingly demonstrates that the proposed method learns graph spectra more accurately, particularly on Weisfeiler–Leman–equivalent graph families. The spectral experiments are seen as solid and well executed, and the presentation is judged to be clear given the technical depth of the work.

At the same time, there is near-unanimous agreement on a central limitation. Despite the paper’s claims that the proposed decomposition preserves all remaining information and enables full graph generation via an eigenvector diffusion, the empirical evaluation validates only spectral learning. While an eigenvector SDE is derived and theoretically motivated, reviewers repeatedly note that full graph reconstruction or adjacency-level generation is not demonstrated, and that the eigenvector component remains a theoretical promise rather than an empirically established capability. As a result, several reviewers characterize the method, in its current form, as a spectral generator rather than a complete graph generative model, which significantly limits its practical impact relative to its stated goals.

Additional concerns raised by multiple reviewers include the initial lack of clarity around computational complexity and scalability, given the numerical challenges of Dyson Brownian Motion, as well as questions about benchmarking choices. These issues are partly mitigated by the rebuttal and revisions, but they contribute to an overall sense that the paper is borderline in completeness. Overall, the reviews are highly consistent: the work is viewed as novel, theoretically strong, and promising, but not yet empirically complete with respect to its central claims, which explains the cluster of borderline scores and the mixed accept–reject sentiment.

**Reviewer Scores:**

Based on the tone of the reviews, the content of the rebuttal, and the partial follow-up comments that are visible, it is unlikely that the overall score distribution would have shifted substantially. Reviewers generally acknowledged that many of their secondary concerns were addressed during the rebuttal, particularly those related to computational complexity, scalability, and clarity of presentation. However, the dominant concern, lack of empirical validation of eigenvector recovery and full graph generation, remained unresolved for all reviewers.

Reviewer 54zp explicitly indicated that they raised their score after the rebuttal, despite still expressing reservations about the lack of experimental demonstration of full graph reconstruction. This suggests that, had they participated fully throughout, their final score would plausibly have increased from an initial borderline reject to a weak accept (approximately a 6), but not higher.

Reviewer uLbz was broadly positive about the theoretical contribution and benchmarking and rated the paper marginally above the acceptance threshold, while explicitly noting uncertainty about scalability and practical graph generation. Given that their technical questions were addressed in the rebuttal, it is plausible that their score would have remained the same or increased slightly, but their stated uncertainty about the central empirical gap makes a strong accept unlikely.

Reviewer STxX was already positive, assigning a weak accept and emphasizing the conceptual strengths of the work. Their main concerns overlapped with others—namely the lack of adjacency-level reconstruction and missing complexity analysis. Since these issues were partly addressed but not fully resolved, it is most likely that this reviewer would have maintained their score rather than increasing it meaningfully.

Reviewer zqUH also rated the paper as a weak accept while clearly identifying the absence of full graph generation and limited evaluation scope as the main weaknesses. Although the rebuttal addressed complexity and numerical robustness, the core concern about empirical completeness remained. As a result, their score would likely have stayed unchanged.

Reviewer Lp4a was among the more critical reviewers, rating the paper marginally below the acceptance threshold and emphasizing limited experimental validation, missing benchmarks, and lack of demonstrated graph reconstruction. While they acknowledged the theoretical elegance of the approach, their critique centered on precisely the issue that was not resolved in the discussion. Even with full participation, it is unlikely that their score would have increased substantially; at best, it might have moved from a weak reject to a borderline score, but not to a confident accept.

In summary, full participation in the discussion would likely have led to minor upward adjustments for one or two reviewers due to improved clarity and added analyses, but it would not have changed the overall picture. Most reviewers would have remained near their original borderline positions, because the primary blocking issue, the lack of empirical validation of the eigenvector-based full graph generation claim, persisted throughout the discussion.

---

### Decision · Program_Chairs · 2026-01-26

Reject